# The Microphysics of Clouds over the Antarctic Peninsula – Part 2: modelling aspects within Polar WRF

Constantino Listowski[1] and Tom Lachlan-Cope[2]

[1]British Antarctic Survey, NERC, High Cross, Madingley Rd, Cambridge, CB3 0ET, UK
[1]now at: LATMOS/IPSL, UVSQ Université Paris-Saclay, UPMC Univ. Paris 06, CNRS, Guyancourt, France
[2]British Antarctic Survey, NERC, High Cross, Madingley Rd, Cambridge, CB3 0ET, UK

*Correspondence to:* C. Listowski (constantino.listowski@latmos.ipsl.fr)

**Abstract.** The first intercomparisons of cloud microphysics schemes implemented in the Weather Research and Forecasting (WRF) mesoscale atmospheric model (version 3.5.1) are performed in the Antarctic Peninsula using the polar version of WRF (Polar WRF) at 5 km resolution, along with comparisons to the British Antarctic Survey's aircraft measurements (presented in Part 1 of this work, Lachlan-Cope et al., 2016). This study follows previous works suggesting the misrepresentation of
the cloud thermodynamic phase in order to explain large radiative biases derived at the surface in Polar WRF continent-wide (at 15 km or coarser horizontal resolution), and in the Polar WRF-based operational forecast model Antarctic Mesoscale Prediction System (AMPS) over the Larsen C Ice shelf at 5 km horizontal resolution. Five cloud microphysics schemes are investigated: the WRF Single-Moment 5-class scheme (WSM5), the WRF Double-Moment 6-class scheme (WDM6), the Morrison double-moment scheme, the Thompson scheme, and the Milbrandt-Yau Double-Moment 7-class scheme. WSM5
(used in AMPS), and WDM6 (an upgrade version of WSM5) lead to the largest biases in observed supercooled liquid phase, and surface radiative biases. The schemes simulating clouds in closest agreement to the observations are the Morrison, Thompson, and Milbrandt schemes for their better average prediction of occurrences of clouds, and cloud phase. Interestingly, those three schemes are also the ones allowing for significant reduction of the longwave surface radiative bias over the Larsen C Ice Shelf (Eastern Peninsula). This is important for surface energy budget consideration with Polar WRF since the cloud
radiative effect is more pronounced in the infrared over icy surfaces. Overall, the Morrison scheme compares better to the cloud observation and radiation measurements. The fact that WSM5 and WDM6 are single-moment parameterisations for the ice crystals is responsible for their lesser ability to model the supercooled liquid clouds compared to the other schemes. However, our investigation shows that all the schemes fail at simulating the supercooled liquid mass at some temperatures (altitudes) where observations show evidence of its persistence. An ice nuclei parameterisation relying on both temperature and aerosol
content like DeMott et al. (2010) (not currently used in WRF cloud schemes) is in best agreement with the observations, at temperatures and aerosol concentration characteristic of the Antarctic Peninsula where the primary ice production occurs (Part 1), compared to parameterisation only relying on the atmospheric temperature (used by the WRF cloud schemes). Overall, a realistic double-moment ice microphysics implementation is needed for the correct representation of the supercooled liquid phase in Antarctic clouds. Moreover, a more realistic INP alone is not enough to improve the cloud modelling, and water
vapour and temperature biases also need to be further investigated, and reduced.

# 1 Introduction

Tropospheric Clouds in Antarctica are amongst the least well observed on Earth due to the remote environment and harsh conditions that make field observation difficult. As a result of this, no modelling study has ever focused on comparing the performances of WRF cloud microphysics schemes to in-situ cloud measurements. Yet, this is a necessary step to improve our ability to model the Antarctic atmosphere. Better understanding the meteorology is also crucial for providing reliable forecast to aircraft or ground operations in the Antarctic.

Much attention has focused on Antarctica's energy budget in recent years notably due to the West Antarctic Ice Sheet warming (O'Donnell et al., 2011; Bromwich et al., 2013b), and on large ice mass loss (gain) recorded in West (East) Antarctica (Harig and Simons, 2015). Assessing how atmospheric driven processes affect the evolution of Antarctica's ice mass and surface energy budget requires to improve our understanding, and modelling of the clouds in that region. Importantly, changes in microphysical properties of Antarctic clouds impact the atmosphere dynamics at lower southern latitudes and even at northern latitudes, since their altered radiative properties modify the North-South temperature gradient (Lubin et al., 1998).

The Antarctic Peninsula is characterised by high mountains forming a barrier to the dominant westerlies, and which roughly extends across the longitudes 67°W to 65°W at the latitude of Rothera station (67.586°S), with altitudes up to around 2500 meters in some places. This major topography feature causes significant differences between each side in terms of temperatures (Morris and Vaughan, 2003), and precipitations (King and Turner, 1997), and also aerosols and cloud microphysics (as concluded in Part 1 of this work, Lachlan-Cope et al., 2016). Significant climate changes have been recently observed across the Peninsula during the last few decades (O'Donnell et al., 2011; Turner et al., 2016). Interestingly, oceanic driven mechanisms are the main contributor to glaciers melting in the Peninsula (Wouters et al., 2015). In this context, improving the modelling of the different components of the energy budget of the Antarctic Peninsula is required to better understand its climatological evolution, and how atmospheric-driven processes act along with ocean-driven processes to impact Antarctica's ice mass balance, and temperatures. Clouds are one of the least well understood of the atmospheric components (Boucher et al., 2013; Flato et al., 2013).

Recent studies have pointed towards Antarctic clouds being responsible for large shortwave (SW) and longwave (LW) surface radiative biases (several tens of W m$^{-2}$) in high resolution models over the whole continent (Bromwich et al., 2013a), and more specifically over the Eastern Peninsula's Larsen C Ice Shelf (King et al., 2015). Improved cloud physics allowing for realistic ice supersaturations led to lower the surface energy budget biases in the RACMO2 high resolution climate model (van Wessem et al., 2014a). King et al. (2015) compared three mesoscale models simulations over the Larsen C Ice Shelf during a summer month, and showed how they differed in the amount of cloud liquid and cloud ice that were simulated. The authors suggested that this explained the comparatively different surface biases, and they pointed towards issues in modelling the thermodynamic phase of clouds, and more specifically the supercooled liquid component (liquid maintained at $T \leq 0$°C). The modelling of the mixed-phase clouds needs to be improved in models, and the misrepresentation (underestimation) of supercooled liquid over Antarctica can be related to its poor representation over the surrounding Southern Ocean as a whole (Lawson and Gettelman, 2014).

A related issue deals with the initiation of the ice phase in clouds, which is driven by the Ice nucleating particles (INPs). They are the substrates needed to activate ice crystals growth either directly from the vapour condensing on the INP (deposition freezing), or from the freezing of supercooled droplets following immersion of, contact with, or condensation on, an INP (Hoose and Möhler, 2012). In the condensation case the INP act as a Cloud Condensation Nuclei (CCN) first to form a droplet.

Homogeneous freezing of droplets (ie without the intervention of an INP) can occur at temperatures usually believed to be colder than -38°(Hoose and Möhler, 2012) although there are possible significant effects already below -30° (Herbert et al., 2015). In a remote place like Antarctica little is known about the exact nature of the INPs although studies have been identifying various plausible sources: biological sources from the snowy surface, blowing snow, sulfate particles resulting from sea-surface emissions, mineral dust lifted from ice-free regions or brought by winds from continental landmasses at lower latitudes e.g South-America. Many candidates are found in the literature to explain the presence of INPs in Antarctica (see Bromwich et al., 2012, for a review). Similar questions arise for INPs in marine air in remote places like in the middle of the Southern Ocean (Burrows et al., 2013), which surrounds the Antarctic continent. Regarding CCNs, which are needed to activate cloud droplet growth, sea salt is known to be an efficient substrate. Interestingly, its emission in polar region's boundary layer is believed to be enhanced in places where brine-rich snow covering sea ice can be lifted by the winds (Yang et al., 2008).

In the last decades, very localized ground-measurements using in-situ or remote sensing techniques have allowed to characterise microphysical properties of clouds (particle phase, particle size, crystals shape), however these observations are sparse (Lachlan-Cope, 2010; Grosvenor et al., 2012). Ground based remote-sensing measurements provide local continuous measurements making it possible to link clouds properties to precipitations or accumulation events (Gorodetskaya et al., 2015).

Two aircraft campaigns led by the British Antarctic Survey (BAS) took place during the summer 2010, and 2011, measuring cloud properties on both sides of the Antarctic Peninsula (Lachlan-Cope et al., 2016, hereafter referred to as Part 1). Analysis of some of the 2010 flights were already presented in Grosvenor et al. (2012) with a focus on cloud ice, and secondary ice multiplication processes. These two campaigns, and the surface radiative biases pointing towards a misrepresentation of antarctic clouds within high resolution models at 5 km resolution (King et al., 2015) or at coarser resolution (Bromwich et al., 2013a) motivate this first attempt to compare some of the existing cloud microphysics schemes implemented in the Weather Research and Forecasting atmospheric model (Skamarock et al., 2009) (WRF) v3.5.1, with simulations performed at 5 km resolution. We use the polar version of WRF (Polar WRF, Hines and Bromwich, 2008), which has optimised representation for polar regions in terms of surface properties (ice, snow, sea ice, and sea water) and processes (heat transfer between the surface and the atmosphere). Polar WRF is widely used by the Antarctic community and it is used by the Antarctic Mesoscale Prediction System (AMPS, Powers et al., 2012), which is an operational forecast model that provides support for international Antarctic efforts. Both Bromwich et al. (2013a) and King et al. (2015) relied on Polar WRF, and AMPS, respectively, in their study.

In section 2 we present the model settings along with the microphysics schemes used in this work, and explain their main characteristics. In section 3 we discuss simple results of radiation biases to illustrate the importance of cloud schemes on the Peninsula energy budget. In section 4 we compare modelling results to in-situ measurements already presented in Part 1, and evaluate the performance of the cloud microphysics schemes. In section 5 we comment on the performances of the cloud

schemes investigated, discuss sensitivity issues of the present study, and comment on the aspects to consider in future work for improving cloud microphysics parameterisations in Antarctica. In section 6 the main aspects of this work are summarized.

## 2 Observations, atmospheric model, and the cloud microphysics schemes

### 2.1 Overview of the airborne observations

Two campaigns of in situ cloud measurements took place on both sides of the Antarctic Peninsula (61–73°W) in February 2010, and January 2011, respectively (Part 1). The observations were made with the British Antarctic Survey's instrumented Twin Otter aircraft (King et al., 2008) based at Rothera research station (67.586°S, 68.133°W). ERA-Interim reanalysis show an intensified northerly flow in 2011 to the west of the Peninsula, expected to bring warmer air. However colder temperatures were observed in the reanalysis, as well as in the radiosonde ascents made at Rothera (not shown), and from the aircraft

measurements (a tendency correctly reproduced in the simulations, see section 4.4). This can be explained by colder air being pulled from the Weddell Sea (to the east of the Peninsula) during the 2011's campaign, following intensification and eastward movement of the Amundsen Sea low to the west of the Peninsula (Part 1, their Figure 3). Results on average cloud properties (predominantly stratus, or altostratus) comparing both campaigns and both sides of the Peninsula are presented in Part 1, and detailed results on some 2010 flights are presented in Grosvenor et al. (2012). The aircraft was fitted with various instruments

measuring notably temperature, pressure, humidity, turbulence, radiation as well as with a Droplet Measurement Technology Cloud, Aerosol, and Precipitating Spectrometer (CAPS) (Baumgardner et al., 2001). The CAPS has a Cloud and Aerosol Spectrometer (CAS), a Cloud Imaging Probe (CIP), and a Hotwire Liquid Water Content (LWC) Sensor. The CAS measures particle size (diameter) between 0.5 and 50 $\mu$m, and the Hotwire was used to validate the supercooled LWC as derived from the CAS, which cannot discriminate between liquid and ice. Also, the CAS showed a distinct peak in the size distribution in

the range 8 and 12 $\mu$m (in diameter) indicative of drop formations. The CIP images particles with sizes between 25 $\mu$m and 1.5 mm, with 25 $\mu$m pixel resolution. Particles smaller than 200 $\mu$m in size cannot be discriminated between crystals and droplets. Their number concentration is very small compared to the CAS, and therefore they were ignored (see also section 4.3.1 of this paper for the impact on the LWC). The ice water content was calculated using the Brown and Francis mass-dimension parameterisation (Brown and Francis, 1995). More details on data processing and the derivation of the ice crystal number

concentration are given in Part 1. Finally, the CIP samples at a rate of a little less than 10 L s$^{-1}$, hence the lower limit for the measured crystal number concentration of a little more than 0.1 L$^{-1}$.

### 2.2 Model settings

Polar WRF v3.5.1 was used with a downscaling method (Figure 1a) where a 45 km resolution domain contains a smaller 15 km resolution nest, which itself contains a smaller nest at 5 km resolution centered over the regions where the 2010 and 2011

flights took place (Figure 1b). The topography is from Fretwell et al. (2013). The simulation outputs of the highest resolution domain were used for the present analysis. We work at a similar horizontal resolution to K15 (5 km), and at a higher resolution

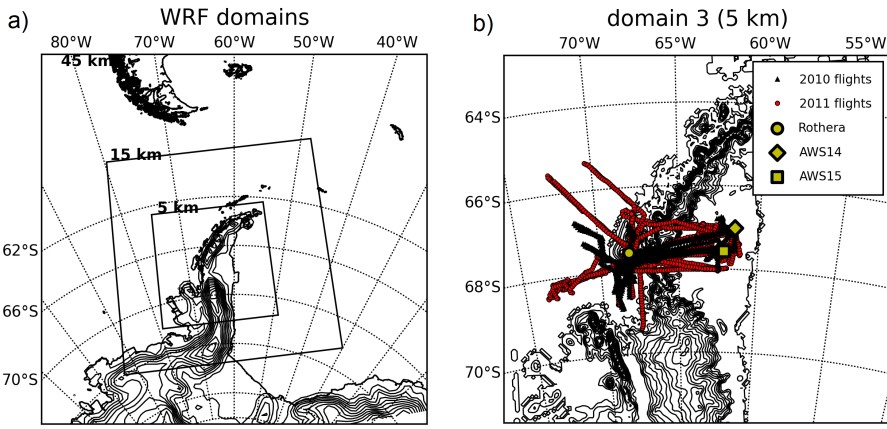

**Figure 1.** (a) WRF configurations for the three domains used for all simulations and (b) close-up on the highest resolution domain with detailed topography from Fretwell et al. (2013). The black triangles indicate the flight tracks of the 2010 campaign, while the red circles indicate the flight tracks of the 2011 campaign. The other markers indicate station (circle), and the Automatic Weather Stations (AWS) 14 (diamond) and 15 (square) located on the Larsen C Ice Shelf.

than Bromwich et al. (2013a) (60 km, and 15 km). Both studies pointed towards the clouds being responsible for the surface radiative biases measured, but they did not investigate the actual effect of using a different cloud microphysics schemes.

The simulation is one-way, in the sense that no information is passed in return from one domain to its parent domain. Table 1 summarizes the WRF settings used for the main physical processes, except for the cloud microphysics schemes, which are

addressed in section 2.3. King et al. (2015) (hereafter referred to as K15) were interested in the surface radiative biases on the Larsen C Ice Shelf on the eastern side of the Peninsula. They used outputs from AMPS (built on Polar WRF v3.0.1 for the 2011 period). For consistency we use the same shortwave and longwave radiation schemes as in K15 (Table 1). More generally we are using the set of WRF physics parameterization used by the operational model AMPS, which should be a relevant framework to testing the cloud microphysics schemes in Antarctica. Regarding the boundary layer parameterization,

Deb et al. (2016) showed that Polar WRF performances at the surface are most sensitive to the choice of the PBL scheme and show that the Mellor-Yamada-Janjic (MYJ) scheme is the best performer in terms of the temperature diurnal cycle (in west Antarctica) at 5 km resolution, and it is the one used in AMPS, and in our study.

One of the main differences with K15 is that our simulation is constrained horizontally and vertically, at the boundaries of the 45 km resolution domain, with ERA-Interim reanalysis data instead of Global Forecast System data (GFS, run by the

U.S. National Centers for Environmental Prediction). ERA-Interim is the latest global atmospheric reanalysis (Dee et al., 2011) provided by the European Centre for Medium-Range Weather Forecasts (ECMWF). This reanalysis is based on archived observations from 1989 onward. It is obtained through data assimilation into an atmospheric model running at a resolution of $0.75 \times 0.75$ degrees, which roughly corresponds to 30 km in longitude by 80 km in latitude, at the latitude of Rothera station

(67.586°S). Bromwich et al. (2013a) showed that using ERA-Interim reanalysis for initial, and boundary conditions produces the best skills within Polar WRF.

We ran two sets of simulations. The first set spans the period 1st February 2010 to 5th of March 2010, and the second set goes from 1st of January 2011 to 11th of February 2011 (the first two days were not included in the analysis as part of the model spin-up). These two periods cover the time of the two aircraft campaigns (see Part 1), including the period during 2011 when a camp was set up on the Larsen C Ice Shelf, close to the Automatic Weather Station fourteen (AWS14, see Figure 1b), as described in K15.

## 2.3  Cloud microphysics schemes

We used five different microphysics scheme to assess their ability to model realistic clouds across the Antarcic Peninsula. None of the WRF microphysics schemes has been specifically developed for Antarctic clouds modelling. As no work has been done so far on comparing microphysics schemes implemented in Polar WRF with respect to their performances for Antarctic clouds, we used them as such with no modification. This appeared to be the most reasonable first step that can then help guide further development of Antarctic clouds microphysics modelling.

Generally speaking, each scheme is a bulk microphysics parameterisation (BMP) where either mass (Single-Moment – SM – scheme), or both mass and number density (Double-Moment – DM – scheme) of the various types of hydrometeors are independently predicted. In the DM case, the scheme allows for a more realistic behaviour of clouds. Indeed, predicting both the mass and the number density of hydrometeors allows the average particle size to be predicted, which in turn allows the modelling of all size-dependent processes like sedimentation, accretion, growth to be improved (Igel et al., 2015). All schemes have non-precipitable, and precipitable hydrometeors. The former (cloud droplets, ice crystals) are considered as having zero sedimentation velocity in the collection or accretion processes, in contrast to the latter (rain drops, snow crystals, graupel, or hail) that act as collector particles. Finally, we did not use any microphysics radius bin model (as opposed to the BMPs). They predict the evolution of cloud particles within given size bins, and allow for the prediction of the actual particle size distributions. Bin models are missing from WRF v3.5.1. However, a bin model is more demanding in terms of computer time, and BMPs are used in current global or regional atmospheric models, and in operational forecast models like AMPS. Table 2 highlights some aspects of the cloud microphysics schemes investigated in this study.

The actual default microphysics scheme of WRF is WRF Single-Moment 3 (WSM3), which has been discarded here because it does not allow for the existence of supercooled liquid droplets. Thus, our default reference scheme is the WRF Single-Moment 5 (WSM5) which allows for mixed-phase cloud formation (Hong et al., 2004). WSM5 is a SM scheme for all the hydrometeors. It is used in the operational model AMPS.

The WRF double-moment 6 (WDM6, Lim and Hong, 2010) is an improvement of WSM5, in which droplets and rain are both treated with DM schemes, graupel is included, and all the ice phase particles are treated with a SM scheme. It is used here in order to test the improvement of the prediction of the supercooled liquid phase that one could expect from the use of a more sophisticated parameterisation for the liquid phase (DM instead of SM as in WSM5).

The Morrison scheme (Morrison et al., 2005; Morrison et al., 2009) is a full DM scheme for all icy hydrometeors, and rain, and SM for water droplets. The Morrison scheme requires the coupling to the WRF Chemistry module (Peckham et al., 2011) in order to act as a DM scheme for the cloud droplets, and since such coupling was not available we used the Morrison scheme as a SM scheme for the water droplets. This scheme is used in the Arctic System Reanalysis (ASR), which is based on

Polar WRF as well. It slightly improved the modelling of the clouds in the northern polar summer compared to WSM5 at 30 km resolution (Wesslén et al., 2014), and this paper investigates its ability to better represent the clouds in the southern polar region.

The Thompson scheme (Thompson et al., 2008) has a state-of-the art parameterisation of snow, which relies on extensive flight measurements, and it uses a more realistic size dependent density for snow particles. The latter are treated as non-

spherical, and their density decreases with increasing size. This was identified as having a major influence on the production of supercooled drops mainly because of a decreased efficiency in the riming process resulting in longer lasting supercooled drops (Thompson et al., 2008).

Finally, The Milbrandt-Yau scheme (Milbrandt and Yau, 2005a, b) (hereafter designated as Milbrandt) is a full double moment scheme (with the shape parameters of the particle distribution being fixed). It is used here in order to test the ability

of a full double moment scheme to predict supercooled drops better than the Morrison or the Thompson schemes.

Table 3 details the way the cloud schemes treat the initiation of the cloud ice phase, and the cloud liquid. The initiation of the ice phase is the most complex part, and it relies on INP parameterisations. They diagnose the number of INPs, hence the number of activated crystals, accounting for the various freezing modes described in the introduction. The INP parameterisations rely on the atmospheric temperature only. They are used in various ways by the different cloud microphysics schemes as illustrated

in Table 3. They deal with primary ice production (droplets or vapour converted to ice through interaction with INPs), as opposed to secondary ice processes, which result from the interaction of already formed crystals with other crystals or with supercooled droplets. Finally, the liquid phase relies on a fixed number of droplets or a predicted number of activated CCN (hence number of drops), depending on the cloud scheme. The liquid phase is formed after the ice microphysics is computed provided there is still an excess of vapour compared to equilibrium (ie if $S_w$>1, where $S_w$ is the saturation ratio with respect

to liquid water).

## 3    Preliminaries : results in radiation biases

Large biases in both surface downward shortwave (SW, solar flux) and longwave (LW) radiation were reported east of the Peninsula over the Larsen C Ice Shelf by K15. The authors compared the summertime surface energy budget as simulated for January 2011 by three mesoscale models: AMPS, The UK Met Office Unified Model (UM) (see Wilson and Ballard, 1999, for

the cloud scheme), and the Regional Atmospheric Climate MOdel (RACMO2) version 2.3 (see van Wessem et al., 2014a, for the cloud scheme). A field camp was established close to AWS14 (see Figure 1b) where radiosonde ascents allowed the water vapour column density to be calculated. AWS14 is fitted with SW and LW radiometers. K15 showed that all models mostly overestimated SW radiation by several tens of W m$^{-2}$ (positive bias) while they underestimated LW radiation (negative bias).

They pointed towards the lack of simulated clouds that blocked the incoming shortwave solar radiation, and emitted thermal radiation back to the surface. The only exception was noted for the UM model which had several tens of W m$^{-2}$ of negative bias in SW suggesting an overestimation of the cloud cover. AMPS simulated clouds predominantly composed of ice with very little or even zero liquid water, during this period over AWS14, providing an explanation to the very large surface radiative biases, especially in SW to which small droplets are the most responsive. Ice clouds, however, were simulated, and K15 pointed towards a misrepresentation of the actual phase of the clouds to explain the biases observed.

Following K15, Table 4 shows average biases of daily averaged SW and LW downward surface fluxes. They were derived by subtracting the observed value to the modelled value. Three sites were selected: British Antarctic Survey's Rothera station, on the western side of the Antarctic Peninsula, and two Automatic Weather stations - AWS14 and AWS15 - on the Eastern side of the Peninsula on the Larsen C ice shelf (see Figure 1b). Both AWS are about 70 km apart on the ice shelf. Table 4 also indicates whether the difference between WSM5 (used in AMPS) and the other schemes is statistically significant (with a Student t-test).

## 3.1 The particular case of AWS14 in January 2011

We first compare results obtained by K15 with the AMPS model over the period 8 January 2011 to 8 February 2011 at AWS14 (see their Table 3), to our results obtained with the WSM5 scheme over the same period (Table 4, fourth column of results). Their computed biases are 56 W m$^{-2}$ and -10 W m$^{-2}$ in SW and LW, respectively). Ours are 53 W m$^{-2}$ and -20 W m$^{-2}$, respectively . Discrepancies in biases can result from the combination of different settings in the AMPS (forcing, number of vertical levels, domain boundaries). However, we do obtain the same orders of magnitude and same signs of biases as K15, consistent with a lack of clouds. A striking result is that the Morrison scheme reduces the biases in both SW and LW in a statistically significant way at the 99% level, while the Milbrandt, and the Thompson schemes reduce it significantly in LW only.

Figure 2 (bottom) shows the cloud liquid mass integrated over the entire atmospheric column (kg m$^{-2}$) for the different simulations as a function of time, in the model gridbox corresponding to the AWS14 location in 2011. Figure 2 (top) shows the simulated column density of water vapour compared to the radiosonde ascent measurements from the field camp at AWS14 (presented in K15, and plotted in their Figure 7). The modelled water vapour is consistent with observations in terms of trend and value (within $\pm 1$ kg m$^{-2}$), between day 8 and day 32. The simulations give similar values within $\pm 1$ kg m$^{-2}$, except between day 15 and day 22 (where no observation is available), and all the simulations capture the sharp increase by 6 kg m$^{-2}$ measured around day 28. Using the Morrison scheme, twice to four times more liquid cloud mass is simulated than when using the WSM5 scheme (Figure 2 bottom). The Milbrandt, and Thompson schemes lead to intermediate amounts between WSM5 and Morrison, and WDM6 is similar to WSM5. The larger amount of liquid clouds simulated with the Morrison scheme compared to WSM5 is consistent with its smaller SW and LW biases at AWS14 in 2011 (Table 4). This is also in line with K15's conclusion that the thermodynamic phase of the clouds was responsible for the SW and LW biases they found in AMPS over the Larsen C. The Thompson scheme, and the Milbrandt schemes do have lower SW biases than WSM5 as well, however the

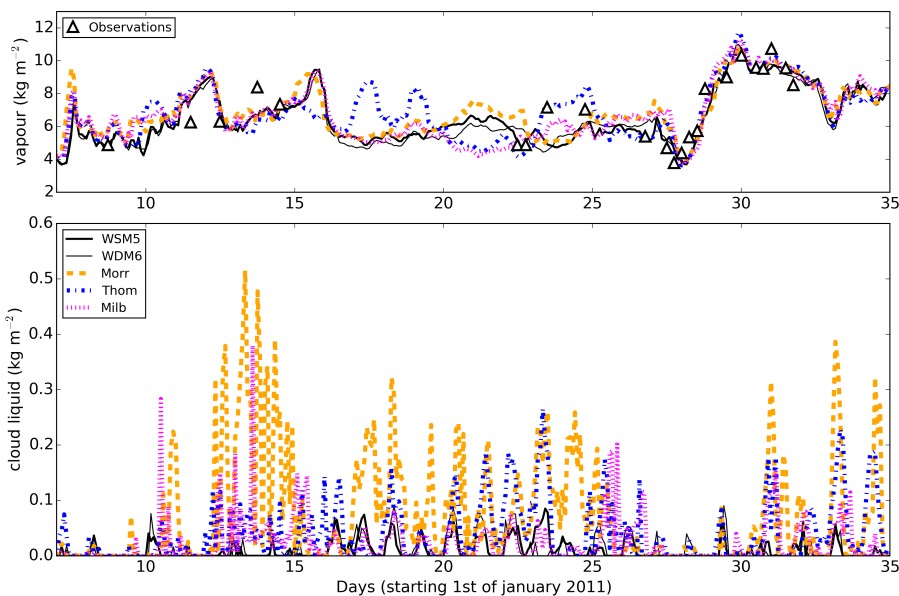

**Figure 2.** (Top) Time series of column density of water vapour (kg m$^{-2}$) for the different simulations computed in the model gridbox corresponding to the AWS14 location, in 2011, along with the radiosonde measurements from the field camp (described in K15, see their Figure 7). (Bottom) Time series of the column density of the cloud liquid (kg m$^{-2}$) for the different simulations.

improvement is smaller than with the Morrison scheme, and less (or not) statistically significant. However, it is still significant for LW radiation.

The total ice mass is similar from one scheme to another, as shown by Figure 3 (top). However an important difference arises when considering the cloud ice crystals mass only (ie the pristine ice – ignoring the main precipitable particles like snow and graupel particles); WSM5 and WDM6 simulate three to four times more ice crystals mass than the other schemes (Figure 3, bottom). The Milbrandt scheme leads only occasionally to as much ice mass as WSM5, and WDM6, around the 19th, the 25th and the 30th of January. Graupel is mainly absent except when using the Milbrandt schemes which leads to low amounts around 0.05 kg m$^{-2}$ on rare occasions (not shown). Overall, the main difference in the cloud microphysics between the various simulations at AWS14 is the ability of the cloud schemes to sustain supercooled liquid drops, which in turn can explain differences in the SW and LW surface biases. The other difference lies in the distribution of the mass within the total ice phase between cloud ice crystals and snow particles.

### 3.2 General results in radiation biases

For the eastern side of the Peninsula (AWS14 and AWS15), the biases shown in Table 4 (right part) demonstrate the importance of the choice of the microphysics scheme for the surface energy budget of the Larsen C in Polar WRF. Similar biases (sign, and order of magnitude) are observed at a given year and for a given scheme, between AWS14 and AWS15. This is consistent with the stations being 70 km apart from each other on the ice shelf, which consists of a relatively flat surface covered with

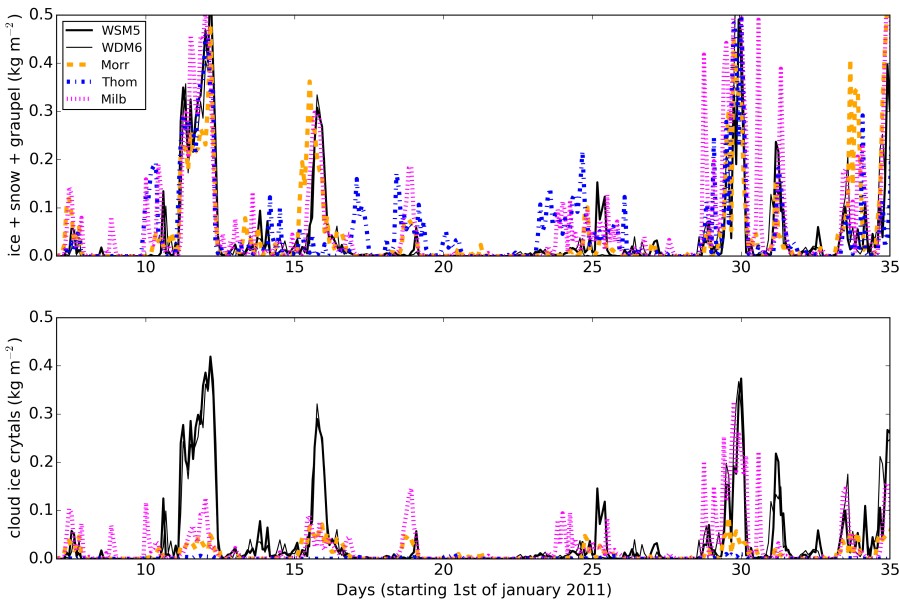

**Figure 3.** (Top) Same as Figure 2 for the total ice phase (ice, snow, and graupel particles). (Bottom) Same as Figure 2 for cloud ice particles only.

snow, and where the large scale influences are likely to be similar in absence of significant local variations in the topography or the nature of the surface. A remarkable result is that the LW bias is significantly reduced during both periods of interest on the Larsen C ice shelf using the Morrison, Thompson, or Mildbrandt schemes, compared to WSM5 (or WDM6 – not shown) as can be seen from the lower right part of Table 4. However, the Thompson, and the Milbrandt schemes still have a negative LW
bias, while Morrison's is slightly positive and gives on average the smallest LW bias at both AWS stations for both years. The standard deviation of daily-averaged measurements remains high but statistical tests show that the differences between WSM5 and the three other schemes are significant, mainly at the 99% level in 2011, and mainly at the 95% level in 2010. The SW bias is significantly reduced only with the Morrison scheme in 2011. However no improvement occurs in 2010 for the SW bias. The Milbrandt, and the Thompson scheme's SW biases are slightly lower in 2011 with differences to WSM5 that are significant
at the 90% level. In 2010 all the schemes have a large negative SW bias with the largest amplitude attributed to the Morrison scheme.

For the western side of the Peninsula (Rothera station), SW biases are always positive, and LW biases always negative, whatever the cloud scheme or the year considered (left part of Table 4). All simulations consistently show this imbalance suggesting no improvement in cloud simulation. Besides, no statistically significant difference is observed between WSM5,
and the other schemes. Note that almost no cloud liquid water (not shown) is simulated above Rothera (as opposed to AWS14, Figure 2), whatever the cloud scheme used, in line with the persistent large SW and LW biases. Ice and snow (graupel),

however, are formed in similar amounts to the ones shown in Figure 3 (not shown). Overall, Table 4 demonstrates the high sensitivity of the simulated downward radiation fluxes to the microphysics scheme used in Polar WRF.

A major issue in assessing the performances of the cloud microphysics schemes by investigating radiation biases is that it does suppose that the appropriate information is passed on from the cloud scheme to the radiative scheme. This aspect can explain the apparent paradox of the significant improvement of the LW bias to the east of the Peninsula with three schemes, while no concomitant SW bias improvement is being observed. The Discrepancies in SW and LW biases improvements will be further discussed in section 5.1. Radiative schemes themselves also require careful examination as they also rely on various assumptions, and simplified geometry to retrieve SW and LW fluxes. The radiative schemes that we used was chosen for consistency with K15, in order to compare their conclusions (using AMPS) to ours (using Polar WRF). We do not intend here to investigate the radiative schemes implemented in WRF. For further assessment of cloud microphysics schemes performances and behaviours at a much wider scale, we now compare the simulations outputs to each other and to the cloud microphysics properties as measured during the BAS aircraft campaigns that took place over the Antarctic Peninsula (presented in Part 1).

## 4 Results: simulated clouds as compared to observations

### 4.1 General trends for simulated clouds across the Peninsula

The topography of the Antarctic Peninsula (Figure 1) makes it interesting to focus on zonal distribution of latitudinal averages for the Liquid Water Content (LWC, in g kg$^{-1}$), and the Solid Water Content (SWC, g kg$^{-1}$). SWC comprises ice, snow and graupel mass. It is different from the Ice Water Content (IWC), which consists only in the mass of the cloud ice crystals. LWC and SWC were respectively averaged between latitudes 65.5°S and 68.5°S and altitudes below 4500 m. This geographical area includes the region where both flight campaigns took place in summer 2010, and 2011 (Figure 1b). For simplicity we designate each simulation by using the name of its cloud microphysics scheme.

Both periods of interest display the same relative trends, and we present an average over both periods to give an overview. Averages are computed considering either all values, including null instances (LWC$_0$ and SWC $_0$ in Figure 4a and Figure 4b, respectively) or only strictly positive values (LWC and SWC in Figure 4c and Figure 4d, respectively). Thus, we always have LWC$_0 \leq$LWC, and SWC$_0 \leq$SWC. LWC (resp. SWC) gives the liquid (resp. ice) content that is simulated disregarding how often the clouds form. LWC$_0$ (resp. SWC$_0$) describes a more realistic average behaviour since it also accounts for the ability of the scheme to lead to liquid (resp. ice) cloud formation, more or less often.

For all the simulations LWC, and LWC$_0$ are in the interval 0.05-0.14 g kg$^{-1}$, and 0.002-0.03 g kg$^{-1}$, respectively. SWC, and SWC$_0$ are in the interval 0.02-0.08 g kg$^{-1}$, and 0.01-0.035 g kg$^{-1}$, respectively. The lower limit of LWC$_0$ (0.002 g kg$^{-1}$) is due to WSM5 decreasing over the mountains down to 0.002 g kg$^{-1}$ around 65°W. For the other cloud schemes, LWC$_0 \geq 0.01$ g kg$^{-1}$. There is roughly a factor of 5 to 10 between LWC$_0$ and LWC while there is a factor of 1.2 to 2 between SWC$_0$ and SWC. The liquid phase features more important changes (from null to non-null values) than the total ice phase, which is simulated more frequently.

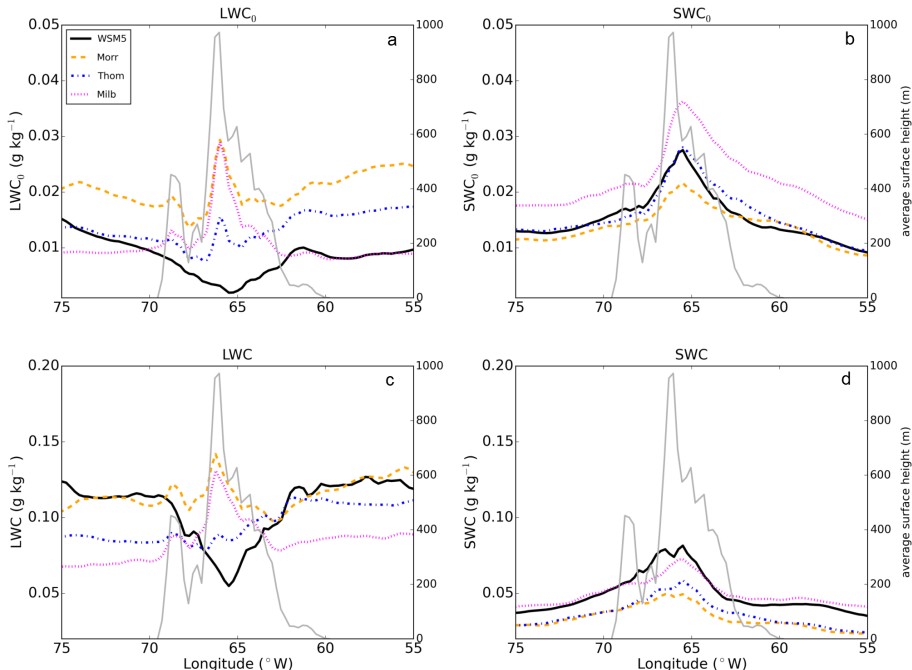

**Figure 4.** Longitudinal distribution of latitudinally (65.5-68.5°S) averaged LWC and SWC (g kg$^{-1}$) over both periods of interest for schemes WSM5, Morrison, Thompson, and Milbrandt. The average is computed over all gridboxes and times leading to (a) LWC$_0$ and (b) SWC$_0$, or only over gridboxes and times where values are non-null leading to LWC (c) and SWC (d). The grey thick line shows the surface height averaged over the same region and labelled on the right vertical axis of each plot. Note the identical scales used for the vertical axes for LWC$_0$ and SWC$_0$, and LWC and SWC, respectively.

WSM5 strikingly differs from the Morrison, Thompson, and Milbrandt schemes in that its LWC and LWC$_0$ decrease above the Peninsula mountains. LWC drops from ∼0.12 g kg$^{-1}$ by more than 50% from 70°W to 65°W, before increasing back from 65°W to 60°W to ∼0.12 g kg$^{-1}$ (Figure 4c). Except East of 62°W where WDM6's LWC is larger than WSM5 by less than 0.03 g kg$^{-1}$ (not shown), both schemes display very similar averages for LWC and SWC, and we only show WSM5. LWC is 5 much steadier for the three other schemes, and a sharp increase for the Milbrandt and Morrison scheme is observed above the highest terrains, caused by the orographic forcing induced by the westerlies or the easterlies (see section 4.2).

The first obvious assessment with respect to the ability of the cloud schemes in forming (supercooled) liquid clouds is that WSM5 (WDM6) leads to less supercooled liquid mass compared to the Morrison, Thompson, and Milbrandt schemes across the Antarctic Peninsula. Eastward of 62°W, WSM5's LWC$_0$ is 100% (50%) smaller than Morrison's (Thompson's), while it 10 is similar to Milbrandt's (Figure 4a). However, in the central region over the mountains, WSM5 (and WDM6) leads to less liquid mass by up to an order of magnitude than the three other schemes. Westward of 62°W LWC$_0$ is similar for WSM5, Milbrandt, and Thompson; they are all twice as small as Morrison's LWC$_0$. WSM5 does not lead as often as the other schemes to supercooled liquid formation, what is illustrated by its lowest LWC$_0$ values, yet it does simulate as large average liquid water

contents as the other schemes when and where liquid forms (similar LWC), except in the central region where orographically induced clouds have systematically less liquid water. The ice phase instead shows a similar behaviour across the different cloud schemes with an increasing SWC closer to the high altitude topography, due to orographic forcing. $SWC_0$ is similar for WSM5, Morrison and Thompson while with Milbrandt it reaches $50\%$ larger value largely due to the graupel mass (not shown).

Comparing $LWC_0$ and $SWC_0$, we see that the simulation with the Morrison scheme is the only one sustaining supercooled liquid mass more frequently than ice mass ($LWC_0 > SWC_0$ by a factor of >1 to 2). For the Thompson scheme, $LWC_0 \sim SWC_0$ on average (but $LWC_0 < SWC_0$ over the mountains, and $LWC_0 > SWC_0$ East of the Larsen C). The Milbrandt scheme leads more often to ice mass formation than liquid mass ($LWC_0 < SWC_0$) by a factor of less than 2 at all longitudes, and WSM5 by a factor comprised between 1 and 5. Finally, the simulation with WSM5 (WDM6) is the only one resulting in an anticorrelation

between LWC ($LWC_0$) and SWC ($SWC_0$) with an increase (resp. decrease) for the cloud ice (resp. cloud liquid) over the mountains.

## 4.2    Dynamics, and microphysics structure of the simulated clouds

The Antarctic Peninsula mountains acts as a barrier to the westerly, or easterly winds that drive the formation of orographic clouds. As a complement to the general picture given above, we identified two periods of sustained westerly, and easterly

wind regime, respectively. We isolated the period 7 January to 10 Januray 2011 when westerlies prevailed almost exclusively, and similarly the period 11 January 2011 to 17 Januray 2011 when the easterly regime prevailed. Note that the average wind directions and speed, and their relative variations are in agreement with upper-air measurements performed daily from Rothera station (not shown), if not always quantitatively at least qualitatively, as well as with measurements from the aircraft (not shown).

Figure 5 shows time- and space-averaged transects of the hydrometeors' mass (including null instances) across the Peninsula on a $\sim 100$ km wide (67-68 °S) latitudinal band approximately centered on Rothera station, for WSM5 (Figure 5a, and Figure 5b), and the Morrison scheme (Figure 5c, and Figure 5d). The westerly cases (b + d), and easterly cases (a + c) show the orographic clouds microphysical structure. It also illustrates the very different contexts of Rothera station on the western side, and of AWS14 and AWS 15 on the eastern side. Rothera itself is in the lee of a mountaneous feature (Adelaide island) and the

topography adds to the complexity in simulating the clouds compared to the flat Larsen C ice shelf.

WSM5 predicts completely glaciated clouds on the Peninsula, and liquid clouds only away from the mountains with a very limited vertical extent (up to 500 meters above surface). The Morrison scheme maintains mixed-phase clouds across the region and at much higher altitudes. This fact alone is in better agreement with observations from the aircraft, which measured almost exclusively mixed-phase clouds (Part 1). Over the Ice shelf the snow increases from 0.01 to 0.05 g kg$^{-1}$ as we get closer to the

mountain barrier for all the schemes (similar amounts are simulated on the western side during the westerly regime). However, WSM5 simulates an IWC (orange lines) as large as the snow particles mass (red lines) down to the surface, contrasting very much with the Morrison scheme.

Note that WDM6 (not shown) gives similar results as WSM5. Also, the microphysical structure of the clouds predicted by the Thompson scheme, and the Milbrandt scheme (not shown) are similar to Morrison's. On either side of the Peninsula,

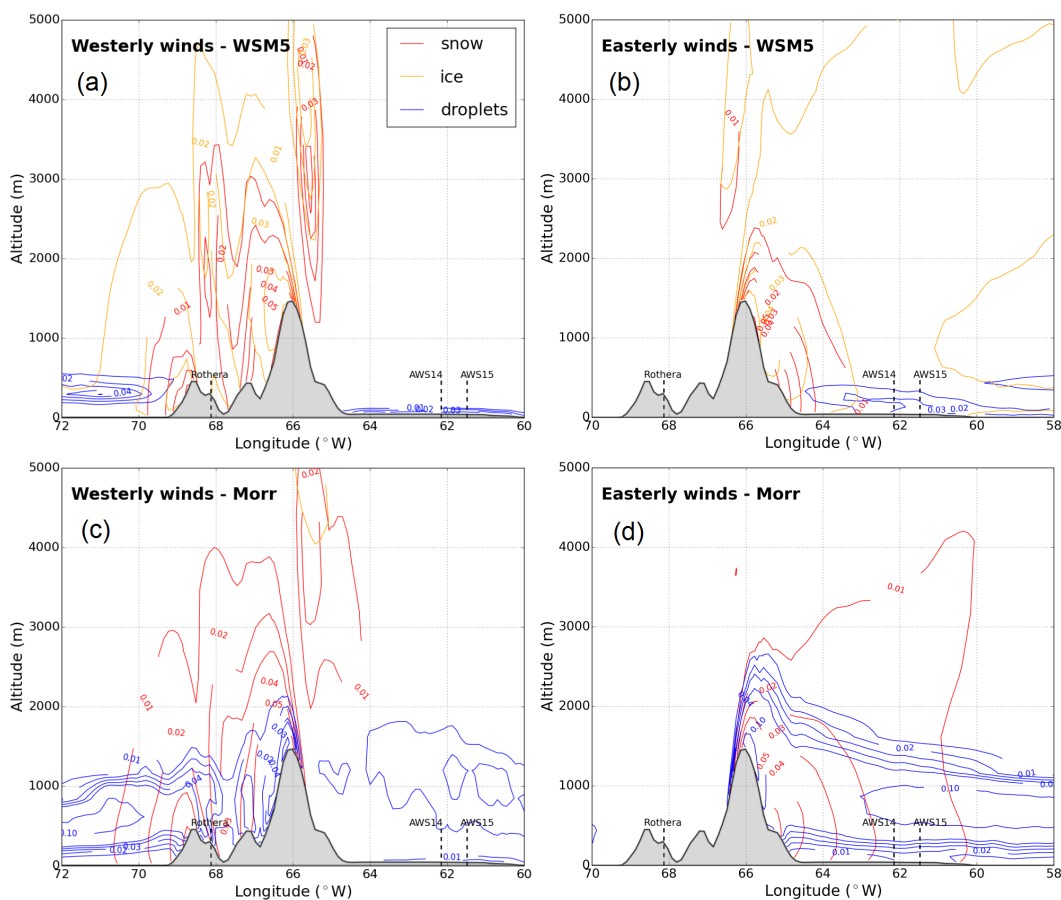

**Figure 5.** Transect of the cloud microphysics for WSM5 (a and b), and Morrison (c and d) averaged over a period (7-10 Jan 2011) dominated by westerly winds (a and c), and over another period (11-17 Jan 2011) dominated by easterly winds (b and d). The transect is approximately centered on Rothera Station (67.586°S), and it is an average over a 100 km wide latitudinal band. The longitudes of Rothera, AWS14, and AWS15 are indicated.

downwind, the Morrison schemes forms the most abundant mixed-phase cloud layer with LWC∼0.1 g/kg, and the clouds extends almost down to the surface (LWC∼0.01 g kg$^{-1}$), whereas the Milbrant, and the Thompson schemes form less than half of that maximum amount, in line with the general picture given in section 4.1. Also, The Milbrandt scheme forms a significant amount of cloud ice crystals (IWC) above 3000 meters, as well as graupel in the mixed-phase orographic clouds above the windward slopes (not shown), which are absent from the average transects of the Morrison, and the Thompson simulations.

### 4.3 Microphysics schemes performances West and East of the Antarctic Peninsula

#### 4.3.1 Liquid phase

To assess the performances of the different cloud schemes we compare the LWC measured from the aircraft to the simulated LWC by restraining the latter to the model gridboxes corresponding to the flight tracks. We only consider non-null LWC values (LWC>0.001 g kg$^{-1}$). For each data point, the closest (both in time and space) gridbox value is extracted from the model. Latitudinal averages are derived for each flight per $0.5°$ longitude bins, for simulations and observations. At the latitude of Rothera station ($67.586°$S) this corresponds to $\sim$10 kilometres (ie two gridboxes). Then, global west and east averages are derived, corresponding to longitudes westward of $67°$W, and to longitudes eastward of $65°$W, respectively (as in Part 1). LWC is derived as presented in Part 1 using the droplet size distribution obtained from the Cloud and Aerosol Spectrometer (CAS). The unknown thermodynamic phase of the smallest particles seen by the Cloud Imaging Probe (CIP, see Part 1), but not resolved, and that could either by drops or small crystals, may induce a bias in the derivation of LWC. However, if all of them were counted as droplets, they would increase LWC by $\leq 8\%$ for all flights, except two flights in 2010 (13%, and 30%) and one flight in 2011 (12%). This bias does not alter the results, and conclusions below. More information on the instruments and the measurement can be found in Part 1 (their section 2.1).

Figure 6 shows the scatter plots of simulated LWC versus observed LWC for 2010 (Figure 6a, and Figure 6b) and 2011 (Figure 6c, and Figure 6d), and for either side of the Peninsula, West (a, and c), and East (b, and d). Regional (East or West) averages are represented by the largest bold markers while smaller markers relate to individual flight averages. Note that the width of the large markers is larger than the length of the errorbar associated with the aforementioned error (bias) related to the LWC derivation. The numbers shown next to each scheme's markers in the legend (in the form $n_5$;$n_{50}$/N) indicate the number of flight tracks for which the simulation forms at least an average of 5% ($n_5$) or 50% ($n_{50}$) of the observed average LWC, over the total number of flight tracks (N) having measured cloud liquid. We refer to those as the $n_5$ criterion, and the $n_{50}$ criterion, respectively.

Three results stand out. First, all the schemes perform worst on the West compared to the East in terms of number of tracks with simulated clouds ($n_5$, and $n_{50}$ criteria), except for the WSM5 scheme which performs equally bad on both sides. Second, the WSM5 scheme has the lowest numbers of flights with some liquid clouds simulated (n5 criterion). For the Morrison, Thompson, and Milbrand schemes, about (Figure 6a) or much less (Figure 6c) than 30% of flights are predicted with some substantial supercooled liquid ($n_{50}$ criterion) in the West, and more than 60% of them in the East (Figure 6b and d). Third, the Morrison scheme performs on average the best in reproducing observed LWC on the western and the eastern portions of the flight tracks, with larger values of LWC simulated compared to the other schemes. When considering the $n_5$ criterion, the Thompson, and Milbrandt schemes show equally good scores compared to Morrison suggesting the same ability to initiate a non-negligible supercooled liquid phase, as opposed to WSM5 (and especially on the eastern side). However, overall the Morrison scheme performs better because it has an averaged simulated LWC closer to the observed one within a factor of less than 2 (except in East 2010 – Figure 6b – where it simulates an average LWC three times larger than the observations).

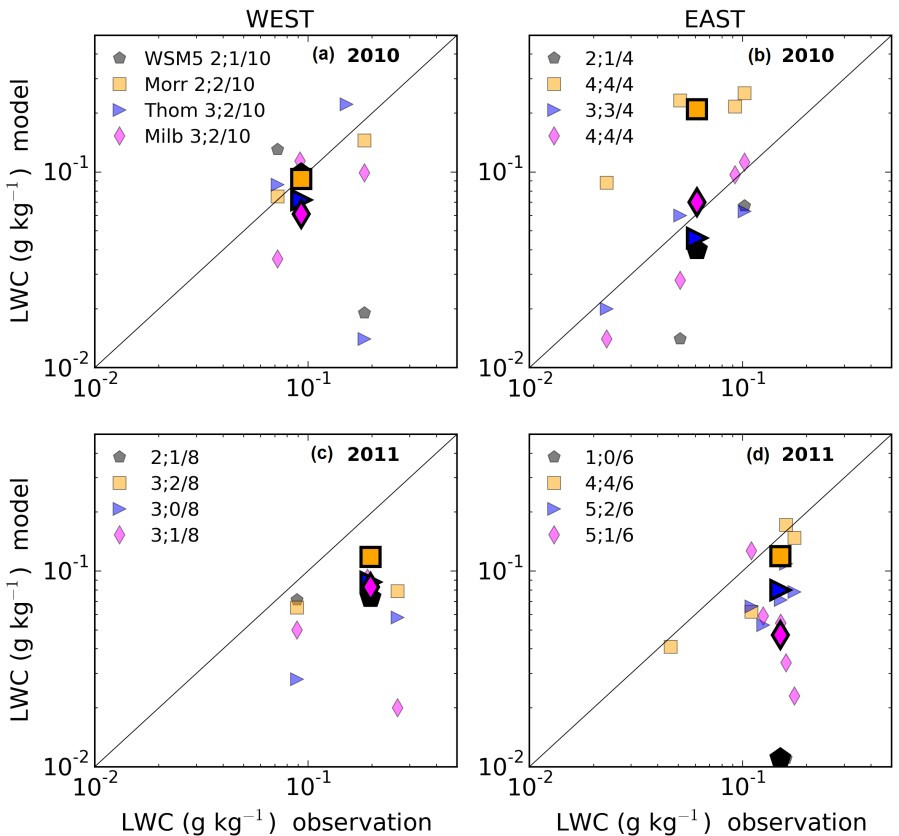

**Figure 6.** Scatter plot of simulated LWC versus observed LWC in 2010 (a and b) and in 2011 (c and d) on the western side of the Peninsula (a and c) and on the Eastern side of the Peninsula (b and d). Light markers show averages per flight track while bold markers give the average of all the tracks on each side of the Peninsula, respectively. The numbers next to each scheme's marker in the legend ($n_5$;$n_{50}$/N) gives the number $n_5$ (resp. $n_{50}$) of simulated flights with a simulated LWC at least 5% (resp. 50%) of the observed LWC, to the total number N of flights measuring an average LWC. Note that in (c) the bold markers (total average) overlay some light markers (flight averages), what explains the actual higher position of the total average on the graph compared to the other discernible lower flight averages.

Those averages do not take into account the duration over which such values are observed. Thus, we use an additional metric that is the average time spent in cloud (or instances of cloud occurrences) on either side of the Peninsula. The average ratio of the time spent in clouds in the model (with LWC>0.01 g kg$^{-1}$) over the one in the measurements, is given in Table 5 for each side, and year. The average is derived as an average of the flight-averages. Over both periods the best scheme appears as the Morrison since the Thompson, and the Milbrandt schemes have very low occurrences of clouds compared to the observation on the western side in 2011 with 4% in 2011 and 5%, respectively. On the eastern side, WSM5 has the poorest performance (<1%), and the Morrison scheme has twice as much occurrences of clouds (although still quite low) in 2011 compared to the two other schemes, and it overpredicts the formation of clouds in 2011 (215%), although not the average LWC (Figure 6d).

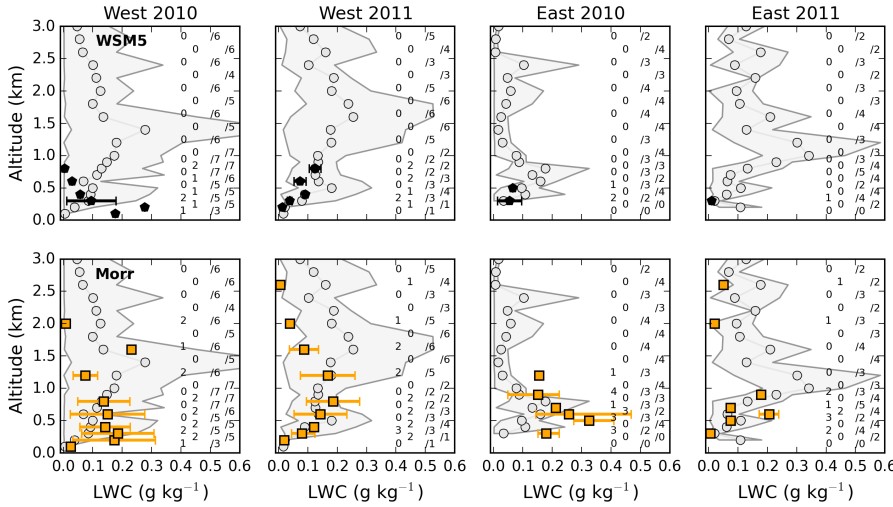

**Figure 7.** Averaged vertical profiles of non-null LWC for WSM5 (top), and Morrison (bottom) and for the observations, west (left) and east (right) of the Peninsula. Grey markers indicate the measured average at each altitude, while the shaded area gives the range of the observed flight averages at each altitude. Similarly, coloured markers and error bars relate to the cloud schemes. The numbers indicate how many simulated flight averages were used to derive the global average at each altitude for each scheme, as compared to ("/") for the observations.

Average vertical profiles of cloud liquid (and ice) were also derived for flights measurements as well as for the model outputs. The altitude grid on which flights observations, and model outputs, were averaged on is finer in its lower layers with one level every 100 meters below 1100 meters, and every 200 meters above 1100 meters. At each altitude levels the average of the flight averages is computed so that every flight has the same weight. Model altitude levels are separated by less than 1000 meters at the highest altitude levels of the atmospheric column. However, below 4500 meters, where the flights took place, the maximum model level separation is approximately 500 meters. Thus, any data point level is less than 250 meters away from the closest model level (less than 100 meters below 1100 m).

Figure 7 compares vertical distribution of observed (grey circles) and simulated (coloured markers) non-null average LWC (>0.001 g kg$^{-1}$), for WSM5 (top), and the Morrison scheme (bottom). The grey shaded area shows the spread of all flight averages. The errorbars show the spread of the simulated flight averages. The numbers at each level indicate how many simulated flights with non-null averages are used to derive the total average of each level, for the simulations as compared to ("/") the observations.

WSM5 scheme does not form liquid clouds above 800 meters on the western side of the Peninsula, as well as no liquid clouds above 500 meters on the eastern side, during both periods of interest. Liquid clouds were observed as high as 4400 meters. The numbers at each level show that WSM5 simulates fewer occurrences of liquid compared to the Morrison scheme, which still underpredicts the occurrences of liquid clouds. The Morrison scheme shows liquid cloud formation up to 2500 meters although only, very few instances above 1500 m.

WDM6 shows no improvement compared to WSM5 (not shown). The Milbrandt, and the Thompson schemes simulate liquid clouds more often than WSM5 in the lowest layers, but no clear systematic difference emerges between those two and the Morrison scheme. The Morrison scheme simulates best the increasing trend of LWC with altitude in the West in 2011. It has the largest LWC below 1000 meters (by 0.1-0.2 g kg$^{-1}$) on either side of the Peninsula in 2010 compared to other schemes, while LWC is comparable for all the three schemes in 2011 (not shown).

### 4.3.2 Ice phase, and mixed phase

For completeness we compare the simulated Solid Water Content (SWC, g kg$^{-1}$) to the observed ice mass. Figure 9 is the same as Figure 6 but for SWC, with the addition of the corresponding Ice Water Content (IWC) regional averages shown as grey bold markers. (The latter are slightly shifted rightwards by 50% of the observed value on the x-axis, in order to be visible). The smaller and lighter markers are individual flight averages. The same n$_5$ criterion, and n$_{50}$ criterion as in Figure 6 are used and referenced on the Figure 9 next to each cloud scheme's name.

As mentioned in Part 1, there is an uncertainty in the smallest particles detected by the Cloud Imaging Probe (CIP), however they contribute to a negligible amount of the total measured ice mass. To the other end of the size distribution, the maximum cut-off for detected ice particles is about 1.5 mm in size (diameter). Thus possible larger particles that could significantly add to the mass are not detected. However, in order to have an estimate of the error caused by the missed larger particles we approximated and extrapolated the average size distribution of the crystals for each flight (examples are shown in Figure 8a, and 8b), using an exponential distribution of the form $N(D) = N_0 \exp(-\lambda D)$ (known as Marshall-Palmer distributions) commonly used for the rain and the ice hydrometeors in the cloud microphysics schemes (e.g Morrison et al., 2009). Using the exponential distribution and the mass-diameter law, we derive an ice water content below 1.5 mm, and above 1.5 mm, respectively. In order to derive the mass for $D \leq 1.5$ mm we integrated over the crystal sizes starting from the peak diameter of the distribution, and up to 1.5 mm. The peak diameter of the observed ice crystal distribution is located in the range 250-425 $\mu$m (with an average of 315 $\mu$m), value from which the exponential distribution can approximate the distribution of the largest crystals. Then, the ice water content for particles with $D > 1.5$ mm, and up to an arbitrary limit of 3.2 mm, was derived (setting the upper limit to even larger sizes does not change the resulting additional ice mass given the even lower amounts of crystal number concentration predicted by the exponential distribution) The ratio of both isce water contents allow to estimate the relative error caused by the undetected particles on the measured SWC when assuming an exponential distribution. For the 2010 flights, this average error is about 5% , including an outlier flight with a 33% error (ignoring this flight brings the average error to 2%). For the 2011 flights, the average relative error is about 8%, including an outlier with a 65% error (shown in Figure 8c) (ignoring this flight brings the average error to 2%). The large error derived for the two flights is related to a shoulder of the crystal distribution for the larger particles, leading to an exponential distribution predicting a number concentration of the largest particles likely to be in large excess compared to the actual one (Figure 8c). Overall, these estimates of the relative errors on SWC do not alter the main conclusions presented here.

Table 3 gives the cut-off radii between ice particles and snow particles in the different cloud schemes. The different definitions of the icy hydrometeors across the cloud schemes add to the difficulty of performing comparisons between the schemes

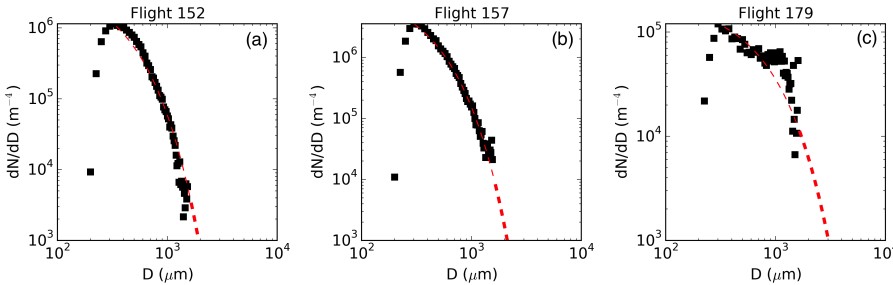

**Figure 8.** Average Size distribution of the crystals identified with the CIP for the flights (a) 152, (b) 157, and (c) 179 (black squares), along with the exponential distribution approximating them (red dashed line). The relative increase in ice mass when further integrating from 1.5 mm to larger diameters (equal to the relative error on the actual ice mass used in this study) is about 3% (a), 2.5% (b), and 65% (c).

as well as to the observations. The observed particles identified unambiguously as crystals in Part 1 span the diameter range 200 $\mu$m to 1.5 mm. Hence, because the cloud microphysics scheme have a lower limit size smaller than 200 $\mu$m for the ice crystal and an upper limit size larger than 1.5 mm for the precipitating ice particles (snow, graupel)(see Table 3), we expect the simulated IWC, and SWC to bracket the observations. However, the measured ice mass should be closer to SWC than to IWC

given the relatively low additional mass expected from particles with D>1.5 mm using the estimates presented above.

In 2010, the instances where SWC and IWC do bracket the observations happen on both sides of the Peninsula (Figure 9a, and Figure 9b) for the Morrison, and the Thompson schemes (note that the Thompson's IWC is comprised between $10^{-4}$ and $10^{-3}$ g kg$^{-1}$). Both WSM5's SWC and IWC equal the observation showing that a significant part of the simulated SWC is on average in the form of cloud ice crystals (IWC) (i.e. radii <250 $\mu$m, see Table 3). In 2010, west of the Peninsula, Milbrandt's

SWC and IWC are lower than the observations suggesting not enough ice formation.

In 2011 (Figure 9c, and Figure 9d), all the scheme have both averaged SWC and IWC lower than the observations, except for WSM5 to the East of the Peninsula, where the averaged IWC exceeds the observed value. East of the Peninsula, all the schemes predict equally well some non-negligible ice phase ($n_5$ criterion) with Morrison, Thompson, and Milbrandt performing better than WSM5 when considering the $n_{50}$ criterion. However, the schemes perform worst west of the Peninsula with less than 40%

of ice occurrences actually simulated ($n_5$ criterion). Overall, As for the liquid phase, the occurrences of the ice phase are less well simulated on the western side of the Peninsula, as compared to its eastern side.

Finally, we focus on the partition of water between the condensed phases, LWC, and SWC, by looking at the total average mixed-phase ratio $f_m$=LWC/(LWC+SWC) as a function of temperature along the flight tracks. Table 6 summarizes the statistics on $f_m$ derived from measurements, and from simulations. First, none of the schemes sustain liquid clouds at temperatures

below $-15°$C, or even below$-9°$C for the WSM5 (WDM6) scheme (leading to $f_m = 0$). This will be further commented in section 5.2. Second, between $-15°$C and $0°$C, the Morrison scheme (0.91$\pm$0.1), and the Milbrandt (0.78$\pm$0.1) schemes have an average $f_m$ in closest agreement with observations (0.83$\pm$0.08). WSM5 performs the least well with $f_m$ around 0.6 on average and down to 0.07 at its minimum. WSM5 ($\sigma$=0.24), and the Thompson scheme ($\sigma$=0.2) have a variability of $f_m$ more than twice larger than the observations ($\sigma$=0.08). Practically, for WSM5 and the Thompson scheme, it results in a highly

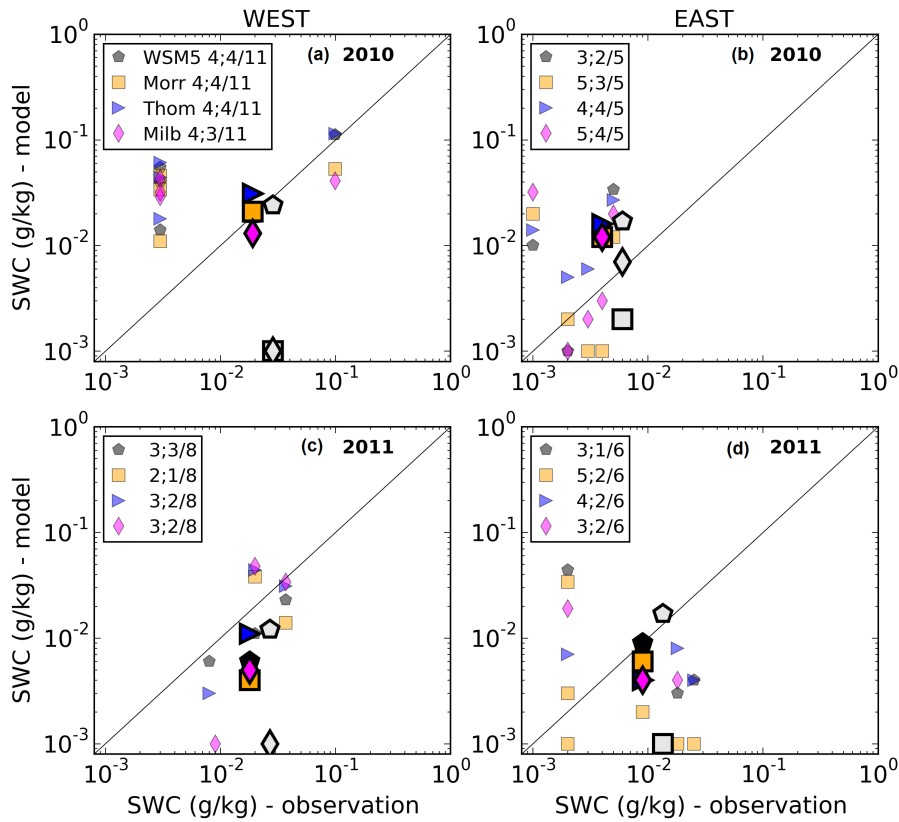

**Figure 9.** Scatter plot of simulated Solid Water Content, SWC (ice, snow, graupel) versus observed SWC in 2010 (a and b) and in 2011 (c and d) on the Western side of the Peninsula (a and c) and on the Eastern side of the Peninsula (b and d). Light markers show average per flight while bold markers gives the average over the whole tracks on each side of the Peninsula, respectively. Each bold grey marker indicates the Ice Water Content (IWC) average corresponding to the SWC average and each IWC marker is slightly shifted on the x-axis by 50% of the observed value for readability. The numbers next to each scheme's name ($n_5$;$n_{50}$/N) gives the number $n_5$ (resp. $n_{50}$) of simulated flights with SWC at least 5% (resp. 50%) of the observed SWC, to the total number N of flights measuring an average SWC of at least 0.0001 g kg$^{-1}$.

variable mixed-phase ratio from one 0.5° temperature bin to the next, what is not observed in the measurements (not shown). The Morrison scheme ($\sigma$=0.1), and the Milbrandt schemes ($\sigma$=0.1) have a steadier $f_m$ across the investigated temperature range where mixed-phase clouds are simulated, in closer agreement to the observations.

### 4.4 Temperatures and Water vapour in Polar WRF over the flight campaigns

5   We take advantage of temperature and water vapour measurements performed along with the cloud in-situ measurements to compare with the averaged simulation outputs. Latitudinal averages (in 0.5° longitude bins) for both observations and simulations, are shown for temperatures (°C) and water vapour mass mixing ratios (g kg$^{-1}$) in Figure 10a, and Figure 10b

respectively. The variability of the water vapour, and of the temperature (shown as the standard deviation of the flight averages in each longitude bin) is indicated with shaded area for the observations, and with errorbars for the different cloud schemes. The measurements uncertainty for the temperature measured with a Rosemount probe is about $0.3°C$ (Stickney et al., 1994), corresponding to less than the width of the solid blue and red lines, respectively. The darkest narrow shaded areas bracketing

solid lines on both years correspond to a conservative estimate of uncertainty on water vapour ($\pm 0.15$ g kg$^{-1}$) as derived using the relative humidity measured with a Vaisala Humicap HMP45 ($\pm 3\%$ estimated relative error), and the atmospheric temperature measurements from the Rosemount probe. A Buck 1011C cooled mirror hygrometer also present on board was used to correct for an offset in the Humicap measurements. At low temperatures and humidity the cooled mirror hygrometer occasionally has difficulty in identifying the frost point correctly and tends to hunt over a wide range. Therefore the Humicap

measurements were used once corrected using the cooled mirror hygrometer during periods when we are confident that it has correctly identified the frost point.

For the temperature, in 2010 all the simulations show best agreement with the measurements to the east of the Peninsula where the overestimation of the temperature ranges between 0 and $1°C$ (Figure 10a, top). Westward of $65°W$ the positive biases are larger and range between 1 and $2°C$. In 2011 and East of the Peninsula, the temperature bias lies between 1 and $2°C$,

whereas West of $69°W$ it ranges between 2 and $3°C$ with the exception of the Thompson scheme leading to overestimations as large as $4°C$ (Figure 10a, bottom).

For the water vapour, the 2011 observed average is underestimated at almost all longitudes except between $68.5°W$ and $64°W$ where it is overestimated by 0.15 g kg$^{-1}$ on average (Figure 10b, top). Eastward of $63°W$, the underestimation increases up to values closer to 1 g kg$^{-1}$, while westward of $71°W$ it remains around 0.5 g kg$^{-1}$. In 2010 the average water vapour

is underestimated by 0.2-0.5 g kg$^{-1}$ , except west of $68°W$ where it reaches 1 g kg$^{-1}$ (Figure 10b, bottom). The bias then decreases to around 0.25 g kg$^{-1}$ in the area $67.5°W$-$63.5°W$, except for WSM5, which remains closer to 0.5 g kg$^{-1}$. Eastward of $62°W$ the underestimation increases up to 1g kg$^{-1}$ but only one flight measured water vapour, hence the poor statistics (as shown by the absence of shaded area). WSM5 has the largest biases in averaged water vapour during both years, 0.6 g kg$^{-1}$ and 0.45 g kg$^{-1}$ in 2010 and 2011 respectively, mostly consisting in an underestimation of the observed water vapour. Other

schemes also mostly underestimate the water vapour, however less than WSM5 by 0.05-0.1 g kg$^{-1}$. No cloud scheme clearly stands out in terms of reducing the negative bias.

Overall across the Peninsula the simulations underestimate the measured water vapour by an average value of 0.5 g kg$^{-1}$ ($\pm 0.2$ g kg$^{-1}$, depending on schemes or regions across the Peninsula), and the temperatures are overestimated by $1°C$ ($\pm 0.2°C$ depending on the scheme) in 2010 and 2 ($\pm 0.5°C$ depending on the scheme) in 2011. Interestingly, the variabilities of the

observations (shaded area), and of the simulations (error bars), are consistent with each other. This suggests a good performance of the model average, and variability. The broad agreement in temperature and water vapour between the simulations suggest that their differences in average simulated clouds cannot be mainly related to the differences in water vapour and temperature, but rather to their microphysics. The biases compared to the observation will be further commented in the section 5.4.

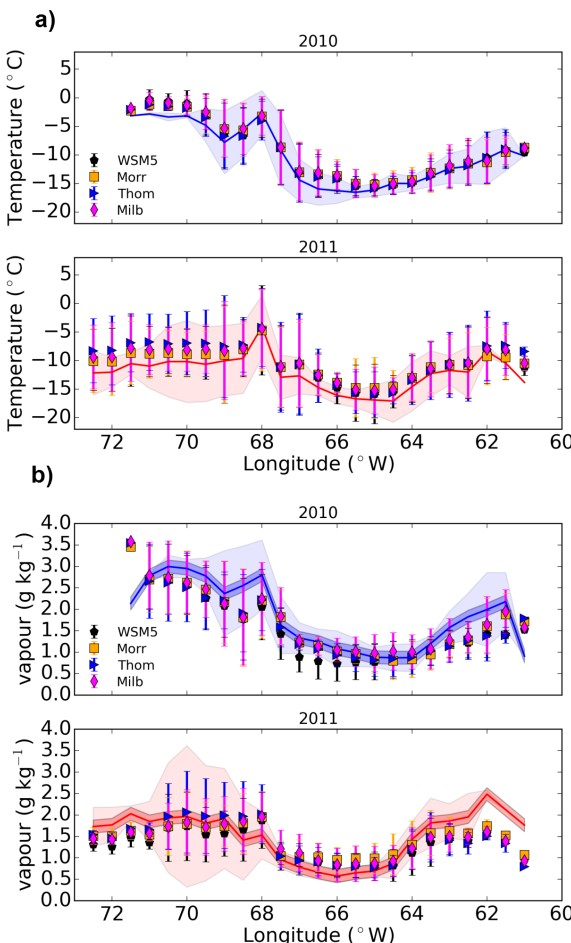

**Figure 10.** (a) Zonal distribution for 2010 and 2011 flight campaigns of (a) averaged temperatures and (b) water vapour (g kg$^{-1}$). Measurements are shown as a solid line, and simulations as markers. Both shaded areas, and error bars, give the standard deviation in each 0.5° longitude bin for the observation, and the simulations, respectively. The dark shaded area correspond to a conservative estimate of the uncertainty on water vapour (see text for details).

## 5 Discussion

### 5.1 On the radiative biases

A deficiency of downward LW radiation responsible for a cold summer surface (temperature) bias in Polar WRF simulations was spotted at a continental scale by Bromwich et al. (2013a) at 60 km, and 15 km resolution, and the authors related this bias to a deficiency in the cloud cover. Bromwich et al. (2013a) showed that using ERA-interim analysis forcing at the domain boundaries (instead of GFS analysis) helped to significantly reduce the average cold summer bias (see their Table 5), although the improvement for surface pressure or dew point is not clear. Bromwich et al. (2013a)'s simulation outputs as well as K15's

AMPS results relied on the WSM5 scheme. We did find a similar negative bias for the LW radiation to K15 over the Larsen C Ice Shelf, as well as a similar positive bias for the SW radiation (section 3.1).

When schemes different from WSM5 or WDM6 are used with Polar WRF in our simulations, a strong decrease in the LW bias for both periods of interest over the Larsen C ice shelf are measured. It suggests that schemes like the Morrison, Thompson or Milbrandt schemes should be preferred to the WSM5, and WDM6 schemes in studies dealing with the evolution of the energy budget of the Larsen C Ice Shelf within Polar WRF. The strong decrease in LW surface biases (by as much as 20 W/m$^2$, see able 4) when using the three cloud schemes, which have a more sophisticated ice microphysics parameterisation (double-moment) is systematically statistically significant on both years and at both AWS (AWS14 and AWS15). Note that the smallest biases are obtained using the Morrison scheme.

The explanation as to why the LW bias is significantly improved while the SW is not always (especially in 2010) is most probably because the variations of the cloud droplets effective radius is not accounted for in the model. The radiative scheme (Goddard scheme, see Table 1) parameterises the optical depth for water and ice as a function of the particle effective radius (Chou and Suarez, 1999). This parameterization does not assume any type of droplet (crystal) size distribution, so it can be used with the different cloud schemes despite their own different assumptions on the hydrometeors size distributions. Also, the SW radiative scheme assumes a constant value of 10 $\mu$m for the cloud droplets effective radius. However, for a given LWC, the SW radiation is scattered in different ways depending on the effective radius of the droplets, with smaller radii reflecting more efficiently SW radiation. The droplet effective radius derived for both campaigns is about 7 $\mu$m, close to the 10 $\mu$m assumed in the radiative scheme. Running a simulation over a shorter period (11-20th January 2011) replacing the constant effective radius of 10 $\mu$m by a constant 7 $\mu$m in the radiative scheme did not lower the SW radiative bias over the Larsen C Ice shelf (not shown). This was expected as it is rather the variations of the effective radius with time that could be expected to improve the SW bias. As noted by Bromwich et al. (2013a), the SW bias is of secondary importance for the surface energy budget because SW radiations not reflected by missing clouds in the model will be reflected by the icy or snowy surface underneath. The cloud radiative effect dominates in the LW radiation over icy surfaces (as opposed to over the ocean).

The poorer performances of the various simulation in terms of surface radiation biases at Rothera station (Table 4, left part) and especially the similarly large LW surface biases for all the schemes are consistent with a poorer representation of the supercooled liquid clouds in the western part of the Peninsula (Figure 6). Indeed, only a few flight tracks were simulated with supercooled liquid phase (3 out of 11 and 3 out of 10 at best, in 2010 and 2011 respectively). This is further commented in section 5.2, where we discuss the simulation of the cloud phase.

## 5.2 Simulating the clouds thermodynamic phase

Cloud schemes form supercooled liquid provided the growth of the activated ice phase does not consume the entire excess of water vapour (compared to RH=100%). Except for the Milbrandt scheme which completely removes the supersaturation by conversion of the excess of water vapour into liquid, the other schemes explicitly derive a condensation growth rate. Thus, the cloud microphysics schemes mainly differ in their ice microphysics, and mixed-phase interactions, which will determine their ability to form and maintain supercooled liquid in the atmosphere.

WSM5 (WDM6) is the only microphysics scheme showing an anticorrelation of the liquid water content and solid water content on the Peninsula, suggesting a systematic depletion of water vapour in favor of the ice phase (Figure 4). Close to, and above the topography WSM5 has a deficit in liquid clouds due to orographic forcing which favours ice clouds, whereas the Morrison, Milbrandt, and Thompson schemes have a steady if not increasing LWC. One of the main differences between WSM5 and WDM6 (hereafter called the WRF schemes), and the other three schemes is that the former are single-moment schemes for the icy hydrometeors, whereas the latter are double-moment schemes for the ice crystals (only the Thompson scheme is not a double-moment scheme for snow/graupel particles). A consequence of The WRF schemes being single-moment schemes for the ice crystals is the use of a relationship for linking the number concentration of ice crystals ($N_i$) to the ice water content ($q_i$), since they cannot evolve independently. $N_i$ is diagnosed from $q_i$ based on empirical relationships (see equations 5a-d Hong et al., 2004, where $N_i$ is proportional to $q_i^{3/4}$). In addition to that, the INP parameterisation used in the WRF schemes (WSM5 and WDM6) is predicting significantly more INPs than any other parameterisation above -15°C (as it will be shown in the next section 5.3). Since it is used to predict the initial $q_i$ when firt ice appears, the latter is biased towards larger than expected values, and so $N_i$ will also biased towards larger values (because of the empirical relationship linking both). This, in turn, impacts the growth rate of the ice crystal mass, which depends on $N_i$ (see equations 9 Hong et al., 2004), favoring an increasing ice water content.

The transects in Figure 5a and b clearly show the ubiquitous ice simulated with WSM5 during westerly and easterly events due to the orographic forcing, whereas the Morrison scheme leads to snow and supercooled liquid formation in both cases (Figure 5c and d) as the Milbrandt, and the Thompson schemes (not shown). As an additional experiment, we made a simulation over the period 6-10th January 2011 where westerlies are simulated (Figure 5a and c). We alter the WSM5 scheme in changing the empirical relationship linking qi to Ni described above. We divide by 100 the resulting $N_i$ (variable `xni` in the code where the empirical equation 5c in Hong et al. (2004) is implemented), which is diagnosed from $q_i$ in the cloud scheme. We plot a similar transect to Figure 5a. This results in more supercooled liquid being simulated (Figure 11a), closer although not yet similar to what - for instance - the Morrison scheme is leading to (Figure 5c). Then, a second simulation with WSM5 was performed (Figure 11b) by changing the INP parameterisation to the one used in the Morrison scheme (more realistic, as shown in section 5.3). It shows no major difference with the original simulation (Figure 5a). The hypotheses (empirical relationships) used for the single-moment ice crystals parameterisation of WSM5 have a more determining impact on the ability to sustain supercooled water than the nature of the INP parameterisation itself in the WRF schemes.

Note that a third simulation was performed with the Morrison scheme using a lower concentration of CCN set to 100 cm$^{-3}$ (instead of 250$^{-3}$). Using a lower CCN concentration does affect the amount of supercooled liquid formed in reducing it (not shown), but the overall distribution of liquid (the order of magnitude) remains similar, as well as the ice, thus not altering the main conclusions of this work. Note that, according to Part 1, the observed average number of drops is 100-120 cm$^{-3}$ in 2010, and 150-200 cm $^{-3}$ in 2011. Hence the Morrison, and Thompson schemes use a drop number similar to the observed upper limit and lower limit, respectively (see Table 3). More simulations focused on particular flights (case studies) would be required to assess in greater details the impact of the CCN concentration compared to the one of the INP parameterisation.

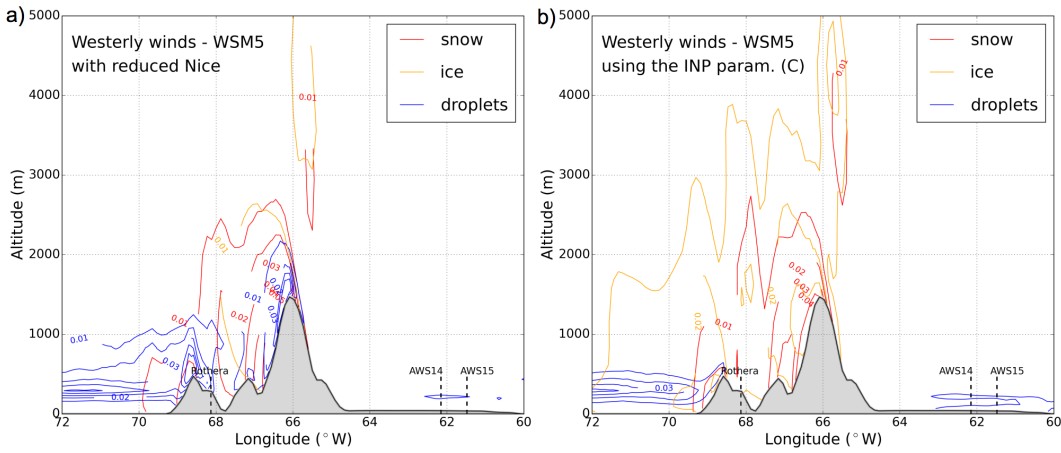

**Figure 11.** Same as Figure 5a with (a) a modified version of WSM5 where the empirical relationship linking the number of ice crystals to IWC is altered (see text for details), and (b) a modified version of WSM5 where the INP parameterisation used (Fmod) is changed to (C) (Cooper, 1986), which predicts less INPs (see section 5.3).

Figure 12 shows the distribution of the cloud mass as a function of the temperature for the transects shown in Figure 5 during westerlies (solid line), and easterlies (dashed line), respectively. The top row is the median simulated mass per 1°C bin for (a) the liquid droplets, (b) the ice crystals, and (c) the snow particles. The bottom row shows the corresponding number of non-null occurrences ($> 0.001$ g kg$^{-1}$) over which the median values are derived in each bin. Figure 12a shows that LWC
simulated down to -10°C by WSM5 is similar to Milbrandt's and Thompson's, and lower than Morrison's for both easterlies and westerlies scenarios. However, the frequency of liquid cloud formation for WSM5 is lower by a factor of two to four compared to other schemes (Figure 12d). At colder temperatures, WSM5 ability to simulate cloud liquid is drastically reduced for both scenarios (Figure 12d). This can be related to the much shallower vertical extent of the WSM5 simulated cloud liquid (Figure 7), which is limited to the warmest subfreezing temperatures. The observations show liquid clouds up to higher
altitudes (Figure 7), and the Morrison, Milbrandt, and Thompson schemes account better for the liquid at these higher altitudes (lower temperatures) than the WRF schemes. Yet, these three schemes do show a decreasing trend in the supercooled liquid mass at temperatures lower than -10°C, despite their steady (slightly growing) ability to simulate ice and snow crystals crystals (Figure 12e and f). Measured vertical profiles do show indeed the presence of ice clouds at temperatures lower than -10°C, and the schemes simulate their occurences better than the supercooled liquid ones (not shown). Interestingly, the frequency of
simulated ice crystals are the most different above -10°C (Figure 12e), where there is an order of magnitude difference between WSM5, and the other schemes. WSM5's IWC is a factor of five to ten larger than for the other schemes (Figure 12b) and this can be mainly explained by its single-moment parameterisation for ice (as described above). This feature also appears in the model outputs at stations AWS14 (Figure 3) and AWS15 (not shown), where the WRF schemes simulate three to four times more cloud ice crystal mass than the other cloud schemes.

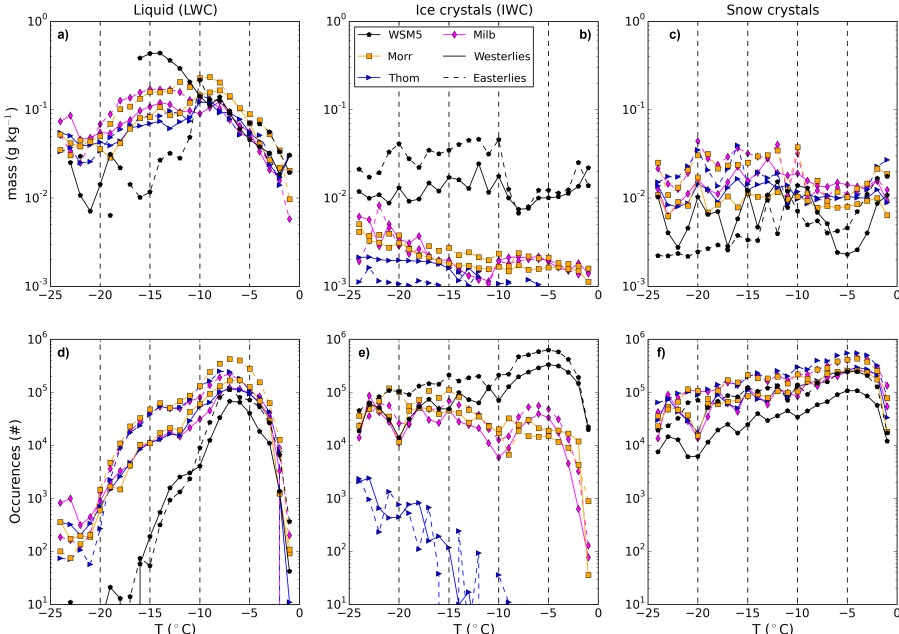

**Figure 12.** Distribution of (a) LWC, (b) IWC, and (c) snow crystal mass, as a function of temperature (per 1°C bin), and the corresponding distribution of non-null occurrences of LWC, ice, and snow (d, e, f, respectively), used to derive those mass distributions, for the transects shown in Figure 5 for westerly (solid line), and easterly wind periods.

Overall the Morrison, Thompson, and Milbrandt schemes are better able than the WRF schemes to form supercooled liquid in both years on both sides of the Peninsula. They perform better in simulating liquid cloud occurrences, and the Morrison scheme gives an average LWC closer to the measured value, except on the east in 2011, when it overestimates the liquid content (section 4.3.1, Figure 6b an d). However, the observed interannual variations of the LWC from 2010 to 2011 East of the Peninsula (Figure 6) and described as statistically significant in Part 1, is not captured by any of the cloud schemes. Part 1 reported on the role of the nature and number of aerosols, of which a subset act as CCN or INP. Based on Part 1, it is likely that the observed regional and interannual cloud microphysics variabilities need an adequate aerosol model in order to be properly simulated. The clouds measured by the aircraft campaigns were exclusively mixed-phase clouds and Table 6 shows that at least down to -13°C the Morrison scheme, and the Milbrandt schemes are the more capable of simulating the observed relative proportions of liquid and ice across this temperature range. However, as shown by Figure 7 little supercooled liquid is simulated above 1500 meters along the flight tracks.

The poorer performances of the schemes to the west of the Peninsula can be seen in the poorer ability to predict the oc-curences of both cloud liquid (Figure 6a and c) and cloud ice (Figure 9a and c) in that region. The number of tracks predicted with some liquid phase represent about 20% (2010) or 40% (2011) of the total observed. Figure 9a and c show a slightly better ability to predict the ice phase but still less than 50% of the tracks are predicted. The associated failure for any scheme to lower the LW surface radiation biases at Rothera station (Table 4) suggest an overall inability to correctly simulate liquid

clouds where they are observed. As noted in section 4.4, the average temperature biases are larger to the West of the Peninsula than to the East of the Peninsula, by 1 to 2°C (Figure 10a). On both sides the average temperature is overestimated. Since the supersaturation depends exponentially on the temperature, the lowest ability of the schemes to predict liquid cloud formation to the West is consistent with the larger temperature biases measured in that region. The warmer oceanic and sea-ice free

influence of the western Peninsula implies more convective processes (compared to the east) that are badly resolved at 5 km resolution and prevent better matching with the aircraft observations. The 5 km horizontal resolution lies in the so-called "grey zone" (resolution 1–10 km) where convective processes are badly simulated and parameterised. An explanation for the bad performances of the schemes above Rothera station may be the complex topography as shown in Figure 5. The station sits in the eastern part of Adelaide island, and is surrounded by mountaineous features. More generally these features will also affect

the air flow reaching the regions where the flights took place, to the west of the Peninsula. By contrast, the eastern part of the Peninsula with the Larsen C Ice shelf has much less complex topographical features, and this should be helping the modelling of the clouds on that side.

Finally, it is worth recalling that working with ice categories, but also different definitions for these ice categories from on scheme to another, makes overall comparisons to flight measurements difficult. The Thompson scheme shows very little

formation of ice crystals, which is readily converted to snow crystals (Figure 12f). Every cloud scheme defines a radius cut-off between ice crystals and snow crystals ranging between 100 and 250 $\mu$m (Table 3), with Thompson having the second largest value at 200 $\mu$m. It is not clear why the Thompson scheme cloud ice crystals numbers are so low (Figure 6, and Figure 12b). Thompson scheme's gives much less frequent and much less abundant crystals at radii below 200 $\mu$m (Figure 12e) compared to the ones above that radius (Figure 12f), and this is at odds with the other schemes, and with the observations. Finally, given

that the observations show an average crystal radius of 150-250 $\mu$m in 2010, and 200-250 $\mu$m in 2011 (not shown), this is probably not ideal to work with cloud schemes having an ice-snow radius cut-off artificially set around those sizes.

## 5.3 The INP parameterisations

All the cloud microphysics schemes investigated in this work rely on INP parameterisations to initiate the ice phase, and here we comment on those. The number concentration of INPs is diagnosed from the modelled atmospheric temperature only.

These empirical parameterisations address the different ice nucleation mechanisms (see introduction). They are triggered at different temperatures or supersaturation thresholds, depending on the cloud scheme (Table 3). They increase exponentially with decreasing temperatures, and can lead to very different INPs concentrations (Figure 13a, coloured lines). The direct consequence of this are clear differences between icy hydrometeors number concentrations, as a function of temperature. To illustrate this Figure 13b shows the median non-null number concentration of total icy condensates (ice crystals, snow and

graupel particles) over the transects shown in Figure 5 for both the westerly (solid lines), and easterly (dashed lines) cases. For deposition/condensation freezing (which does not require the presence of supercooled droplets) the Milbrandt scheme uses the INP parameterisation from Meyers et al. (1992) (Their equation 2.4), while Morrison, and Thompson use the one from Cooper (1986). Both parameterisations are now referred to as (M) and (C), respectively (Table 3). This translates into the drastically different number concentrations at temperatures above -15°C (Figure 13b), because INPs concentrations predicted by (M)

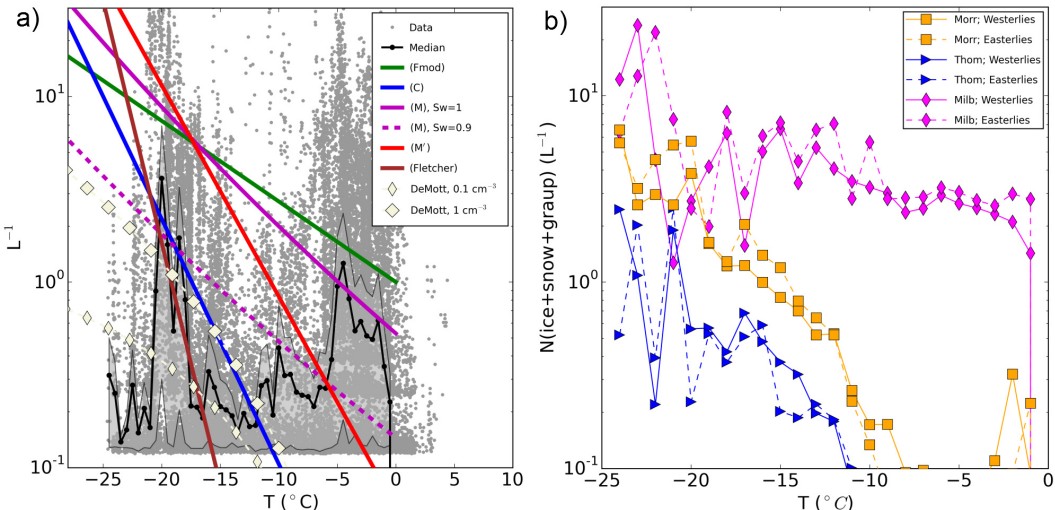

**Figure 13.** (a) Ice crystals measurements data points as a function of the temperature (grey) with their median per 0.5°C bins (black solid line), along with the absolute median deviation in shaded grey. The labelled INP parameterisations used by the different cloud schemes (Table 3) are overplotted. DeMott refers to DeMott et al. (2010)'s INP parameterisation (see text for details). (b) Same as (a) for the Median of the total number concentrations of icy hydrometeors in the same transects used in Figure 5 for both cases, the westerly case (solid lines), and the easterly case (dashed lines).

(Figure 13a, purple lines) are much larger than the ones predicted by (C) (blue line). For contact freezing the Milbrandt, and the Morrison schemes use the INP parameterisation from Meyers et al. (1992) (their equation 2.6), which is referred to as (M′) in Table 3. The Thompson scheme does not explicitly parameterise contact freezing. The consequence is that the Morrison scheme predicts larger amount of icy condensates than the Thompson scheme since (M′) predicts much larger INPs concentrations than

(C). The latter effect is enhanced by the more constraining thresholds on temperature and ice supersaturation for the Thompson scheme to allow for ice formation (Table 3). The Milbrandt scheme relies on the INP parameterisation (M), which predicts much larger amounts of INPs in the deposition mode, which does not depend on the scheme ability to simulate supercooled liquid water in a first place. Interestingly, the Milbrandt scheme has an average solid water content ($SWC_0$, and SWC) almost twice as big as the Morrison, and the Thompson schemes (Figure 4b, and Figure 4b, respectively).

DeMott et al. (2010) developed an INP parameterisation (hereafter called DeMott) using both the temperature and the observed aerosol number concentration, for aerosols larger than 0.5 $\mu$m in diameter (as presented in Part 1) believed to be the main contributors to the worldwide INP population (DeMott et al., 2010). Aircraft measurements used in Part 1 made it possible to derive out-of-clouds aerosol concentration for diameters larger than 0.5 $\mu$m. Using this information to describe the aerosol background, we compared the measured total ice crystals number concentrations to INPs predictions by DeMott, and

to the other INP parameterisations implemented in the WRF cloud schemes. The comparison to the observations was done at temperatures below -9°C, as this is the temperature range over which DeMott parameterisation was derived. This also allows

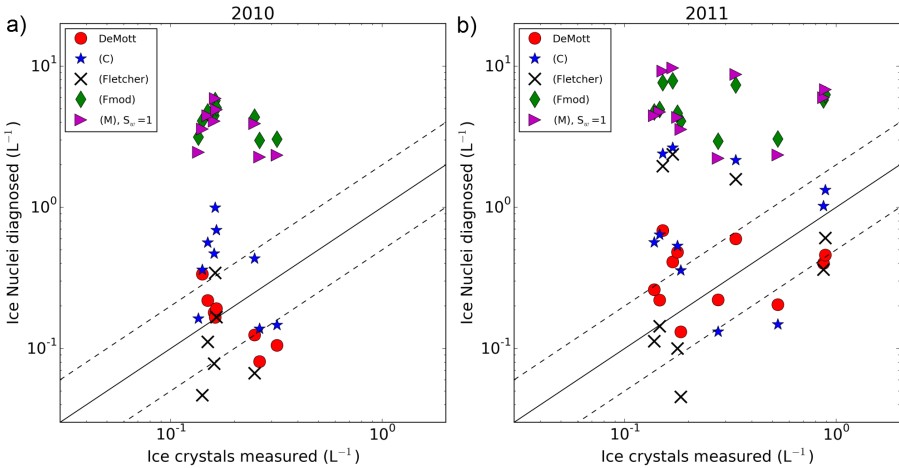

**Figure 14.** Predicted INP number densities versus observed number densities of crystals (flights averages) for various INP parameterisations for (a) 2010, and (b) 2011. The solide lines corresponds to the one-to-one line, and the dashed lines corresponds to a factor of two difference between y and x axes. See Table 3 for the references of the various INP parameterisations.

to discard the warmer temperatures where a secondary ice process was identified as responsible for the ice crystal production around -5°C (see Part 1). The INP parameterisations are meant to account only for the primary ice production process.

For each measurement of crystals below -9°C (one data point every second), a corresponding number of INPs is derived for each parameterisation using the measured temperature. For the background aerosol input to the DeMott parameterisation, we derived a 1 min averaged out-of-cloud aerosol number concentration ($n_{aer}$) within $\pm 30$ s of any crystal measurement. As shown in Part 1 (see their Figure 13) the average $n_{aer}$ ranges between 0.1 cm$^{-3}$ and 1 cm$^{-3}$. Figure 13 shows the DeMott parameterisation for those two values (white markers). We computed flight averages for the observations, and for each INP parameterisation. Figure 14 shows the observed and predicted average values for both years. Table 7 gives the median relative difference ($\epsilon$) between flight-averaged observations (Obs), and the INP parameterisations predictions (INPparam), along with the associated median absolute deviation ($\Delta\epsilon$).

The DeMott parameterisation performs better than any other INP parameterisation as suggested by Figure 14a and Figure 14b. Table 7 shows that DeMott ($\epsilon$=0.5-0.6) performs better than Cooper's (C) ($\epsilon$=1.6-2) (used by the Morrison, and the Thompson's schemes). This is because of its ability to take into account the number concentration of aerosols. For instance if we force a constant value of $n_{aer}$=1 cm$^{-3}$, the DeMott parameterisation performs as poorly as (C) and worst than the original Fletcher's (as opposed to Fmod used by Hong et al., 2004, see their equation 8). However, if we force $n_{aer}$=0.1 cm$^{-3}$ (the average $n_{aer}$ across the Antarctic Peninsula above 2000 m where most primary ice production occurs – see Part 1, and their Figures 13, and 14) then DeMott is performing better than (C), and than any other parameterisation. It performs as good as DeMott with a varying $n_{aer}$ (compare the two first lines of Table 7). The modified version of Fletcher's (Fmod) used in

WSM5 is the worst performer ($\epsilon$>20), followed by the Meyer's (M) parameterisation, which is used by the Milbrandt scheme. ($\epsilon$=4.5-24, for relative humidities characteristics of mixed-phase clouds, 90–100%, respectively).

It should be recalled that the DeMott parameterisation is based on analysis of aerosols, which exclude strong marine influence, and so sea salts were not included (DeMott et al., 2010). Also, aerosol concentrations below 0.3 cm$^{-3}$ have less weight in the DeMott parameterisation's analytical derivation compared to the larger values (0.5-5 cm$^{-3}$), as shown by (DeMott et al., 2010)'s Figure S1 of their supplementary materials. Despite these caveats, the strength of the DeMott parameterisation is to be able to account for the low aerosol number densities at altitudes higher than 2000 m, where primary ice production occurs in the Antarctic Peninsula (Part 1). This makes it, on average, a better candidate than any other IN parameterisations for future work meant to improve the cloud microphysics scheme for Antarctic clouds.

Finally, note that the comparisons are made in times and places where ice crystals were indeed measured, ignoring instances where cloud ice was not measured, but where any INP parameterisation would still predict some crystal production. This challenging issue could probably be dealt with only in managing the coupling of the cloud scheme to an aerosol model able to predict the absence of INPs. Moreover, given existing biases in water vapour and temperature along each flight tracks separately (as opposed to the averages discussed in section 4.4), better calibrating the INP parameterisation consists in only one of the needed improvements for Antarctic clouds modelling, as discussed below.

## 5.4 Additional results on water vapor and temperature biases, and cloud nuclei paramaterisations

In section 4.4 it was shown that the model was able to capture the average temperature within 0.5-2.5°C, and the average water vapour within 0.3-0.7 g kg$^{-1}$. The average simulated temperature and water vapour are within the variability (standard deviation within each longitude bin) of the observations. Although the average behavior of the model matches the average observations, it should be noted that water vapour and temperature biases do hamper the good prediction of the clouds by the model. As an example, Figure 15a shows the time series of the water vapour, and of the temperature, as measured (black line) and as simulated when using the Morrison scheme (red line) for the flight 150. The model fails in simulating the liquid cloud before 18.5h, and after 18.7h (not shown), where the water vapour and temperature biases are the greatest (see the red solid line in Figure 15a), while it does simulate liquid cloud (not shown) where the bias is much reduced (at warmer temperatures, lower altitudes) between those two times (red solid line, in Figure 15a).

An additional simulation was performed over the period 11-20th January 2011 using the Morrison scheme, initializing it ten days later than the simulations used so far (on the 11th of January instead of starting on the 1st). During this period, four flights took place (150-153) and we show flights 150, and 152 (Figure 15). The result is shown as the blue solid line in the Figure 15a and b. The initialization of the model closer to the dates of the airborne measurements does lead to a lower bias in water vapour for both flights. However, the bias in terms of temperature is relatively less reduced across the flights, suggesting that the initialisation of the model has a greater impact on the quality of the water vapour prediction. However the improvement in terms of water vapour does not lead to an improvement of the liquid cloud prediction along the flight tracks, and it even leads to the suppression of the liquid cloud initially simulated along flight 152's flight tracks (not shown). Note that further doubling the number of vertical levels for the above shorter simulation (using 60 eta levels, instead of 30 eta levels) leads to

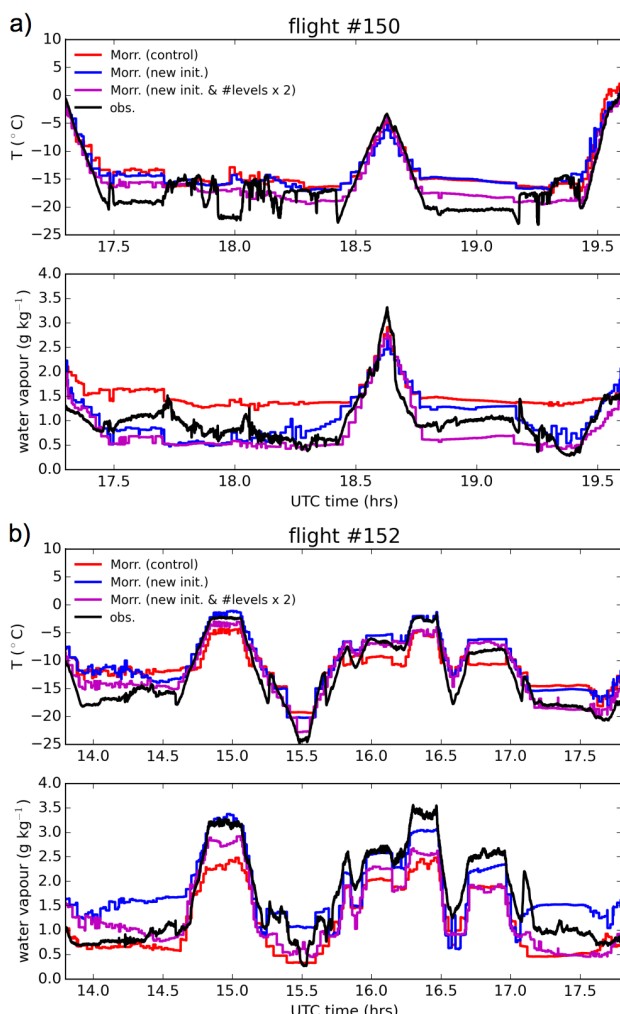

**Figure 15.** Time series of water vapour (g kg$^{-1}$), and temperature (°C) measured during (a) flight 150, and (b) flight 152. For (a), and (b), the observation is the black solid line. Simulation outputs are overplotted: the simulations used to derive the averages presented in this work (red line), a different simulation over the period 11-20th January 2011 (thus with an initialisation closer to the date of the measurements)(blue solid line), and the latter simulation with – additionnally – a doubling of the number of vertical levels (60 instead of 30) (magenta solid line).

a very limited further reduction of the water vapour bias over the four flights, while it does reduce the temperature bias over the flights (magenta solid line in Figure 15a and b). Overall, those results suggest that initializing the model at a closer date to the observations reduces on average the water vapour bias, while doubling the number of levels helps reducing the temperature bias. However, this improvement is not systematic along the flights, and not significant enough, so that the cloud prediction is not really impacted, at least in the investigated cases.

Another run was performed over the same period of the four flights, this time only replacing the INP parameterisation used in the Morrison scheme (see section 5.3), by the Demott parameterization. The result was a much reduced ice crystal water content during the flights (actually lowering the quality of the ice phase prediction; not shown), but no improvement was obtained for the supercooled liquid, what may be explained by the remaining water vapour and temperature biases (not shown) preventing from supersaturation with respect to liquid water to be simulated.

This discussion stresses the need for further work investigating the water vapour and temperature biases in addition to using appropriate cloud scheme (double-moment scheme for the ice crystals), and INP parameterisation, for improving antarctic clouds simulation.

## 6   Summary and Perspective

In this work we provide the first intercomparison of WRF microphysics schemes performances in Antarctica over the Antarctic Peninsula within Polar WRF at 5 km resolution, as well as the first comparisons with in-situ cloud measurements on both sides of the Peninsula. The specificities and properties of the schemes are summarized in Tables 2 and 3. We compared the simulations to averaged aircraft measurements of cloud microphysics properties (Part 1) as well as other atmospheric properties on both sides of the Peninsula and over the two periods of interest (February 2010 and January 2011). This paper was motivated by King et al. (2015) which pointed towards possible problems in the thermodynamic phase simulation in three high resolution model at 5 km resolution over the Eastern Peninsula's Larsen C Ice Shelf, as well as Bromwich et al. (2013a), which demonstrated the presence of Antarctic-wide surface radiative biases within Polar WRF at coarser (60–15 km) resolution. This study is a first step towards the improvement of cloud modelling and operational forecast, with Polar WRF, and AMPS, respectively.

The main results are as follows.

- The surface longwave radiative bias is significantly reduced over the Larsen C Ice Shelf when using the Morrison, the Thompson and the Milbrandt schemes, compared to WSM5 or WDM6.

- Importantly, the Morrison, the Thompson and the Milbrandt schemes, are also the schemes leading to better agreement with aircraft cloud measurements (occurences of the liquid and ice phase, as well as values of the cloud mass mixing ratio), compared to WSM5 and WDM6.

- The Morrison, the Thompson and the Milbrandt schemes perform better than the WSM5 and WDM6 schemes because of their double-moment parameterisation for the ice phase.The latter are single-moment schemes for the ice crystals. A realistic ice parameterisation is essential to the simulation of supercooled liquid.

- The DeMott parameterisation (DeMott et al., 2010), which is not currently implemented in any of the WRF microphysics scheme, better accounts for the ice crystals number densities measured during both campaigns, when using as input the typical concentrations of out-of-clouds aerosols measured above 2000m, where primary ice production occurs (see Part 1). However, the INP alone cannot improve the simulation of the observed clouds.

- The model can simulate the average water vapour, and temperature distribution across the Peninsula, however biases in both fields can still explain the failure in simulating clouds when looking at specific flights (as opposed to the average fields). Moreover, larger bias in temperatures to the west of the Peninsula can explain the lesser ability of the simulations to reproduce the observed clouds.

- As WSM5 is the scheme used in the Antarctic Mesoscale Prediction System, the present work provide new results promoting the improvement of the current cloud scheme implementation in the operational model.

Future work will look at case studies focusing on specific flights at higher spatial and vertical resolution. This will also make use of the latest campaign for measuring Antarctic clouds in the eastern Weddell Sea in November-December 2015 (O'Shea et al., 2017). More investigation of the impact of smaller (temporal or spatial) scale temperature and water vapour biases on 10 mixed-phase clouds simulation will be needed. Often disregarded in simulation works performed over Antarctica not related to cloud studies, cloud schemes should be more systematically considered. Investigating Antarctic clouds, and their impact on the energy budget, is an important step to help quantify the role of atmospheric-driven processes in the evolution of the ice shelves, the glaciers, and the Antarctic ice mass balance, and importantly to improve the forecast for field operations.

*Acknowledgements.* The authors thank Tony Phillips, and Pranab Deb for providing the Antarctic topography adapted to WRF, and derived 15 from Fretwell et al. (2013). The study was funded by Natural Environment Research Council (NERC) under grant NE/K01305X/1. CL also thanks CNES for postdoctoral fellowship funding. The authors thank the two anonymous reviewers, who helped to improve the presentation, and the content of the manuscript.

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

**Table 1.** WRF settings used for the simulations. The number in parenthesis indicates the scheme number (option) in the WRF settings.

| Setting | Value |
|---|---|
| Number of domains | 3 |
| Domains size (px) | 80/130/208 |
| Resolution (km) | 45/15/5 |
| Nummber of vertical levels | 30 |
| Top pressure (hPa) | 50 |
| time step (s) | 180/60/20 |
| Cumulus param | on/on/off |
| LW radiation scheme | RRTM (1) |
| SW radiation scheme | Goddard (2) |
| Surface atmospheric layer | Eta similarity (2) |
| Land surface physics | Noah Land Surface model (2) |
| Planetary Boundary layer | Mellor-Yamada-Janjic (2) |

**Table 2.** Microphysics schemes of WRF (version 3.5.1) used in this work with their predicted cloud variables. DM stands for Double-Moment scheme (see text for details). All prognosed hydrometeors variables are designated by letters as follows. c:clouds droplets; i: ice crystals: r:rain drops; s:snow crystals; g:graupel; h:hail. The Morrison scheme can be used as a Double-moment scheme for droplets only when WRF is used with WRF/Chem. See text for the references related to the cloud microphysics schemes.

| Scheme | Mass | Number | Comment |
|--------|------|--------|---------|
| WSM5 | c,r,i,s | - | Used in the Antarctic Mesoscale Prediction System (AMPS) |
| WDM6 | c,r,i,s,g | c,r | Upgrade of WSM5 to DM for c,r + predicted CCN |
| Morrison | c,r,i,s,g | r,i,s,g | Used in the Arctic System Reanalysis (ASR) |
| Thompson | c,r,i,s,g | r,i | State of the art parameterisation of snow |
| Milbrandt | c,r,i,s,g,h | c,r,i,s,g,h | DM for all hydrometeors + predicted CCN |

**Table 3.** Characteristics of the ice phase, and liquid phase activation, for the microphysics schemes. T refers tp the atmospheric temperature, and qc to the liquid water content. $S_i$ (resp. $S_w$) is the saturation ratio with respect to ice (resp. liquid water). $r_{\text{ice/snow}}$ indicate the cut-off size for icy particles consider either as ice crystals (smaller particles) or snow (larger particles). INP parameterisations (INP param.) account for the various freezing processes presented in section 1: *imm* is immersion freezing; *dep* is deposition freezing; *cont* is contact freezing: *cond.* is condensation freezing; *hom.* is homogeneous freezing (considered as instantaneous, ie straight conversion of liquid to ice). IN/freezing parameterisations' references: (Fmod) is a modified version of Fletcher (1962) presented in Hong et al. (2004); (C) is Cooper (1986); (M) is Meyers et al. (1992) eq. 2.4; (M') is Meyers et al. (1992) eq. 2.6; (B) is Bigg (1953) for probabilistic freezing; (DeM) for Demott et al. (1994) for probabilistic freezing. CCN activation parameterisations: (K) is Khairoutdinov and Kogan (2000); (CP) is Cohard and Pinty (2000).

| Scheme | Triggering of Ice formation | INP param. | $r_{\text{ice/snow}}$ | Droplets/CCN |
|---|---|---|---|---|
| WSM5 | $S_i$>1 | (Fmod) *dep* | | 300 cm$^{-3}$ |
| | [T<0°C & qc>0] | (B) *imm* | 250 $\mu$m | |
| | [T<-40°C & qc>0] | *hom* | | |
| WDM6 | same as WSM5 | | | CCN (K) |
| Morrison | [T<-8°C & $S_w > 0.999$] or [$S_i > 1.08$] | (C) *dep,cond* | | 250 cm$^{-3}$ |
| | [T<-4°C & qc>0] | (M') *cont* + (B) *imm* | 125 $\mu$m | |
| | [T<-40°C & qc>0] | *hom* | | |
| Thompson | [T<-12°C & $S_w > 1.$] or [$S_i > 1.25$] | (C) *dep,cond* | | 100 cm$^{-3}$ |
| | [T<0°C & qc>0] | (B) *imm* | 200 $\mu$m | |
| | [ T<-38°C & qc>0] | *hom* | | |
| Milbrandt | [T<-5°C & $S_i > 1$] | (M) *dep,cond* | 100 $\mu$m | CCN (CP) |
| | [T<-2°C & qc>0] | (M') *cont* | | |
| | [T<-30°C & qc>0] | (DeM) *hom* | | |
| | [T<-50°C & qc>0] | *hom* | | |

**Table 4.** Monthly averaged shortwave (SW) and longwave (LW) surface radiative biases of daily averaged biases over Rothera, AWS14, and AWS15 for the two time periods of interest. The exponent gives the standard deviation (STD) of the daily biases. The number of "x" symbols as subscript tells how significant the difference is between WSM5 and each of the other three schemes (one, two, or three "x" mean statistical significance at the 90%, 95%, or 99% level, respectively). No symbol means that the difference is not significant. Statistically significant reductions in SW/LW biases are emphasised with bold characters.

| Radiation bias | Microphysics scheme | Rothera | | AWS 14 | | AWS 15 | |
|---|---|---|---|---|---|---|---|
| | | 2010 | 2011 | 2010 | 2011 | 2010 | 2011 |
| $SW^{STD}$ (W m$^{-2}$) | WSM5 | $15^{68}$ | $49^{76}$ | $-28^{61}$ | $53^{52}$ | $-22^{52}$ | $48^{51}$ |
| | Morrison | $20^{63}$ | $52^{84}$ | $-51^{60}_{x}$ | $\mathbf{-5^{61}_{xxx}}$ | $-43^{52}_{x}$ | $\mathbf{-12^{50}_{xxx}}$ |
| | Thompson | $46^{67}_{xx}$ | $70^{89}$ | $-24^{62}$ | $37^{52}$ | $-24^{51}$ | $29^{50}_{x}$ |
| | Milbrandt | $7^{63}$ | $48^{82}$ | $-33^{62}$ | $\mathbf{31^{56}_{x}}$ | $-30^{50}$ | $\mathbf{28^{53}_{x}}$ |
| $LW^{STD}$ (W m$^{-2}$) | WSM5 | $-28^{25}$ | $-26^{26}$ | $-11^{28}$ | $-20^{23}$ | $-10^{29}$ | $-22^{20}$ |
| | Morrison | $-22^{22}$ | $-22^{26}$ | $\mathbf{2^{25}_{xx}}$ | $\mathbf{1^{21}_{xxx}}$ | $\mathbf{4^{21}_{xx}}$ | $\mathbf{1^{19}_{xxx}}$ |
| | Thompson | $-24^{26}$ | $-25^{27}$ | $\mathbf{0.5^{26}_{x}}$ | $\mathbf{-6^{21}_{xxx}}$ | $\mathbf{3^{25}_{xx}}$ | $\mathbf{-9^{22}_{xxx}}$ |
| | Milbrandt | $\mathbf{-19^{20}_{x}}$ | $-19^{23}$ | $\mathbf{3^{26}_{xx}}$ | $\mathbf{-6^{20}_{xxx}}$ | $\mathbf{5^{24}_{xx}}$ | $\mathbf{-9^{25}_{xx}}$ |

**Table 5.** Average ratio (%) of the number of occurrences of LWC>0.01 g kg$^{-1}$ in the simulations over the observations. The average is derived from the flight averages.

| Region Year | WSM5 | Morrison | Thompson | Milbrandt |
|---|---|---|---|---|
| West 2010 | 47 | 72 | 69 | 5 |
| West 2011 | 54 | 49 | 4 | 88 |
| East 2010 | <1 | 7 | 3 | 3 |
| East 2011 | <1 | 215 | 130 | 105 |

**Table 6.** Statistics over the flight tracks on the mixed-phase ratio $f_m$=LWC/(LWC+SWC), for temperatures $T > -15°$ (see text for details)

| $f_m$ | Observation | WSM5 | Morrison | Thompson | Milbrandt |
|---|---|---|---|---|---|
| average | 0.84 | 0.6 | 0.91 | 0.66 | 0.79 |
| $\sigma$ | 0.08 | 0.24 | 0.1 | 0.20 | 0.10 |
| max | 0.94 | 0.9 | 1 | 0.92 | 0.95 |
| min | 0.60 | 0.07 | 0.61 | 0.21 | 0.56 |

**Table 7.** The median of the flight-averaged $\epsilon = |\mathrm{INPparam} - \mathrm{Obs}|/\mathrm{Obs}$, and the corresponding median absolute deviation for different INP parameterisations, for the two different years, respectively. DeMott refers to (DeMott et al., 2010), and Fletcher to (Fletcher, 1962), see Table 3 for the other references.

|  | 2010 | | 2011 | |
|---|---|---|---|---|
|  | $\epsilon$ | $\Delta\epsilon$ | $\epsilon$ | $\Delta\epsilon$ |
| DeMott, variable $n_{\mathrm{aer}}$ | 0.48 | 0.26 | 0.61 | 0.27 |
| DeMott, $n_{\mathrm{aer}}$=0.1 | 0.42 | 0.21 | 0.65 | 0.18 |
| DeMott, $n_{\mathrm{aer}}$=1 | 1.82 | 1.04 | 1.98 | 1.33 |
| Fletcher | 0.73 | 0.24 | 0.75 | 0.43 |
| (Fmod) | 26.7 | 4.50 | 21.10 | 12.16 |
| (C) | 1.56 | 1.08 | 2.00 | 1.47 |
| (M), S$_w$=1 | 24.35 | 7.21 | 23.35 | 16.32 |
| (M), S$_w$=0.9 | 4.65 | 1.40 | 4.38 | 3.48 |