# Peer review of "The Microphysics of Clouds over the Antarctic Peninsula – Part 2: modelling aspects within Polar WRF"

_Atmospheric Chemistry and Physics, 2016_

## Referee Comment (RC1) · Anonymous Referee #1 · 21 Feb 2017

Review of "The microphysics of clouds over the Antarctic Peninsula – Part 2: modeling aspects within Polar WRF" by Listowski and Lachlan-Cope

Recommendation: Requires revision before publication

This paper presents comparisons of simulations of clouds over the Antarctic Peninsula using different cloud microphysical schemes in WRF, and includes comparisons against observations described in Part 1 of the manuscript. The main finding of Part 2 is that, regardless of which scheme is used for the simulations, a large misrepresentation of the cloud thermodynamic phase explains a lot of the large radiative biases derived at the poles continent wide (i.e., all schemes fail at predicting as much supercooled liquid mass as seen in observations). The authors conclude that a parameterization scheme

for ice nucleation that depends on both temperature and aerosol content should be implemented in WRF to get a more realistic representation of primary ice production, and hence a better representation of supercooled liquid phase in Antarctic clouds.

The study should be published because unique modeling simulations are compared with a unique set of data over the Antarctic Peninsula. However, the paper can be improved in a number of ways. It is lacking in that it only focus on sensitivities to cloud microphysics, ignoring impacts from other parameterization schemes that could also be affecting the simulations and comparison with observed quantities. The uncertainties associated with the representation of the microphysics needs to be better placed in context of uncertainties associated with the representation of other processes such as boundary layer parameterization schemes. Otherwise, it is possible that the sought after agreement between models and observations may occur due to reasons not associated with the representation of microphysics (i.e., are the right answers being obtained for the wrong reasons). I'm not convinced that simulations that do not consider uncertainties due to other processes can truly assess the ability of the microphysics schemes to model realistic clouds across the Antarctic Peninsula (page 5, line 10) in the absence of these other sensitivities. This is especially true given the statement that none of the microphysics schemes adequately predict the supercooled water: maybe some other process other than microphysics is causing this problem. For example, I think that the amount of fetch off the ocean and its representation would be very important.

The sensitivity studies in the paper emphasize how different schemes affect the modeled cloud parameters. But, in addition to sensitivity in choice of scheme, there can also be sensitivity to some of the parameters (i.e., constants) that are assumed within an individual scheme. Was any effort made to look at the sensitivity to constants within a scheme?

Another weakness of the paper is that the paper does a reasonable job in describing the differences between the simulations and the simulations that are most consistent

with the observations (note, I think most consistent with the observations is a better way of wording it rather than saying the best simulation). However, I a paper in ACP should do a better job in showing why these differences between parameterization schemes are occurring. This can be done by looking at various prognostic terms in the model (i.e., production rates due to various microphysical processes). This could explain why different schemes produce different modeled fields rather than just stating that different schemes produce different cloud fields; the result that different microphysical parameterization schemes gives different modeled fields is not an incredibly new or exciting result in that it has been previously demonstrated in a wide number of other meteorological conditions.

The paper attributes a lot of the disagreement between the modeled and observed fields to the performance of the ice nucleation parameterization: namely, it is state that the ice nucleation parameterization is a major reason why supercooled water is under-estimated. For example, it is stated that forcing a dependence on aerosol amount and temperature, rather than just temperature, could give better agreement. Can more be done with adjusting constants in the ice nucleation schemes within the different cloud parameterizations to better demonstrate this? This way it would be easier to define how much of the difference is due to the different microphysical schemes, and how much is due to the microphysical parameterization schemes (rather than just stating that different parameterization schemes use different representations of INPs that may be causing some of the differences).

The authors conclude that the Morrison scheme is the best performing scheme. It would have been beneficial to show some production and depletion rates of processes producing various hydrometeor categories (especially supercooled water) so that there could be a focus on specific terms and processes that are well represented in that scheme, and hence to better determine where the supercooled water is coming from. That might also help explain why the Morrison scheme is the better performing scheme.

The authors use the Brown and Francis (1995) mass-dimension parameterization to

derive the ice water content. However, this is a very crude parameterization whose use may be inappropriate depending on the mixtures of the shapes and sizes of crystals present. What types of habits were noted in the CIP images? Are they consistent with those assumed in the Brown and Francis parameterization? How sensitive are the calculated IWCs to the assumed parameterization? Does this assumption affect any of the qualitative conclusions?

The smallest domain used in the model simulations is 5 km. This seems quite coarse for a study looking at the impacts of microphysical processes. What vertical resolution is used? Does the choice of horizontal and vertical resolution have any effect on the most important findings of this study? In addition, how sensitive were the simulated fields to the choice of initial conditions (e.g., either the product used or the time at which the simulations were initialized)?

I am also worried about some of the inconsistencies between the microphysics and radiative parameterization schemes that seem to exist in the paper. For example, radiative schemes may make specific assumptions about the shape distributions of various hydrometeor categories. Are these assumptions consistent with what is assumed in the microphysical schemes themselves? Otherwise, an inconsistency between the radiative and microphysical parameterizations could be responsible for some of the discrepancies in the radiative parameterization schemes. Further, the authors themselves seem to state that the assumption of a constant effective radius of 10 micrometers in the radiative scheme may not be consistent with what the microphysical parameterization scheme is assuming.

SPECIFIC COMMENTS

Page 4, line 13: What diameter is the distinct peak located at? Also, recommend using size distribution instead of spectra. Spectra typically refers to radiative quantities.

Page 5, line 14, recommend "graupel is" rather than "graupel are"

Page 8, line 9-10: Degree of agreement is always a very relative term. I think when talking about whether simulations agree or disagree, the writing should be more quantitative stating what the degree of agreement is.

Page 8, line 19: How much graupel was in the observations? Can you give some indication of how much agreement there is between modeled/observed graupel?

Page 9, line 16: Is the radiative scheme consistent with each of the microphysical parameterization schemes?

Page 10, line 4: Shouldn't the radiative schemes be consistent with each of the microphysical parameterization schemes rather than just the K15 scheme?

Page 12, line 8: It would be nice to know the specific terms that are producing the supercooled liquid mass content in the model.

Line 12, line 22: If two schemes use the INP parameterizations, is it fair to say that the microphysical parameterizations are causing the differences, or should it just be attributed to the INP parameterization. This sentence is clear, but I am wondering how this affects some of the preceding discussions in the paper.

Page 13, line 19: If the effects of the mountains are variable/valuable, what are the prognosed values that are affecting the generation of the LWC? This might be especially good to look at given that quantities east and west of the mountain range are being compared.

Page 14, line 7: the statement that the schemes perform worst on the West compared to the East is interesting. But, the paper would be much more insightful if it could identify the processes that are at work so that the schemes are performing better on the West compared to on the East.

Page 15, line 4: There is an overemphasis on what is the best scheme. You might be getting the right answer for the wrong reasons. You should also look at what are some of the prognosed terms allowing this scheme to perform better, and how the sensitivity

to microphysical parameterizations is affected by sensitivity to other parameterization schemes.

Page 17, line 9: How much of the missing mass might not be detected? I think a simple estimate of this can be made if you extend the mass distributions to larger sizes (simple exponential or lognormal fit) to estimate how much of the mass that you are missing.

Page 19, line 1: How much of a difference is important? What difference is acceptable in terms of being a good match with observations?

Page 19, line 7: This paragraph has a very good description of errors associated with these measurements. It would be nice if a similar comprehensive discussion of the uncertainties and errors associated with the microphysical measurements were included, especially given that microphysics is the focus of this study.

Page 21, line 12: Should you state that WSM5 and WDM6 should not be used for these studies when the other boundary layer/other parameterization schemes are being used? This conclusion might not apply if some other boundary layer parameterization schemes are being used.

Page 21, line 17: Is the assumption of a constant 10 micrometer effective radius consistent with the assumptions that are made in the microphysical parameterization schemes? See earlier comment about consistency between microphysical and radiative schemes.

Page 21, lines 29-30: If the INP parameterization is so important, why diagnose it only from temperature if an alternate parameterization is available? It would seem that this could be a greater focus of the study if the INP parameterization is so important.

Page 22, paragraph beginning line 22: Can you show how much specific terms are contributing to the ice production processes rather than having generic descriptions of these processes?

---

## Referee Comment (RC2) · Anonymous Referee #2 · 24 Feb 2017

The authors present a modeling study of various microphysics schemes and compare model results to observations to aircraft measurements in the Antarctic. They find significant differences in predictions of the cloud thermodynamic phase and conclude that the Morrison scheme is the best scheme as it leads to the best model/observation comparison in terms of clouds and radiation. Fore The number of model studies on cloud in Antarctica are sparse, even though the region is very sensitive to changes in climate. Therefore, the current study might represent an important contribution to the literature if my comments below will be addressed.

General comments

1) I am not convinced that the authors can indeed conclude on 'the best performing'

microphysics scheme based on their model studies. They show that the Morrison scheme predicts best the super-cooled liquid in clouds, followed by the Milbrandt and Thompson schemes since all three have a sophisticated description of the various ice categories. Given the large uncertainties that are associated with the representation of clouds in general and of mixed-phase clouds in particular, the comparison to only a few parameters might not be sufficient to identify 'the best' scheme. Uncertainties in the radiation and boundary layer schemes might lead to a prediction bias such as the microphysics scheme may give the right answer for the wrong reason. I suggest a more careful wording throughout the manuscript. At several places in the manuscript, the language is rather colloquial; I listed several instances below but encourage the authors to carefully read and improve language where possible.

2) The authors state that none of the applied microphysics schemes was specifically developed for Antarctic clouds and, therefore, their study is a first step in exploring the skills of the different schemes for such scenarios. However, I am missing a more conclusive statement on what improvements should be done in these schemes to optimize them for Antarctic clouds.

3) It is mentioned that the liquid phase of clouds (drop activation) is either described based on a fixed number of droplets or it is a function of the CCN (p. 7, l. 5). Drop activation is an essential process that determines the microphysical properties of a cloud. How much difference is caused between the different microphysics schemes due to differences in the prescription of the cloud droplet number?

4) The authors describe in length the differences in the results predicted by the microphysics schemes in Section 4.1. However, in order to understand the differences and to assess the skill of the various schemes, a more detailed discussion of the underlying processes is needed. Such analysis will help to identify the 'best scheme' for the right reasons and to improve existing microphysics schemes.

5) Several previous studies have highlighted the importance of ice particle shapes for

ice growth and sedimentation and therefore for the partitioning between ice and liquid phases (e.g., Sulia and Harrington, JGR, 2001). How are ice particle shapes treated here? Are the predicted shapes in general agreement with observations?

6) Only in Section 5, it is mentioned that the radiation scheme assumes a drop radius of cloud droplets. How does this value compare to observations and how does it affect in general the radiation prediction? Are there any studies (not necessarily in Antarctica) that have discussed biases in predicted radiation due to a constant effective radius?

7) Each microphysics scheme includes a different parameterization of ice formation. Different INP parameterizations can lead to significant differences, as it has been shown in several literature studies (e.g. Eidhammer et al., JGR, 2009). How much of the predicted differences in comparison to observations and model/model comparisons can be ascribed to the differences in INP parameterizations? Why not using the DeMott parameterization as the base case as it has been shown to perform better than the temperature-dependent-only parameterization in previous studies?

8) The authors should better justify their choice of a 5 km resolution. At the end of Section 5, they state that convective processes are badly resolved on this resolution; however, they fail to discuss the consequences of this caveat for simulating clouds.

9) Throughout the paper, expression such as 'the xxx scheme forms more liquid' etc should be avoided (e.g. p. 13, l. 15). The schemes themselves do not form or produce any ice or water. Wording such as 'Using the xx scheme, it is predicted ...' (or similar) should be used instead.

Minor comments

p. 2, l. 34: Not clear what 'the latter case' refers to. – Do you mean all three, i.e. immersion, contact or condensation freezing? If so, it might not be fully correct, since droplets may scavenge INP before ice nucleation and therefore INP and CCN are not the same.

[Figure]

p. 7, l. 18: The beginning of the sentence is not clear. Does the AMPS model predict a liquid phase or not?

p. 8, l. 10: What do you mean exactly with 'twice to four times more liquid clouds'? – Does it refer to the number of clouds or to the liquid water content of the clouds?

Figures 2 and 3: In particular, in the upper panels, the dotted lines are pretty hard to distinguish.

p. 11, l. 6: I think (ice) should be removed – or SWC0 added at the beginning of the sentence (?)

p. 17, l. 5: Can an estimate be given how much the observed mass is biased due to the small and large cut-off diameters? – An extrapolation of the measured size spectrum might be better than the current complete omission.

Technical comments

p. 1, l. 9: 'struggle' is rather colloquial

p.2, l. 10; p. 3, l. 7, and other places: Antarctic

p. 3, l. 6: Southern

p. 4, l. 14: discriminated

p. 4, l. 26: one domain

p. 5, l. 11: developed

p. 6, l. 10: cloud formation

p. 6, l. 25: flight measurements

p. 7, l. 21: observed

p. 8, l. 1: domain

p. 8, l. 27: demonstrate

p. 9, l. 2: tests

p. 11, l. 8: Add unit to LWC (0.02)

p. 12, l. 28: wind regimes

p. 12, l. 33: either 'shows a. . .transect' or 'shows . . .transects

p. 16, l. 19: flights

p. 17, l. 14: equal

p. 19, l. 27: increases

p. 21, l. 9: simulation

p. 21, l. 24: are consistent

p. 23, l. 7: agrees

p. 25, l. 3: crystal

p. 26, l. 15: 'The transects in Figure 5a and b clearly show..'

---

## Author Comment (AC1) · 20 Jun 2017

Dear Reviewers, and Editor,

Please note that the full response (including the highlighted version of the paper, at the end of the document) is attached to this reply. Thank you.

Regards.

——————- Below is the plain copy of the full response (without the highlighted paper) (Note that the pdf version has colorcoded comments/answers) ——————-

Revision of "The Microphysics of Clouds over the Antarctic Peninsula – Part 2:

[Figure]

modelling aspects within Polar WRFÂăÂż We thank the reviewers for their comments.

This document is organised as followsÂă: - p. 1 (of the pdf) Introduction to the answers to the reviewers (some general comments) - p. 2 Answers to reviewer #1 - p. 11 Answers to reviewer #2 - p. 18 The highlighted version of the new manuscript

Introduction to our answers :

The motivation for this paper is the hypothesis by King et al. (2015) that radiative biases over the Larsen C ice shelf (Eastern Peninsula) were due to the lack of cloud liquid water. They worked at 5 km resolution and use three models – including AMPS (the Antarctic Mesoscale Prediction System, based on Polar WRF). Bromwich et al. (2013a) also reached this conclusion with Polar WRF at coarser resolution (15 km, and 60 km) continent-wide. However Bromwich et al. (2013a) not investigate the phase simulated by the clouds (only the cloud cover as a whole).

The main goal of the present paper is to provide with a first evaluation of the cloud schemes as they are using the (averaged) flight measurements, to show and discuss their impact on the surface radiative bias, to highlight some important aspects like the need to use double-moment ice microphysics parameterization (unlike WSM5/WDM6) to better capture supercooled cloud water, to show that the current scheme used in AMPS (WSM5) has a lesser ability to model the clouds, to discuss the biases in water vapour and temperature, and to discuss the INP parameterisation. The conclusion has been simplified to clearly emphasize those points.

Note that section 5.2 (ÂńÂăThe ice phase parameterisationÂăÂż, now ÂńÂăThe INP parameterisationsÂăÂż) and section 5.3 (ÂńÂăSimulating the clouds thermodynamic phaseÂăÂż) have been swaped to putt less emphasis on the effect of the INP parameterisation because it is indeed the case that we do not investigate its effect in details (although we do discuss it).

Note that in sectin 5.3 (former section 5.2), we have decided to delete the paragraph

and Figure about the Hallett Mossop process as this discussion was making the section lenghty and would be more adapted to more in-depth investigation of the processes producing the ice using case studies (specific flights). The deleted text and Figure are shown at the very end of this document, as a reminder.

Additionnally a section 5.4 has been added to emphasize the role of water vapour, and temperature biases (and other model sensitivity consideration on the initialization of the model, or the number of vertical levels). In doing so we insist on the fact that changing the INP parameterisation is not enough for improving the simulation of the clouds. The previous version of the paper suggested that the INP parameterisations were the main reason of missing liquid cloud in the simulation, and we wanted to correct this.

We have simplifed the conclusion to highlight more clearly the main aspects of this work, while we now insist on the need for further work on improving/investigating the temperature/water vapour biases as well. (The abstract has been amended, accordingly.)

Please note thatÂă: - Edited/new text is highlighted in the paper. A highlighted section title means that the section is new (5.4) or fully changed/reformulated (conclusion).

- The reviewers comments below are in black. Our answers are in blue color. Below each answer, the page, and line numbers of the related edited/new text of the paper are indicated.

Anonymous Referee #1

————————————————————————————- This paper presents comparisons of simulations of clouds over the Antarctic Peninsula using different cloud microphysical schemes in WRF, and includes comparisons against observations described in Part 1 of the manuscript. The main finding of Part 2 is that, regardless of which scheme is used for the simulations, a large misrepresentation of the cloud thermodynamic phase explains a lot of the large radiative biases derived at the poles continent wide (i.e., all

schemes fail at predicting as much supercooled liquid mass as seen in observations). The authors conclude that a parameterization scheme for ice nucleation that depends on both temperature and aerosol content should be implemented in WRF to get a more realistic representation of primary ice production, and hence a better representation of supercooled liquid phase in Antarctic clouds.

We have emphasized that the main finding of the paper is that changing from the AMPS scheme WSM5 to other schemes does reduce the radiative bias and some schemes substantially reduce the bias on the Larsen C ice shelf. We feel that this makes the paper clearer. In particular we find that double moment schemes (Morrison, Thompson, Milbrandt) perform better than the single moment scheme that is used as standard in AMPS, to simulate liquid cloud. We feel that this is an important finding as it is easy to implement this simple change within operation model while more detailed changes such as implementing new INP parameterisation would require more invesigation first. Moreover we also stress the fact that working on the water vapour/temperature biases is also a mandatory step.

————————————————————————————- The study should be published because unique modeling simulations are compared with a unique set of data over the Antarctic Peninsula. However, the paper can be improved in a number of ways. It is lacking in that it only focus on sensitivities to cloud microphysics, ignoring impacts from other parameterization schemes that could also be affecting the simulations and comparison with observed quantities. The uncertainties associated with the representation of the microphysics needs to be better placed in context of uncertainties associated with the representation of other processes such as boundary layer parameterization schemes. Otherwise, it is possible that the sought after agreement between models and observations may occur due to reasons not associated with the representation of microphysics (i.e., are the right answers being obtained for the wrong reasons). I'm not convinced that simulations that do not consider uncertainties due to other processes can truly assess the ability of the microphysics schemes to model realistic clouds across the

[Figure]

Antarctic Peninsula (page 5, line 10) in the absence of these other sensitivities. This is especially true given the statement that none of the microphysics schemes adequately predict the supercooled water: maybe some other process other than microphysics is causing this problem.

We have concentrated on the cloud physics schemes as we felt that this scheme was most likely to be the one that was causing the errors seen by King et al (2015), and because no study focused on Antarctic clouds have investigated this before. Apart from the cloud scheme we have not altered any other physics schemes as although they can be responsible for large errors we feel that many have already been investigated egĂ: the boundary layer parameterization, although it is one of the key parameterization, due to its implication in the heat/energy transfer between the surface and the troposphere. We are using the one used in the operational forecast model AMPSĂ; furthermore Deb et al. (2016) showed that the model performance at the surface is most sensitive to the choice of PBL scheme and show that the Mellor Yamada-Janjic (MYJ) scheme is the best performer in terms of the temperature diurnal cycle (in west Antarctica) at 5 km resolution. Hence we are confident that we are using a relevant framework to investigate the cloud schemes' behavior.

————————————————————————- For example, I think that the amount of fetch off the ocean and its representation would be very important.

It is possible that larger scale dynamical effects can be affecting the clouds but on the whole the wind field is well forecast by WRF (for instance, wind direction measured at the flights/station is well captured by the model as recalled at the beginning of section 4.2) and so it is unlikely that errors in the fetch over the open ocean are the problem.

————————————————————————- The sensitivity studies in the paper emphasize how different schemes affect the modeled cloud parameters. But, in addition to sensitivity in choice of scheme, there can also be sensitivity to some of the parameters (i.e., constants) that are assumed within an individual scheme. Was any effort made to look at the sensitivity to constants within a scheme?

There are many constants in the cloud schemes but we were concerned in this study with difference between the schemes – if we had altered constants in every scheme we would have had to do many more runs and this was not feasible we the resources we had available. In this paper we focus on more general issues like the performance of the schemes as they are implemented, and the fact that the schemes are single/double moment for the ice, the water vapour/temperature biases, and the INP parameterisation (in the discussion) and this allows a managable number of runs. However we do show some new results discussing the change of the INP parameterisationÂǎ:

See p. 24 Line 18-26, and Figure 11

AlsoÂǎin the new section 5.4, page 30.

—————————————————————- Another weakness of the paper is that the paper does a reasonable job in describing the differences between the simulations and the simulations that are most consistent with the observations (note, I think most consistent with the observations is a better way of wording it rather than saying the best simulation). However, I a paper in ACP should do a better job in showing why these differences between parameterization schemes are occurring. This can be done by looking at various prognostic terms in the model (i.e., production rates due to various microphysical processes). This could explain why different schemes produce different modeled fields rather than just stating that different schemes produce different cloud fields; the result that different microphysical parameterization schemes gives different modeled fields is not an incredibly new or exciting result in that it has been previously demonstrated in a wide number of other meteorological conditions.

Although we appreciate the reviewer's point this study is intended to give a general

picture of the cloud simulation performances of different schemes over the Antarctic Peninsula, and as this has never carried out before we did not feel it nessecary to start investigating the production terms of all relevant processes of the different schemes. A major difference as we show it is the fact of having single-moment ice microphysics (and the related dependency between the number of crystals, and the ice water content) (WSM5/WDM6) versus double-moment ice microphysics (Morrison, Thompson, and Milbrandt). Some additional simulation results have been carried out and are presented (section 5.2, and new section 5.4) or just reported (section 5.1) to improve the discussion.

These different additions are further mentioned in the answers below.

————————————————————————————- The paper attributes a lot of the disagreement between the modeled and observed fields to the performance of the ice nucleation parameterization: namely, it is state that the ice nucleation parameterization is a major reason why supercooled water is underestimated. For example, it is stated that forcing a dependence on aerosol amount and temperature, rather than just temperature, could give better agreement. Can more be done with adjusting constants in the ice nucleation schemes within the different cloud parameterizations to better demonstrate this? This way it would be easier to define how much of the difference is due to the different microphysical schemes, and how much is due to the microphysical parameterization schemes (rather than just stating that different parameterization schemes use different representations of INPs that may be causing some of the differences).

We feel that we may have been missunderstood and so have changed the way the

results are presenting in the paper. We have now put less emphasis ont the importance of INP parameterisation as the main cause of the differences between WSM5/WDM6, and Thompson/Milbrandt/Morrison, is the fact that the former rely on singe-moment ice microphysics with a parameterisation linking Ni (number of crystals) and qi (mass of the ice phase), that lead to enhanced production of ice.

To this respect, the choice of the INP appears to be secondary in comparison whether the scheme is single or double moment. Also we stress the need of reducing the water vapour/temperature biases to help improving the cloud simulation but this is probably outside the scope of the present study.

As such these are interesting results to Polar WRF/AMPS users.

————————————————————————- The authors conclude that the Morrison scheme is the best performing scheme. It would have been beneficial to show some production and depletion rates of processes producing various hydrometeor categories (especially supercooled water) so that there could be a focus on specific terms and processes that are well represented in that scheme, and hence to better determine where the supercooled water is coming from. That might also help explain why the Morrison scheme is the better performing scheme.

In this study we have concentrated on the average performance of the schemes over many different cases – those that were reported in part one of this study – rather than look in detail at a particular case. We think that the detailed study of production/depletion rates is best left to future work focusing on higher resolution simulation and case-studies (specific flights) and rather than the mean values we use here. These average results show that the double moment microphysics schemes (Morrison, Thompso, Milbrandt) perform better than (WSM5, WDM6) in modelling the liquid

phase. We find that the Morrison scheme is performing better but no longer claim that it is the Ấńấăbest performing schemeĂăĂż as we agree more detailed study of the processes will be needed.

———————————————————————- The authors use the Brown and Francis (1995) mass-dimension parameterization to derive the ice water content. However, this is a very crude parameterization whose use may be inappropriate depending on the mixtures of the shapes and sizes of crystals present. What types of habits were noted in the CIP images? Are they consistent with those assumed in the Brown and Francis parameterization? How sensitive are the calculated IWCs to the assumed parameterization? Does this assumption affect any of the qualitative conclusions?

Although there are more recent mass diameter laws that we could have consdered using Brown and Francis was the only scheme available within our image processing software (OASIS) and considering the errors we would expect anyway from our low resolution CIP probe. It also meant that our results were directly comparable with other results from polar studies who had used the same method (for example Lloyd et al (2011).

———————————————————————- The smallest domain used in the model simulations is 5 km. This seems quite coarse for a study looking at the impacts of microphysical processes. What vertical resolution is used? Does the choice of horizontal and vertical resolution have any effect on the most important findings of this study? In addition, how sensitive were the simulated fields to the choice of initial conditions (e.g., either the product used or the time at which the simulations were initialized)?

We answer the various points belowĂă:

* It is a similar horizontal resolution to previous work (ie King et al. 2015) that we wish compare this study too. It is also a higher resolution to Bromwich et al. 2013a (15-60 km resolution), who investigated radiative biases over the continent using Polar WRF. We have completed many runs for this study – we have been looking at more

than 1 month for each scheme, each year (2010, 2011) during which many observational flights were completed. This has meant that resources would not have allowed a significantly higher horizontal resolution. (However, we can say that other unpublished/preliminary simulations comparing WSM5 and the Morrison scheme at slightly higher resolution (2 km), and doubling the number of vertical eta levels (60, instead of 30), over the Weddell sea, and related to our 2015 campaign do lead to the same conclusions presented in the manuscriptÂă: that the WSM5 scheme underpredicts significantly more the supercooled liquid water, compared to the Morrison scheme, that the radiative biases are significantly reduced using the Morrison scheme, that the water vapour/temperature biases are also responsible for disagreement between the model and the cloud measurements). (For the vertical resolution, see further below).

* Bromwich et al. (2013a) showed that using ERA-Interim reanalysis products for initial, and boundary conditions produces the best skills within Polar WRF, and thus we decided to use ERA-Interim data.

See p.6 , L. 1-2

* 30 (eta) levels are used, with the levels being closer to each other closer to the surface. See Table 1, and explanation when deriving the averaged vertical LWC profiles (p17. L4-7)Âă; the maximum distance between model vertical levels is approximately 500m below 4500 meters, with an average distance of approximately 350m, and down to 10-50 m above the surface).

See p. 17 L4-7

* A doubling of the number of levels does allow to reduce the biases of water vapor and temperature in some places, but it should not affect our general result, as a significant improvement in the cloud simulation is not observed. The initialisation closer to the date of observation reduces the water vapour bias (mainly), but it does not necessarily need to better cloud simulation, and the general picture we provide here when comparing the schemes will not be altered. A discussion with small sensitivity test on specific flights

was added to raise those concerns, as we agree they must be discussed to some extent. Hence, the additional section 5.4.

See new section 5.4 p. 30.

———————————————————————- I am also worried about some of the inconsistencies between the microphysics and radiative parameterization schemes that seem to exist in the paper. For example, radiative schemes may make specific assumptions about the shape distributions of various hydrometeor categories. Are these assumptions consistent with what is assumed in the microphysical schemes themselves? Otherwise, an inconsistency between the radiative and microphysical parameterizations could be responsible for some of the discrepancies in the radiative parameterization schemes. Further, the authors themselves seem to state that the assumption of a constant effective radius of 10 micrometers in the radiative scheme may not be consistent with what the microphysical parameterization scheme is assuming.

We understand complitly that radiation scheme may not be consitant with the cloud physics scheme and we did consider changing both at the same time. However we felt that it would be better to only change one scheme at a time. More generally, those inconsistencies mentioned by the reviewer are part of the drawbacks in using a flexible model like WRF where different parameterizations can be used in different combinations. However, and importantly, the approach used in the SW radiative scheme (where reff is held constant) is to parameterize the optical depth for water/ice cloud as a function of reff based on the fact that (A solar radiation parameterization for atmospheric studies.NASA Tech. Rep. NASA/TM-1999-10460,vol.15,38 pp)¢a: Âń˘AˇaTheoretical considerations and radiative transfer calculations have shown that cloud single-scattering properties are not significantly affected by details of the particle size distribution and can be adequately parameterized as functions of the effective particle sizeˇAˇaˇA¿z (Fu, 1996; Hu and Stamnes, 1993; Tsay et al., 1989). We appreciate that this may be a strong assumption, especially that the width of the size distribution should impact the radiative properties of the cloud (but then the width also impact

the effective radius value). Nevertheless, as the parameterization used does not assume any type of size distribution, we think that it can be used with the different cloud schemes despite their own different assumptions on the hydrometeors size distributions.

See p. 23 L. 11-14

Also, we derived an averaged reff for the droplets in the measurements of about 7.6 um in 2010, and 7.2 um in 2011, close to the 10 um assumed (constant) in the SW radiative scheme. We have ran the model over the period 11th-20th 2011 setting reff=7um and observed no consistently better radiative bias. Having this reff constant throught the simulation is certainly problematic (more than the fact of having a fixed value either to 7 or to 10 um) and could be one of the reason of larger shortwave biases. The LW bias is more sensitive to the ice/liquid content, hence is systematically reduced (to the east of the Peninsula) with cloud schemes simulating more liquid clouds (Morrison, Thompson, Milbrandt).

See p. 23 L. 17-21

SPECIFIC COMMENTS ——————————————————————————-

Page 4, line 13: What diameter is the distinct peak located at? Also, recommend using size distribution instead of spectra. Spectra typically refers to radiative quantities.

It is for a diameter between 8 and 12 um.

See p. 4 L 19-20

and we have replaced spectrum by size distribution.

Page 5, line 14, recommend "graupel is" rather than "graupel are"

We have corrected that.

Page 8, line 9-10: Degree of agreement is always a very relative term. I think when

talking about whether simulations agree or disagree, the writing should be more quantitative stating what the degree of agreement is.

We made the writing more quantitative.

See p 8 L. 24-29

Page 8, line 19: How much graupel was in the observations? Can you give some indication of how much agreement there is between modeled/observed graupel?

We don't think we can resolve graupel properly in our observations. It is difficult to tell the difference between an irregular ice crystal and a heavily rimed one (graupel) by looking at the shadow images.

Page 9, line 16: Is the radiative scheme consistent with each of the microphysical parameterization schemes?

Please refer to our answer to the last major comment above, and related to this specific point.

Page 10, line 4: Shouldn't the radiative schemes be consistent with each of the microphysical parameterization schemes rather than just the K15 scheme?

Please refer to our answer to the related major comments above.

Page 12, line 8: It would be nice to know the specific terms that are producing the supercooled liquid mass content in the model.

We did not perform simulations allowing to target specifically the production terms, as it is not the goal of this paper. However, the production of supercooled liquid results from the supersaturation remaining after the ice has condensed. The process is condensational growth and is depending on how efficient the ice formation is. In particular, in this case, we show that the double-moment schemes for ice are performing better than the single-moment schemes for ice WSM5/WDM6 (which are single- and double-moment for the liquid phase, respectively). The latter have Nice and IWC not evolving

independently (single-moment), and the relationship used to link both leads to high values of IWC. We now explain this in section 5.2, particularlyĂă:

Line 12, line 22: If two schemes use the INP parameterizations, is it fair to say that the microphysical parameterizations are causing the differences, or should it just be attributed to the INP parameterization. This sentence is clear, but I am wondering how this affects some of the preceding discussions in the paper.

To avoid confusion, we removed this sentence, as it anticipates too much on what corresponds to the discussion part, and since – as stated in our introduction to this document – we decided to put let emphasis on the INP parameterisation, when revising our work.

Page 13, line 19: If the effects of the mountains are variable/valuable, what are the prognosed values that are affecting the generation of the LWC? This might be especially good to look at given that quantities east and west of the mountain range are being compared.

Note that we compare quantities to the east and to the west of the Peninsula because the statistics of the observations were much better on each side as the aircraft tended to avoid clouds when crossing the mountains. Also the data have been grouped in East/West ÂńÂăbinsÂăÂż to improve the statistics when comparing averaged quantities (as in Part 1).

The supercooled LWC will be generated maintly depending on the supersaturation removal by the ice microphysics processes. This is shown for instance by the additional simulation using WSM5 scheme with a modified relationship between qice (mass) and nice (number), where the factor linking qice to nice is divided by 100 in the empirical relationship used for WSM5's ice crystals. This edit makes WSM5 generating supercooled LWC (say comparable to Morrison). In comparison, changing the INP parameterisation does not increase the supercooled liquid simulated with WSM5 across the Peninsula. As presented above, this is now emphasize in the discussion part section 5.2.

Page 14, line 7: the statement that the schemes perform worst on the West compared to the East is interesting. But, the paper would be much more insightful if it could identify the processes that are at work so that the schemes are performing better on the West compared to on the East.

It is actually hard to say.

The average temperature bias is larger to the west compared to the east, and now it is clearly recalled in the discussion, as well as the topography which is completely different to the west and would require higher resolution simulation in future work.

and

Page 15, line 4: There is an overemphasis on what is the best scheme. You might be getting the right answer for the wrong reasons. You should also look at what are some of the prognosed terms allowing this scheme to perform better, and how the sensitivity to microphysical parameterizations is affected by sensitivity to other parameterization schemes.

We agree that we emphasized too much on what the best scheme could be and we changed that in the paper (as already stated above). As for the second part of the comment, we rather state that the double-moment ice schemes is more important than a double-moment liquid scheme to improve the supercooled liquid water simulation, and has to be preferred to a single-moment ice scheme. In terms of other parameterization schemes we explain above that choosing the AMPS (Antarctic operational model) configuration is a relevant choice, and mention Deb et al. (2016)'s paper as for validating

the most relevant PBL scheme in an Antarctic environment.

See p. 5 L. 7-12

Page 17, line 9: How much of the missing mass might not be detected? I think a simple estimate of this can be made if you extend the mass distributions to larger sizes (simple exponential or lognormal fit) to estimate how much of the mass that you are missing.

We have approximated the crystal size distribution by exponential distributions to get some estimate of the error related to the particles that are missed above D=1.5 mm. Overall, the average relative error on the water ice content is about 5% in 2010, and 8% in 2011. We also introduced a new Figure 8 to show this.

See P18 L.12-32, and the related new Figure 8.

Page 19, line 1: How much of a difference is important? What difference is acceptable in terms of being a good match with observations?

In this sentence we compare the averaged value of the mixed-phase ratio, and its variability to the observed quantities, and state which scheme produces the closest agreement to the observations. Here, the best match is considered as the one which gives the closest value of the so-called mixed-phase ratio to the observed one. (but we might be missing your point, here.)

This paragraph is now p. 19 L17-22.

Page 19, line 7: This paragraph has a very good description of errors associated with these measurements. It would be nice if a similar comprehensive discussion of the uncertainties and errors associated with the microphysical measurements were included, especially given that microphysics is the focus of this study.

The errors associated with the microphysical measurements and the instruments are considered in part one of this study (Lachlan-Cope et al. 2016, ACP). ÂăFor instanceÂăwe explained that the ÂńÂăthe CAS data for most flights agrees within 15%

with the Hotwire LWC Sensor – part of this discrepancy is attributed to the Hotwire LWC Sensor's tendency to underread at high values of LWC. The CIP images particles between a diameter of 25 $\mu$m and 1.5 mm, with 25 $\mu$m pixel resolution, and had not at the time of this campaign been fitted with anti-shatter tips. However, a study of the particle inter-arrival times indicated very few shattered particle, and these were removed by eliminating particles that arrive within 1 $\mu$s.ÂǎÂż (Part 1, Lachlan-Cope et al. 2016, section 2.1)

In the present paperÂǎ:

- the errors (or bias) we can speak of, related to the LWC, are explained in the section 4.3.1 (first paragraph) already and are related to the inability to identify correctly the entire population of droplet in the large size-range (small size range of the CIP/crystals). Errors on the CAS measurements itself as such are in comparison negligible. Also, the hotwire probe (bulk measurements) backs up the LWC derived from the Cloud Aerosol Spectrometer (it is said in part I, although we do not recall this here).

See p14 L. 1-2 and p15 L1-9

- As explained above, we now present some errors for the IWC in section 4.3.2 (Ice phase) related to the icy particles not detected above D=1.5 mm (using exponential distributions to extrapolate the number of crystals), that lead to an underestimation of the real ice water content.

See p18 L 7-26

Page 21, line 12: Should you state that WSM5 and WDM6 should not be used for these studies when the other boundary layer/other parameterization schemes are being used? This conclusion might not apply if some other boundary layer parameterization schemes are being used.

We think we can state that given our settings WSM5/WDM6 which rely on the same ice microphysics parameterization (the fact that WDM6 has also graupel does not change

the results since graupel are very few and almost always absent in the simulation output) should not be used because they perform less well in terms of supercooled water simulation. Since we explain that the ice microphysics part of the cloud parameterization is responsible for this, we don't think using another PBL would lead to different result. Also, the PBL scheme we are using was shown by Deb et al. (2016) to perform the best and is used in AMPS, and as such is a relevant framework to work with.

Page 21, line 17: Is the assumption of a constant 10 micrometer effective radius consistent with the assumptions that are made in the microphysical parameterization schemes? See earlier comment about consistency between microphysical and radiative schemes.

Please refer to our answer in the major comments.

Page 21, lines 29-30: If the INP parameterization is so important, why diagnose it only from temperature if an alternate parameterization is available? It would seem that this could be a greater focus of the study if the INP parameterization is so important.

Although we have decided to keep some discussion about the INP parameterization in the paper - because we believe such report can motivate further work - we toned down the statement that INP are mostly responsible for the differences between schemes. Indeed, for our work, the most important factor is the ice microphysics single-moment param. of WSM5/WDM6 (See page 24 L14-24) and local vapour/temperature biases that can lead to missing clouds in the model (see new section 5.4)

Page 22, paragraph beginning line 22: Can you show how much specific terms are contributing to the ice production processes rather than having generic descriptions of these processes?

We have reorganised the discussion and made it insist less on the possible importance of the INP parameterisation (e.g. compared to the the single-moment param. for the ice in the WSM5/WDM6 which has a more obvious effect. See additional simulation in

section 5.2)

The goal of the present paper is to use some of the WRF schemes as they are, comparing averaged quantities at 5 km resolution. We keep the paper general as to provide with a first report on the importance of choosing a relevant cloud scheme when using Polar WRF, and showing that the double-moment ice microphysics schemes simulate a LWC in better agreement with observations (and reduced radiative biases over the Larsen C ice shelf).

We plan to use the aircraft campaigns for further modelling case-studies (at higher horizontal/vertical resolution) in the future (along with the recent 2015 campaign that took place at Halley) where specific production terms will be more easily investigated by focusing on specific flights, rather than deriving/comparing averages.

Anonymous Referee #2

The authors present a modeling study of various microphysics schemes and compare model results to observations to aircraft measurements in the Antarctic. They find significant differences in predictions of the cloud thermodynamic phase and conclude that the Morrison scheme is the best scheme as it leads to the best model/observation comparison in terms of clouds and radiation. Fore The number of model studies on cloud in Antarctica are sparse, even though the region is very sensitive to changes in climate. Therefore, the current study might represent an important contribution to the literature if my comments below will be addressed. General comments

——————————————————————————————- 1) I am not convinced that the authors can indeed conclude on 'the best performing' microphysics scheme based on their model studies. They show that the Morrison scheme predicts best the super-cooled liquid in clouds, followed by the Milbrandt and Thompson schemes since all three have a sophisticated description of the various ice categories. Given the large uncertainties that are associated with the representation of clouds in general and of mixed-phase clouds in particular, the comparison to only a few parameters might not be sufficient to identify 'the best' scheme. Uncertainties in the radiation and boundary layer schemes might lead to a prediction bias such as the microphysics scheme may give the right answer for the wrong reason. I suggest a more careful wording throughout the manuscript. At several places in the manuscript, the language is rather colloquial; I listed several instances below but encourage the authors to carefully read and improve language where possible.

We rather state now that the Morrison, the Milbrand, and the Thompson, lead to simulation results that are in closer agreement to the observation than the WSM5/WDM6 schemes, which are single-moment schemes for the ice crystals. We avoid the ÂńÂăbest performingÂăÂż expression, as we agree it is not adapted.

In terms of the other physics schemes (radiation, PBL, etc.)Âă:

Please note that the physics schemes (other than the cloud scheme) we are using are the ones used in the operational forecast model AMPS (used by King et al. 2015, whose conclusion motivated the present study at 5 km resolution). We decided to discuss the microphysics schemes in the framework of the AMPS physics settings. Many choices arise when working with WRF in terms of physics parameterization – indeed - but it is very difficult to test all the parameterizations and possibilities and so using the AMPS ones seemed to be a reasonable choice. In this framework it is relevant to discuss the ability of the schemes to simulate the clouds. As for the boundary layer parameterization, we agree that it is one of the key parameterization, due to its implication in the heat/energy transfer between the surface and the troposphere. Again, we are using the one used in AMPSÂă; furthermore Deb et al. (2016) showed that the model performance at the surface is most sensitive to the choice of PBL scheme and show that the Mellor Yamada-Janjic MYJ scheme is the best performer in terms of the temperature diurnal cycle (in west Antarctica) at 5 km resolution. Hence we are confident that we are using a relevant framework to investigate the cloud schemes' behavior.

See page 5, L.7-12

———————————————————————-

2) The authors state that none of the applied microphysics schemes was specifically developed for Antarctic clouds and, therefore, their study is a first step in exploring the skills of the different schemes for such scenarios. However, I am missing a more conclusive statement on what improvements should be done in these schemes to optimize them for Antarctic clouds.

The paper concludes on what is required for a Polar WRF user in order to have a better simulation of the supercooled liquid water (a double moment parameterisation for the ice microphysics, and not a single-moment like WSM5/WDM6). We show that the ice double-moment schemes are also the one allowing for reduction of the LW bias (in addition to simulating clouds in better agreement with observations). It is of importance as WSM5 is the scheme implemented in the AMPS operational model, widely used in the antarctic community. Also, we take advantage of having ice crystals measurements to compare to the INP parameterisations to suggest a best choice for future simulations/work. However, we do now emphasize that water vapour, and temperature biases are also responsible for the failure of cloud simulation (new section 5.4), and improving the INP parameterisation is not the only thing to do.

Section 5.4Âǎp.   30 with additonal results related to sensitivity of water vapour/temperature bias: See new conclusion

———————————————————————-

3) It is mentioned that the liquid phase of clouds (drop activation) is either described based on a fixed number of droplets or it is a function of the CCN (p. 7, l. 5). Drop activation is an essential process that determines the microphysical properties of a cloud. How much difference is caused between the different microphysics schemes due to differences in the prescription of the cloud droplet number?

We have tested a different CCN concentration for the Morrison scheme (for instance,

100/cc instead of 250/cc) in four flights to see the difference, and it lead to a reduction of the liquid formation, without being conclusive for the discussion and the conclusions we want to present. We do mention another additional simulation with CCN=100 cm-3 for the Morrison scheme in section 5.2., but again the differences induced by these changes is much less important than the difference between WSM5/WDM6 and Morrison/Thompson/Milbrandt that we want to stress.

However, a complete picture addressing this would require many more simulations, in prescribing different CCN parameterizations to all the different schemes used, but this is not what the present paper is intended to present. The main aspect of the present paper is to provide with a first evaluation of the schemes as they are comparing with the (average) flight measurements, to show and discuss their impact on the surface radiative bias, and to highlight some important aspects like the need to use a double-moment ice microphysics parameterization (unlike WSM5/WDM6). It is also important to show that the current scheme used in AMPS (WSM5) has a lesser ability to model clouds in agreement with observations. Finally we discuss the biases in water vapour and temperature, and the INP parameterisation.

——————————————————————————————————- 4) The authors describe in length the differences in the results predicted by the microphysics schemes in Section 4.1. However, in order to understand the differences and to assess the skill of the various schemes, a more detailed discussion of the underlying processes is needed. Such analysis will help to identify the 'best scheme' for the right reasons and to improve existing microphysics schemes.

The present paper aims at comparing several schemes within Polar WRF (among which the one used in AMPS) in an averaged way and we decided to group the schemes in categoriesÂă: the one having a single-moment ice microphysics parameterization (WSM5/WDM6), and the one shaving a double-moment ice microphysics

paramterization (Morrison/Thompson/Milbrandt). One of the result is that the latter is best in simulation supercooled liquid. The good/poor ice microphysics simulation is directly responsible for the good/poor supercooled liquid simulation as it is explained in section 5.2Ăă:

See p. 24 L.18-27

———————————————————————————-

5) Several previous studies have highlighted the importance of ice particle shapes for ice growth and sedimentation and therefore for the partitioning between ice and liqud phases (e.g., Sulia and Harrington, JGR, 2001). How are ice particle shapes treated here? Are the predicted shapes in general agreement with observations?

The ice particle shapes are not handled in any of those schemes. Each scheme assumes a fixed shape for the ice particles, and uses a mass-diameter (m=cDd) law, which is needed in the single- or double- moment parameterisations used. For instance the Morrison scheme has c=pi*rho/6 (where rho is the particle density), and d=3Ăă: thus the scheme assumes that all particles are spheres. In the Thompson scheme, all hydrometeors are modelled as spheres, except from the snow with c=0.069 and d=2 (fractal like particles). In the Milbrandt scheme, all hydrometeors are assumed to be spherical, except for the ice crystals, which are assumed to be bullet rosettes (c=440 kg m-3 and d=3). In WSM5/WDM6 the mass-diameter equation is derived based on empirical formulas giving the terminal velocity of falling crystals. The fixed shape of the ice particles is a commonly used assumption in the derivations of moments of the size distributions and allow for explicit equations to be used for cloud microphysis' bulk parameterisation in atmospheric models. The sphericity often assumed is a crude assumption but is largely used, for simplicity. We agree that this aspect will need to be addressed in case-studies simulation.

———————————————————————————-

6) Only in Section 5, it is mentioned that the radiation scheme assumes a drop radius of cloud droplets. How does this value compare to observations and how does it affect in general the radiation prediction? Are there any studies (not necessarily in Antarctica) that have discussed biases in predicted radiation due to a constant effective radius?

Our derived value from the observation is an averaged effective radius of 7.5 um for both campaign (7.2 um for 2010, and 7.6 for 2011), so that it is close to 10 um. Also, we have rerun a simulation over the period 11th -20th Jan. 2011 to compare the new radiative bias to the former one and found no improvement in terms of bias. We believe a time-dependent effective radius should be implemented as an input to the radiative scheme, but this is out of scope of the present paper. We could not find studies discussing biases due to a constant reff. We have added some additional discussion about this in the paper, also related to the other reviewer's comment on this aspect.

————————————————————————-

7) Each microphysics scheme includes a different parameterization of ice formation. Different INP parameterizations can lead to significant differences, as it has been shown in several literature studies (e.g. Eidhammer et al., JGR, 2009). How much of the predicted differences in comparison to observations and model/model comparisons can be ascribed to the differences in INP parameterizations? Why not using the DeMott parameterization as the base case as it has been shown to perform better than the temperature-dependent-only parameterization in previous studies?

We have changed the way we present the results. We put less weight on the INP parameterisation, as we rather emphasize the main difference between WSM5/WDM6 (single-moment for the ice microphysics) and Morrison/Thompson/Milbrandt (double-moment for the ice microphysics). We do comment on the INP parameterisation as we believe it is a matter of future improvements of the cloud scheme, however we also emphasize the need of better capturing the water vapour, and temperature.

In this paper, we do not intend to investigate the detailed reason of the differences, rather to show the differences between the two groups of schemes, and explain the main reason of their different behavior (which is the single-moment paramterization for ice). Also, in shorter runs we have not observed the DeMott parameterization being enough to lead alone to a better prediction of supercooled liquid as we explain it in the new section 5.4. We also emphasize the importance of water vapour, and temperature biases.

Obviously, more simulations would be needed, certainly at higher resolution to make an even better use of the aircraft measurement, and this is what we are planning in future work.

————————————————————————————-

8) The authors should better justify their choice of a 5 km resolution. At the end of Section 5, they state that convective processes are badly resolved on this resolution; however, they fail to discuss the consequences of this caveat for simulating clouds.

\* We chose the 5 km resolution because it was the resolution used in King et al. (2015) where the authors examine the surface energy budget on the Larsen C Ice Shelf, using three models including AMPS (the Antarctic Mesoscale Prediction System, based on Polar WRF). The authors noted the very little amount of liquid formed in the AMPS model, and they suggested that the radiative biases measured were pointing towards the misrepresentation of the clouds. Also Bromwich et al. (2013) reached similar conclusion using Polar WRF at coarser resolution (15 km, and 60 km), however continent-wide. We decided to work with the higher resolution of 5 km, used in King et al. (2015).

p. 4, Line 30-31 and p.5, Line 1-2

\* We mention that ÂńÂăwarmer oceanic and sea-ice free influence of the western Peninsula and the Southern Ocean implies more convective processes (compared to the east) that are badly resolved at 5 km resolution and prevent better matching with

the aircraft observationsÂăÂż. Hence we expect convective processes to play a bigger role to the west of the Peninsula.

\* Using the convective parameterisation in the 5km resolution domain did not allow to produce better agreement with the observations in terms of clouds and clouds phase, for instanc on the specific flights mentioned in the new section 5.4 (we do not mention it, though). We believe working at 1 km horizontal resolution on case studies in future work will help better address the possible issues related to convection processes by capturing them, without having to rely on the parameterisation.

———————————————————————————

9) Throughout the paper, expression such as 'the xxx scheme forms more liquid' etc should be avoided (e.g. p. 13, l. 15). The schemes themselves do not form or produce any ice or water. Wording such as 'Using the xx scheme, it is predicted : : :' (or similar) should be used instead.

We corrected the wording, when appropriate.

Minor comments ——————————————————————————–

p. 2, l. 34: Not clear what 'the latter case' refers to. – Do you mean all three, i.e. immersion, contact or condensation freezing? If so, it might not be fully correct, since droplets may scavenge INP before ice nucleation and therefore INP and CCN are not the same.

We meant the ÂńÂăcondensation freezingÂăÂż case. We corrected this by saying ÂńÂăin the condensation caseÂăÂż. See p3 L 4

p. 7, l. 18: The beginning of the sentence is not clear. Does the AMPS model predict a liquid phase or not?

AMPS simulate clouds predominantly composed of ice with very little or even zero liqui

water. We corrected the sentence.

See p8 L3-5

p. 8, l. 10: What do you mean exactly with 'twice to four times more liquid clouds'? – Does it refer to the number of clouds or to the liquid water content of the clouds? Figures 2 and 3: In particular, in the upper panels, the dotted lines are pretty hard to distinguish.

We have corrected the wordingÂă:

* Using the Morrison scheme, twice to four times more liquid cloud mass is simulated than when using the WSM5 scheme.

* We made the dotted lines appear more clearly in Figures 2, and 3.

See p.8 L23-28

p. 11, l. 6: I think (ice) should be removed – or SWC0 added at the beginning of the sentence (?)

We edited the couple of sentences.

See p13 L7-11

p. 17, l. 5: Can an estimate be given how much the observed mass is biased due to the small and large cut-off diameters? – An extrapolation of the measured size spectrum might be better than the current complete omission.

We have approximated the crystal size distribution by exponential distributions to get some estimate of the error (bias) related to the particles that are missed above D=1.6 mm. Overall, the average relative error (bias) due to this aspect on the ice water content is about 5% in 2010, and 8% in 2011. See explanationsÂă:

See p18 L12-32 and Figure 11

Technical comments p. 1, l. 9: 'struggle' is rather colloquial p.2, l. 10; p. 3, l. 7, and

other places: Antarctic p. 3, l. 6: Southern p. 4, l. 14: discriminated p. 4, l. 26: one domain p. 5, l. 11: developed p. 6, l. 10: cloud formation p. 6, l. 25: flight measurements p. 7, l. 21: observed p. 8, l. 1: domain p. 8, l. 27: demonstrate p. 9, l. 2: tests p. 11, l. 8: Add unit to LWC (0.02) p. 12, l. 28: wind regimes p. 12, l. 33: either 'shows a: : :transect' or 'shows : : :transects p. 16, l. 19: flights p. 17, l. 14: equal p. 19, l. 27: increases p. 21, l. 9: simulation p. 21, l. 24: are consistent p. 23, l. 7: agrees p. 25, l. 3: crystal p. 26, l. 15: 'The transects in Figure 5a and b clearly show..'

These are corrected.
* * *
Deleted paragraph and figure from the previous version of the paper, about the secondary ice process Hallett-Mossop (former section 5.2, now section5.3)

ÂăAs discussed in Part 1, the distribution of ice crystals as a function of the temperature shows that below -10 C the production of ice crystals is dominated by primary ice production processes, which are represented in the cloud schemes by the INP parameterisations. However, a secondary ice production process peaking around -5 C, and identified as the Hallett-Mossop (HM) process (Hallett and Mossop, 1974) (see Part 1, section 3.2) has to be accounted for with a dedicated parameterisation in the temperature range -10 C to -3 C. Except from WSM5 and WDM6, which do not account for that process, the Milbrandt, Thompson and Morrison schemes use the same parameterisation for that mechanism (Reisner et al., 1998). It is derived using the equations of collection of water droplets by snow and graupels, multiplied by a temperature-dependent parameter based on Figure 2 of (Hallett and Mossop, 1974). Figure 11 shows the median number density of ice crystals and snow particles hydrometeors per temperature bin along all the flight tracks. The primary ice production peak which relies on the INP parameterisations clearly appears for the three schemes although with different amplitudes, that can be related to the above discussion on the various efficiencies of the INP parameterisations. In the HM temperature regime the

Thompson scheme does show a clear increase of the number density around -5 C, while the Morrison schemes does not. The Milbrandt scheme shows more or less steady and larger concentrations than the Thompson, and the Morrison schemes, but with no clear signature of the HM process. The one to two orders of magnitude difference in the predicted total number concentrations of icy condensates between the Milbrandt scheme, and the other schemes (Figure 10b) below -10 C suggests that the INP parameterisations used in the Milbrandt scheme would blur any possible signal from the HM parameterisation (the -5 C peak). The fact that the Morrison scheme does not show the HM process while the Thompson scheme does can be explained by a threshold effect. The mechanism is triggered only if the snow content is > 0.1 g kgôĂĂ1Âă and LWC > 0.5 g kgôĂĂ1 , values 5Âă which are above simulated averages for the snow and for LWC in the Morrison scheme (Figure 4c, and Figure 6). Interestingly the median number of ice and snow hydrometeors for the Thompson scheme agree within a factor of 2 with the observed median around -5 C. This shows that the Hallett-Mossop process can be to some extent correctly accounted for at this spatial resolution with the current parameterisation.

Please also note the supplement to this comment:
http://www.atmos-chem-phys-discuss.net/acp-2016-1135/acp-2016-1135-AC1-supplement.pdf

**Supplement:**

Revision of « The Microphysics of Clouds over the Antarctic Peninsula – Part 2: modelling aspects within Polar WRF »

We thank the reviewers for their comments.

This document is organised as follows :
- p.   1 (of the pdf)   Introduction to the answers to the reviewers (some general comments)
- p.   2                Answers to reviewer #1
- p.  11               Answers to reviewer #2
- p.  18               The highlighted version of the new manuscript

**Introduction to our answers** :

The motivation for this paper is the hypothesis by King et al. (2015) that radiative biases over the Larsen C ice shelf (Eastern Peninsula) were due to the lack of cloud liquid water. They worked at 5 km resolution and use three models – including AMPS (the Antarctic Mesoscale Prediction System, based on Polar WRF). Bromwich et al. (2013a) also reached this conclusion with Polar WRF at coarser resolution (15 km, and 60 km) continent-wide. However Bromwich et al. (2013a) not investigate the phase simulated by the clouds (only the cloud cover as a whole).

The main goal of the present paper is to provide with a first evaluation of the cloud schemes as they are using the (averaged) flight measurements, to show and discuss their impact on the surface radiative bias, to highlight some important aspects like the need to use double-moment ice microphysics parameterization (unlike WSM5/WDM6) to better capture supercooled cloud water, to show that the current scheme used in AMPS (WSM5) has a lesser ability to model the clouds, to discuss the biases in water vapour and temperature, and to discuss the INP parameterisation. The conclusion has been simplified to clearly emphasize those points.

Note that section 5.2 (« The ice phase parameterisation », now « The INP parameterisations ») and section 5.3 (« Simulating the clouds thermodynamic phase ») have been swaped to putt less emphasis on the effect of the INP parameterisation because it is indeed the case that we do not investigate its effect in details (although we do discuss it).

Note that in sectin 5.3 (former section 5.2), we have decided to delete the paragraph and Figure about the Hallett Mossop process as this discussion was making the section lenghty and would be more adapted to more in-depth investigation of the processes producing the ice using case studies (specific flights). The deleted text and Figure are shown at the very end of this document, as a reminder.

Additionnally a section 5.4 has been added to emphasize the role of water vapour, and temperature biases (and other model sensitivity consideration on the initialization of the model, or the number of vertical levels). In doing so we insist on the fact that changing the INP parameterisation is not enough for improving the simulation of the clouds. The previous version of the paper suggested that the INP parameterisations were the main reason of missing liquid cloud in the simulation, and we wanted to correct this.

We have simplifed the conclusion to highlight more clearly the main aspects of this work, while we now insist on the need for further work on improving/investigating the temperature/water vapour biases as well. (The abstract has been amended, accordingly.)

Please note that : - Edited/new text is highlighted in the paper. A highlighted section title means that the section is new (5.4) or fully changed/reformulated (conclusion).

                    - The reviewers comments below are in black. Our answers are in blue color. Below each answer, the page, and line numbers of the related edited/new text of the paper are indicated.

**Anonymous Referee #1**
* * *
This paper presents comparisons of simulations of clouds over the Antarctic Peninsula
using different cloud microphysical schemes in WRF, and includes comparisons against
observations described in Part 1 of the manuscript. The main finding of Part 2 is that,
regardless of which scheme is used for the simulations, a large misrepresentation of
the cloud thermodynamic phase explains a lot of the large radiative biases derived at
the poles continent wide (i.e., all schemes fail at predicting as much supercooled liquid
mass as seen in observations). The authors conclude that a parameterization scheme
for ice nucleation that depends on both temperature and aerosol content should be
implemented in WRF to get a more realistic representation of primary ice production,
and hence a better representation of supercooled liquid phase in Antarctic clouds.

We have emphasized that the main finding of the paper is that changing from the AMPS scheme WSM5
to other schemes does reduce the radiative bias and some schemes substantially reduce the bias on the
Larsen C ice shelf. We feel that this makes the paper clearer. In particular we find that double moment
schemes (Morrison, Thompson, Milbrandt) perform better than the single moment scheme that is used
as standard in AMPS, to simulate liquid cloud. We feel that this is an important finding as it is easy to
implement this simple change within operation model while more detailed changes such as
implementing new INP parameterisation would require more invesigation first. Moreover we also stress
the fact that working on the water vapour/temperature biases is also a mandatory step.
* * *
The study should be published because unique modeling simulations are compared
with a unique set of data over the Antarctic Peninsula. However, the paper can be
improved in a number of ways. It is lacking in that it only focus on sensitivities to
cloud microphysics, ignoring impacts from other parameterization schemes that could
also be affecting the simulations and comparison with observed quantities. The uncertainties
associated with the representation of the microphysics needs to be better
placed in context of uncertainties associated with the representation of other processes
such as boundary layer parameterization schemes. Otherwise, it is possible that the
sought after agreement between models and observations may occur due to reasons
not associated with the representation of microphysics (i.e., are the right answers being
obtained for the wrong reasons). I'm not convinced that simulations that do not
consider uncertainties due to other processes can truly assess the ability of the microphysics
schemes to model realistic clouds across the Antarctic Peninsula (page 5,
line 10) in the absence of these other sensitivities. This is especially true given the
statement that none of the microphysics schemes adequately predict the supercooled
water: maybe some other process other than microphysics is causing this problem.

We have concentrated on the cloud physics schemes as we felt that this scheme was most likely to be
the one that was causing the errors seen by King et al (2015), and because no study focused on Antarctic
clouds have investigated this before. Apart from the cloud scheme we have not altered any other physics
schemes as although they can be responsible for large errors we feel that many have already been
investigated eg : the boundary layer parameterization, although it is one of the key parameterization,
due to its implication in the heat/energy transfer between the surface and the troposphere. We are using
the one used in the operational forecast model AMPS ; furthermore Deb et al. (2016) showed that the
model performance at the surface is most sensitive to the choice of PBL scheme and show that the
Mellor Yamada-Janjic (MYJ) scheme is the best performer in terms of the temperature diurnal cycle (in
west Antarctica) at 5 km resolution. Hence we are confident that we are using a relevant framework to
investigate the cloud schemes' behavior.

See page 5, L.7-12
* * *
For example, I think that the amount of fetch off the ocean and its representation would
be very important.

It is possible that larger scale dynamical effects can be affecting the clouds but on the whole the wind
field is well forecast by WRF (for instance, wind direction measured at the flights/station is well
captured by the model as recalled at the beginning of section 4.2) and so it is unlikely that errors in the
fetch over the open ocean are the problem.
* * *
The sensitivity studies in the paper emphasize how different schemes affect the modeled
cloud parameters. But, in addition to sensitivity in choice of scheme, there can
also be sensitivity to some of the parameters (i.e., constants) that are assumed within
an individual scheme. Was any effort made to look at the sensitivity to constants within
a scheme?

There are many constants in the cloud schemes but we were concerned in this study with difference
between the schemes – if we had altered constants in every scheme we would have had to do many more
runs and this was not feasible we the resources we had available. In this paper we focus on more general
issues like the performance of the schemes as they are implemented, and the fact that the schemes are
single/double moment for the ice, the water vapour/temperature biases, and the INP parameterisation
(in the discussion) and this allows a managable number of runs. However we do show some new results
discussing the change of the INP parameterisation :

Also in the new section 5.4, page 30.
* * *
Another weakness of the paper is that the paper does a reasonable job in describing
the differences between the simulations and the simulations that are most consistent
with the observations (note, I think most consistent with the observations is a better way
of wording it rather than saying the best simulation). However, I a paper in ACP should
do a better job in showing why these differences between parameterization schemes
are occurring. This can be done by looking at various prognostic terms in the model
(i.e., production rates due to various microphysical processes). This could explain
why different schemes produce different modeled fields rather than just stating that
different schemes produce different cloud fields; the result that different microphysical
parameterization schemes gives different modeled fields is not an incredibly new or
exciting result in that it has been previously demonstrated in a wide number of other
meteorological conditions.

Although we appreciate the reviewer's point this study is intended to give a general picture of the cloud
simulation performances of different schemes over the Antarctic Peninsula, and as this has never carried
out before we did not feel it nessecary to start investigating the production terms of all relevant processes
of the different schemes. A major difference as we show it is the fact of having single-moment ice
microphysics (and the related dependency between the number of crystals, and the ice water content)
(WSM5/WDM6) versus double-moment ice microphysics (Morrison, Thompson, and Milbrandt). Some
additional simulation results have been carried out and are presented (section 5.2, and new section 5.4)
or just reported (section 5.1) to improve the discussion.

See p. 24 Line 18-26, and the related new Figure 11, in section 5.2

See p. 30 section 5.4, and the related new Figure 15

These different additions are further mentioned in the answers below.
* * *
The paper attributes a lot of the disagreement between the modeled and observed
fields to the performance of the ice nucleation parameterization: namely, it is state that
the ice nucleation parameterization is a major reason why supercooled water is underestimated.
For example, it is stated that forcing a dependence on aerosol amount and
temperature, rather than just temperature, could give better agreement. Can more be
done with adjusting constants in the ice nucleation schemes within the different cloud
parameterizations to better demonstrate this? This way it would be easier to define
how much of the difference is due to the different microphysical schemes, and how
much is due to the microphysical parameterization schemes (rather than just stating
that different parameterization schemes use different representations of INPs that may
be causing some of the differences).

We feel that we may have been missunderstood and so have changed the way the results are presenting
in the paper. We have now put less emphasis ont the importance of INP parameterisation as the main
cause of the differences between WSM5/WDM6, and Thompson/Milbrandt/Morrison, is the fact that
the former rely on singe-moment ice microphysics with a parameterisation linking Ni (number of
crystals) and qi (mass of the ice phase), that lead to enhanced production of ice.

See p. 24 Line 18-26, and the related new Figure 11, in section 5.2

To this respect, the choice of the INP appears to be secondary in comparison whether the scheme is
single or double moment. Also we stress the need of reducing the water vapour/temperature biases to
help improving the cloud simulation but this is probably outside the scope of the present study.

See p. 30 section 5.4, and the related new Figure 15

As such these are interesting results to Polar WRF/AMPS users.
* * *
The authors conclude that the Morrison scheme is the best performing scheme. It
would have been beneficial to show some production and depletion rates of processes
producing various hydrometeor categories (especially supercooled water) so that there
could be a focus on specific terms and processes that are well represented in that
scheme, and hence to better determine where the supercooled water is coming from.
That might also help explain why the Morrison scheme is the better performing scheme.

In this study we have concentrated on the average performance of the schemes over many different cases
– those that were reported in part one of this study – rather than look in detail at a particular case. We
think that the detailed study of production/depletion rates is best left to future work focusing on higher
resolution simulation and case-studies (specific flights) and rather than the mean values we use here.
These average results show that the double moment microphysics schemes (Morrison, Thompso,
Milbrandt) perform better than (WSM5, WDM6) in modelling the liquid phase. We find that the
Morrison scheme is performing better but no longer claim that it is the « best performing scheme » as
we agree more detailed study of the processes will be needed.
* * *
The authors use the Brown and Francis (1995) mass-dimension parameterization to derive the ice water content. However, this is a very crude parameterization whose use may be inappropriate depending on the mixtures of the shapes and sizes of crystals present. What types of habits were noted in the CIP images? Are they consistent with those assumed in the Brown and Francis parameterization? How sensitive are the calculated IWCs to the assumed parameterization? Does this assumption affect any of the qualitative conclusions?

Although there are more recent mass diameter laws that we could have consdered using Brown and Francis was the only scheme available within our image processing software (OASIS) and considering the errors we would expect anyway from our low resolution CIP probe. It also meant that our results were directly comparable with other results from polar studies who had used the same method (for example Lloyd et al (2011).
* * *
The smallest domain used in the model simulations is 5 km. This seems quite coarse for a study looking at the impacts of microphysical processes. What vertical resolution is used? Does the choice of horizontal and vertical resolution have any effect on the most important findings of this study? In addition, how sensitive were the simulated fields to the choice of initial conditions (e.g., either the product used or the time at which the simulations were initialized)?

We answer the various points below :

* It is a similar horizontal resolution to previous work (ie King et al. 2015) that we wish compare this study too. It is also a higher resolution to Bromwich et al. 2013a (15-60 km resolution), who investigated radiative biases over the continent using Polar WRF. We have completed many runs for this study – we have been looking at more than 1 month for each scheme, each year (2010, 2011) during which many observational flights were completed. This has meant that resources would not have allowed a significantly higher horizontal resolution. (However, we can say that other unpublished/preliminary simulations comparing WSM5 and the Morrison scheme at slightly higher resolution (2 km), and doubling the number of vertical eta levels (60, instead of 30), over the Weddell sea, and related to our 2015 campaign do lead to the same conclusions presented in the manuscript : that the WSM5 scheme underpredicts significantly more the supercooled liquid water, compared to the Morrison scheme, that the radiative biases are significantly reduced using the Morrison scheme, that the water vapour/temperature biases are also responsible for disagreement between the model and the cloud measurements). (For the vertical resolution, see further below).

* Bromwich et al. (2013a) showed that using ERA-Interim reanalysis products for initial, and boundary conditions produces the best skills within Polar WRF, and thus we decided to use ERA-Interim data.

See p.6 , L. 1-2

* 30 (eta) levels are used, with the levels being closer to each other closer to the surface. See Table 1, and explanation when deriving the averaged vertical LWC profiles (p17. L4-7) ; the maximum distance between model vertical levels is approximately 500m below 4500 meters, with an average distance of approximately 350m, and down to 10-50 m above the surface).

See p. 17 L4-7

 * A doubling of the number of levels does allow to reduce the biases of water vapor and temperature in some places, but it should not affect our general result, as a significant improvement in the cloud simulation is not observed. The initialisation closer to the date of observation reduces the water vapour bias (mainly), but it does not necessarily need to better cloud simulation, and the general picture we

provide here when comparing the schemes will not be altered. A discussion with small sensitivity test on specific flights was added to raise those concerns, as we agree they must be discussed to some extent. Hence, the additional section 5.4.

See new section 5.4 p. 30.
* * *
I am also worried about some of the inconsistencies between the microphysics and radiative parameterization schemes that seem to exist in the paper. For example, radiative schemes may make specific assumptions about the shape distributions of various hydrometeor categories. Are these assumptions consistent with what is assumed in the microphysical schemes themselves? Otherwise, an inconsistency between the radiative and microphysical parameterizations could be responsible for some of the discrepancies in the radiative parameterization schemes. Further, the authors themselves seem to state that the assumption of a constant effective radius of 10 micrometers in the radiative scheme may not be consistent with what the microphysical parameterization scheme is assuming.

We understand complitly that radiation scheme may not be consitant with the cloud physics scheme and we did consider changing both at the same time. However we felt that it would be better to only change one scheme at a time. More generally, those inconsistencies mentioned by the reviewer are part of the drawbacks in using a flexible model like WRF where different parameterizations can be used in different combinations.
However, and importantly, the approach used in the SW radiative scheme (where reff is held constant) is to parameterize the optical depth for water/ice cloud as a function of reff based on the fact that (A solar radiation parameterization for atmospheric studies.NASA Tech. Rep. NASA/TM-1999-10460,vol.15,38 pp) : « Theoretical considerations and radiative transfer calculations have shown that cloud single-scattering properties are not significantly affected by details of the particle size distribution and can be adequately parameterized as functions of the effective particle size »
(Fu, 1996; Hu and Stamnes, 1993; Tsay et al., 1989). We appreciate that this may be a strong assumption, especially that the width of the size distribution should impact the radiative properties of the cloud (but then the width also impact the effective radius value). Nevertheless, as the parameterization used does not assume any type of size distribution, we think that it can be used with the different cloud schemes despite their own different assumptions on the hydrometeors size distributions.

See p. 23 L. 11-14

Also, we derived an averaged reff for the droplets in the measurements of about 7.6 um in 2010, and 7.2 um in 2011, close to the 10 um assumed (constant) in the SW radiative scheme. We have ran the model over the period 11th-20th 2011 setting reff=7um and observed no consistently better radiative bias. Having this reff constant throught the simulation is certainly problematic (more than the fact of having a fixed value either to 7 or to 10 um) and could be one of the reason of larger shortwave biases. The LW bias is more sensitive to the ice/liquid content, hence is systematically reduced (to the east of the Peninsula) with cloud schemes simulating more liquid clouds (Morrison, Thompson, Milbrandt).

See p. 23 L. 17-21

SPECIFIC COMMENTS
* * *
Page 4, line 13: What diameter is the distinct peak located at? Also, recommend using size distribution instead of spectra. Spectra typically refers to radiative quantities.

It is for a diameter between 8 and 12 um.

See p. 4 L 19-20

and we have replaced spectrum by size distribution.

Page 5, line 14, recommend "graupel is" rather than "graupel are"

We have corrected that.

Page 8, line 9-10: Degree of agreement is always a very relative term. I think when talking about whether simulations agree or disagree, the writing should be more quantitative stating what the degree of agreement is.

We made the writing more quantitative.

See p 8 L. 24-29

Page 8, line 19: How much graupel was in the observations? Can you give some indication of how much agreement there is between modeled/observed graupel?

We don't think we can resolve graupel properly in our observations. It is difficult to tell the difference between an irregular ice crystal and a heavily rimed one (graupel) by looking at the shadow images.

Page 9, line 16: Is the radiative scheme consistent with each of the microphysical parameterization schemes?

Please refer to our answer to the last major comment above, and related to this specific point.

Page 10, line 4: Shouldn't the radiative schemes be consistent with each of the microphysical parameterization schemes rather than just the K15 scheme?

Please refer to our answer to the related major comments above.

Page 12, line 8: It would be nice to know the specific terms that are producing the supercooled liquid mass content in the model.

We did not perform simulations allowing to target specifically the production terms, as it is not the goal of this paper. However, the production of supercooled liquid results from the supersaturation remaining after the ice has condensed. The process is condensational growth and is depending on how efficient the ice formation is. In particular, in this case, we show that the double-moment schemes for ice are performing better than the single-moment schemes for ice WSM5/WDM6 (which are single- and double- moment for the liquid phase, respectively). The latter have Nice and IWC not evolving independently (single-moment), and the relationship used to link both leads to high values of IWC. We now explain this in section 5.2, particularly :

See p. 24 L19-27

Line 12, line 22: If two schemes use the INP parameterizations, is it fair to say that the microphysical parameterizations are causing the differences, or should it just be attributed to the INP parameterization. This sentence is clear, but I am wondering how this affects some of the preceding discussions in the paper.

To avoid confusion, we removed this sentence, as it anticipates too much on what corresponds to the discussion part, and since – as stated in our introduction to this document – we decided  to put let emphasis on the INP parameterisation, when revising our work.

Page 13, line 19: If the effects of the mountains are variable/valuable, what are the prognosed values that are affecting the generation of the LWC? This might be especially good to look at given that quantities east and west of the mountain range are being compared.

Note that we compare quantities to the east and to the west of the Peninsula because the statistics of the observations were much better on each side as the aircraft tended to avoid clouds when crossing the mountains. Also the data have been grouped in East/West « bins » to improve the statistics when comparing averaged quantities (as in Part 1).

The supercooled LWC will be generated maintly depending on the supersaturation removal by the ice microphysics processes. This is shown for instance by the additional simulation using WSM5 scheme with a modified relationship between qice (mass) and nice (number), where the factor linking qice to nice is divided by 100 in the empirical relationship used for WSM5's ice crystals. This edit makes WSM5 generating supercooled LWC (say comparable to Morrison). In comparison, changing the INP parameterisation does not increase the supercooled liquid simulated with WSM5 across the Peninsula. As presented above, this is now emphasize in the discussion part section 5.2.

Page 14, line 7: the statement that the schemes perform worst on the West compared to the East is interesting. But, the paper would be much more insightful if it could identify the processes that are at work so that the schemes are performing better on the West compared to on the East.

It is actually hard to say.

The average temperature bias is larger to the west compared to the east, and now it is clearly recalled in the discussion, as well as the topography which is completely different to the west and would require higher resolution simulation in future work.

and

Page 15, line 4: There is an overemphasis on what is the best scheme. You might be getting the right answer for the wrong reasons. You should also look at what are some of the prognosed terms allowing this scheme to perform better, and how the sensitivity to microphysical parameterizations is affected by sensitivity to other parameterization schemes.

We agree that we emphasized too much on what the best scheme could be and we changed that in the paper (as already stated above). As for the second part of the comment, we rather state that the double-moment ice schemes is more important than a double-moment liquid scheme to improve the supercooled liquid water simulation, and has to be preferred to a single-moment ice scheme. In terms of other parameterization schemes we explain above that choosing the AMPS (Antarctic operational model) configuration is a relevant choice, and mention Deb et al. (2016)'s paper as for validating the most relevant PBL scheme in an Antarctic environment.

Page 17, line 9: How much of the missing mass might not be detected? I think a simple estimate of this can be made if you extend the mass distributions to larger sizes (simple exponential or lognormal fit) to estimate how much of the mass that you are missing.

We have approximated the crystal size distribution by exponential distributions to get some estimate of the error related to the particles that are missed above D=1.5 mm. Overall, the average relative error on the water ice content is about 5% in 2010, and 8% in 2011. We also introduced a new Figure 8 to show this.

See P18 L.12-32, and the related new Figure 8.

Page 19, line 1: How much of a difference is important? What difference is acceptable in terms of being a good match with observations?

In this sentence we compare the averaged value of the mixed-phase ratio, and its variability to the observed quantities, and state which scheme produces the closest agreement to the observations. Here, the best match is considered as the one which gives the closest value of the so-called mixed-phase ratio to the observed one. (but we might be missing your point, here.)

This paragraph is now p. 19 L17-22.

Page 19, line 7: This paragraph has a very good description of errors associated with these measurements. It would be nice if a similar comprehensive discussion of the uncertainties and errors associated with the microphysical measurements were included, especially given that microphysics is the focus of this study.

The errors associated with the microphysical measurements and the instruments are considered in part one of this study (Lachlan-Cope et al. 2016, ACP). For instance we explained that the « the CAS data for most flights agrees within 15% with the Hotwire LWC Sensor – part of this discrepancy is attributed to the Hotwire LWC Sensor's tendency to underread at high values of LWC. The CIP images particles between a diameter of 25 $\mu$ m and 1.5 mm, with 25 $\mu$ m pixel resolution, and had not at the time of this campaign been fitted with anti-shatter tips. However, a study of the particle inter-arrival times indicated very few shattered particle, and these were removed by eliminating particles that arrive within 1 $\mu$ s. » (Part 1, Lachlan-Cope et al. 2016, section 2.1)

In the present paper :

- the errors (or bias) we can speak of, related to the LWC, are explained in the section 4.3.1 (first paragraph) already and are related to the inability to identify correctly the entire population of droplet in the large size-range (small size range of the CIP/crystals). Errors on the CAS measurements itself as such are in comparison negligible. Also, the hotwire probe (bulk measurements) backs up the LWC derived from the Cloud Aerosol Spectrometer (it is said in part I, although we do not recall this here).

See p14 L. 1-2 and p15 L1-9

- As explained above, we now present some errors for the IWC in section 4.3.2 (Ice phase) related to the icy particles not detected above D=1.5 mm (using exponential distributions to extrapolate the number of crystals), that lead to an underestimation of the real ice water content.

See p18 L 7-26

Page 21, line 12: Should you state that WSM5 and WDM6 should not be used for these studies when the other boundary layer/other parameterization schemes are being used? This conclusion might not apply if some other boundary layer parameterization schemes are being used.

We think we can state that given our settings WSM5/WDM6 which rely on the same ice microphysics parameterization (the fact that WDM6 has also graupel does not change the results since graupel are

very few and almost always absent in the simulation output) should not be used because they perform less well in terms of supercooled water simulation. Since we explain that the ice microphysics part of the cloud parameterization is responsible for this, we don't think using another PBL would lead to different result. Also, the PBL scheme we are using was shown by Deb et al. (2016) to perform the best and is used in AMPS, and as such is a relevant framework to work with.

Page 21, line 17: Is the assumption of a constant 10 micrometer effective radius consistent with the assumptions that are made in the microphysical parameterization schemes? See earlier comment about consistency between microphysical and radiative schemes.

Please refer to our answer in the major comments.

Page 21, lines 29-30: If the INP parameterization is so important, why diagnose it only from temperature if an alternate parameterization is available? It would seem that this could be a greater focus of the study if the INP parameterization is so important.

Although we have decided to keep some discussion about the INP parameterization in the paper - because we believe such report can motivate further work - we toned down the statement that INP are mostly responsible for the differences between schemes. Indeed, for our work, the most important factor is the ice microphysics single-moment param. of WSM5/WDM6 (See page 24 L14-24) and local vapour/temperature biases that can lead to missing clouds in the model (see new section 5.4)

Page 22, paragraph beginning line 22: Can you show how much specific terms are contributing to the ice production processes rather than having generic descriptions of these processes?

We have reorganised the discussion and made it insist less on the possible importance of the INP parameterisation (e.g. compared to the the single-moment param. for the ice in the WSM5/WDM6 which has a more obvious effect. See additional simulation in section 5.2)

The goal of the present paper is to use some of the WRF schemes as they are, comparing averaged quantities at 5 km resolution. We keep the paper general as to provide with a first report on the importance of choosing a relevant cloud scheme when using Polar WRF, and showing that the double-moment ice microphysics schemes simulate a LWC in better agreement with observations (and reduced radiative biases over the Larsen C ice shelf).

We plan to use the aircraft campaigns for further modelling case-studies (at higher horizontal/vertical resolution) in the future (along with the recent 2015 campaign that took place at Halley) where specific production terms will be more easily investigated by focusing on specific flights, rather than deriving/comparing averages.

**Anonymous Referee #2**

The authors present a modeling study of various microphysics schemes and compare model results to observations to aircraft measurements in the Antarctic. They find significant differences in predictions of the cloud thermodynamic phase and conclude that the Morrison scheme is the best scheme as it leads to the best model/observation comparison in terms of clouds and radiation. Fore The number of model studies on cloud in Antarctica are sparse, even though the region is very sensitive to changes in climate. Therefore, the current study might represent an important contribution to the literature if my comments below will be addressed.
General comments
* * *
1) I am not convinced that the authors can indeed conclude on 'the best performing' microphysics scheme based on their model studies. They show that the Morrison scheme predicts best the super-cooled liquid in clouds, followed by the Milbrandt and Thompson schemes since all three have a sophisticated description of the various ice categories. Given the large uncertainties that are associated with the representation of clouds in general and of mixed-phase clouds in particular, the comparison to only a few parameters might not be sufficient to identify 'the best' scheme. Uncertainties in the radiation and boundary layer schemes might lead to a prediction bias such as the microphysics scheme may give the right answer for the wrong reason. I suggest a more careful wording throughout the manuscript. At several places in the manuscript, the language is rather colloquial; I listed several instances below but encourage the authors to carefully read and improve language where possible.

We rather state now that the Morrison, the Milbrand, and the Thompson, lead to simulation results that are in closer agreement to the observation than the WSM5/WDM6 schemes, which are single-moment schemes for the ice crystals. We avoid the « best performing » expression, as we agree it is not adapted.

In terms of the other physics schemes (radiation, PBL, etc.) :

Please note that the physics schemes (other than the cloud scheme) we are using are the ones used in the operational forecast model AMPS (used by King et al. 2015, whose conclusion motivated the present study at 5 km resolution). We decided to discuss the microphysics schemes in the framework of the AMPS physics settings. Many choices arise when working with WRF in terms of physics parameterization – indeed - but it is very difficult to test all the parameterizations and possibilities and so using the AMPS ones seemed to be a reasonable choice. In this framework it is relevant to discuss the ability of the schemes to simulate the clouds. As for the boundary layer parameterization, we agree that it is one of the key parameterization, due to its implication in the heat/energy transfer between the surface and the troposphere. Again, we are using the one used in AMPS ; furthermore Deb et al. (2016) showed that the model performance at the surface is most sensitive to the choice of PBL scheme and show that the Mellor Yamada-Janjic
 MYJ scheme is the best performer in terms of the temperature diurnal cycle (in west Antarctica) at 5 km resolution. Hence we are confident that we are using a relevant framework to investigate the cloud schemes' behavior.

See page 5, L.7-12
* * *
2) The authors state that none of the applied microphysics schemes was specifically developed for Antarctic clouds and, therefore, their study is a first step in exploring the skills of the different schemes for such scenarios. However, I am missing a more conclusive

statement on what improvements should be done in these schemes to optimize
them for Antarctic clouds.

The paper concludes on what is required for a Polar WRF user in order to have a better simulation of the supercooled liquid water (a double moment parameterisation for the ice microphysics, and not a single-moment like WSM5/WDM6). We show that the ice double-moment schemes are also the one allowing for reduction of the LW bias (in addition to simulating clouds in better agreement with observations). It is of importance as WSM5 is the scheme implemented in the AMPS operational model, widely used in the antarctic community. Also, we take advantage of having ice crystals measurements to compare to the INP parameterisations to suggest a best choice for future simulations/work. However, we do now emphasize that water vapour, and temperature biases are also responsible for the failure of cloud simulation (new section 5.4), and improving the INP parameterisation is not the only thing to do.

Section 5.4 p. 30 with additonal results related to sensitivity of water vapour/temperature bias:
See new conclusion
* * *
3) It is mentioned that the liquid phase of clouds (drop activation) is either described based on a fixed number of droplets or it is a function of the CCN (p. 7, l. 5). Drop activation is an essential process that determines the microphysical properties of a cloud. How much difference is caused between the different microphysics schemes due to differences in the prescription of the cloud droplet number?

We have tested a different CCN concentration for the Morrison scheme (for instance, 100/cc instead of 250/cc) in four flights to see the difference, and it lead to a reduction of the liquid formation, without being conclusive for the discussion and the conclusions we want to present. We do mention another additional simulation with CCN=100 cm-3 for the Morrison scheme in section 5.2., but again the differences induced by these changes is much less important than the difference between WSM5/WDM6 and Morrison/Thompson/Milbrandt that we want to stress.

See p. 24 L. 28-33

However, a complete picture addressing this would require many more simulations, in prescribing different CCN parameterizations to all the different schemes used, but this is not what the present paper is intended to present. The main aspect of the present paper is to provide with a first evaluation of the schemes as they are comparing with the (average) flight measurements, to show and discuss their impact on the surface radiative bias, and to highlight some important aspects like the need to use a double-moment ice microphysics parameterization (unlike WSM5/WDM6). It is also important to show that the current scheme used in AMPS (WSM5) has a lesser ability to model clouds in agreement with observations. Finally we discuss the biases in water vapour and temperature, and the INP parameterisation.
* * *
4) The authors describe in length the differences in the results predicted by the microphysics
schemes in Section 4.1. However, in order to understand the differences and
to assess the skill of the various schemes, a more detailed discussion of the underlying
processes is needed. Such analysis will help to identify the 'best scheme' for the right
reasons and to improve existing microphysics schemes.

The present paper aims at comparing several schemes within Polar WRF (among which the one used in AMPS) in an averaged way and we decided to group the schemes in categories : the one having a single-moment ice microphysics parameterization (WSM5/WDM6), and the one shaving a double-moment ice microphysics paramterization (Morrison/Thompson/Milbrandt). One of the result is that the latter is best in simulation supercooled liquid. The good/poor ice microphysics simulation is directly responsible for the good/poor supercooled liquid simulation as it is explained in section 5.2 :

See p. 24 L.18-27
* * *
5) Several previous studies have highlighted the importance of ice particle shapes for
ice growth and sedimentation and therefore for the partitioning between ice and liqud
phases (e.g., Sulia and Harrington, JGR, 2001). How are ice particle shapes treated
here? Are the predicted shapes in general agreement with observations?

The ice particle shapes are not handled in any of those schemes. Each scheme assumes a fixed shape for the ice particles, and uses a mass-diameter ($m=cD^d$) law, which is needed in the single- or double-moment parameterisations used. For instance the Morrison scheme has $c=pi*rho/6$ (where rho is the particle density), and $d=3$ : thus the scheme assumes that all particles are spheres. In the Thompson scheme, all hydrometeors are modelled as spheres, except from the snow with $c=0.069$ and $d=2$ (fractal like particles). In the Milbrandt scheme, all hydrometeors are assumed to be spherical, except for the ice crystals, which are assumed to be bullet rosettes ($c=440$ kg m$^{-3}$ and $d=3$). In WSM5/WDM6 the mass-diameter equation is derived based on empirical formulas giving the terminal velocity of falling crystals. The fixed shape of the ice particles is a commonly used assumption in the derivations of moments of the size distributions and allow for explicit equations to be used for cloud microphysis' bulk parameterisation in atmospheric models. The sphericity often assumed is a crude assumption but is largely used, for simplicity. We agree that this aspect will need to be addressed in case-studies simulation.
* * *
6) Only in Section 5, it is mentioned that the radiation scheme assumes a drop radius of
cloud droplets. How does this value compare to observations and how does it affect in
general the radiation prediction? Are there any studies (not necessarily in Antarctica)
that have discussed biases in predicted radiation due to a constant effective radius?

Our derived value from the observation is an averaged effective radius of 7.5 um for both campaign (7.2 um for 2010, and 7.6 for 2011), so that it is close to 10 um. Also, we have rerun a simulation over the period 11th -20th Jan. 2011 to compare the new radiative bias to the former one and found no improvement in terms of bias. We believe a time-dependent effective radius should be implemented as an input to the radiative scheme, but this is out of scope of the present paper. We could not find studies discussing biases due to a constant reff. We have added some additional discussion about this in the paper, also related to the other reviewer's comment on this aspect.

See p. 23 L 11-21
* * *
7) Each microphysics scheme includes a different parameterization of ice formation.
Different INP parameterizations can lead to significant differences, as it has been
shown in several literature studies (e.g. Eidhammer et al., JGR, 2009). How much
of the predicted differences in comparison to observations and model/model comparisons
can be ascribed to the differences in INP parameterizations? Why not using the
DeMott parameterization as the base case as it has been shown to perform better than
the temperature-dependent-only parameterization in previous studies?

We have changed the way we present the results. We put less weight on the INP parameterisation, as
we rather emphasize the main difference between WSM5/WDM6 (single-moment for the ice
microphysics) and Morrison/Thompson/Milbrandt (double-moment for the ice microphysics). We do
comment on the INP parameterisation as we believe it is a matter of future improvements of the cloud
scheme, however we also emphasize the need of better capturing the water vapour, and temperature.

In this paper, we do not intend to investigate the detailed reason of the differences, rather to show the
differences between the two groups of schemes, and explain the main reason of their different behavior
(which is the single-moment paramterization for ice). Also, in shorter runs we have not observed the
DeMott parameterization being enough to lead alone to a better prediction of supercooled liquid as we
explain it in the new section 5.4. We also emphasize the importance of water vapour, and temperature
biases.

Obviously, more simulations would be needed, certainly at higher resolution to make an even better use
of the aircraft measurement, and this is what we are planning in future work.
* * *
8) The authors should better justify their choice of a 5 km resolution. At the end of
Section 5, they state that convective processes are badly resolved on this resolution;
however, they fail to discuss the consequences of this caveat for simulating clouds.

* We chose the 5 km resolution because it was the resolution used in King et al. (2015) where the authors
examine the surface energy budget on the Larsen C Ice Shelf, using three models including AMPS
(the Antarctic Mesoscale Prediction System, based on Polar WRF). The authors noted the very little
amount of liquid formed in the AMPS model, and they suggested that the radiative biases measured
were pointing towards the misrepresentation of the clouds. Also Bromwich et al. (2013) reached similar
conclusion using Polar WRF at coarser resolution (15 km, and 60 km), however continent-wide. We
decided to work with the higher resolution of 5 km, used in King et al. (2015).

p. 4, Line 30-31 and p.5, Line 1-2

* We mention that « warmer oceanic and sea-ice free influence of the western Peninsula and the
Southern Ocean implies more convective processes (compared to the east) that are badly resolved at 5
km resolution and prevent better matching with the aircraft observations ». Hence we expect convective
processes to play a bigger role to the west of the Peninsula.

See p.26 L.30-32

* Using the convective parameterisation in the 5km resolution domain did not allow to produce better
agreement with the observations in terms of clouds and clouds phase, for instanc on the specific flights

mentioned in the new section 5.4 (we do not mention it, though). We believe working at 1 km horizontal resolution on case studies in future work will help better address the possible issues related to convection processes by capturing them, without having to rely on the parameterisation.
* * *
9) Throughout the paper, expression such as 'the xxx scheme forms more liquid' etc should be avoided (e.g. p. 13, l. 15). The schemes themselves do not form or produce any ice or water. Wording such as 'Using the xx scheme, it is predicted : : :' (or similar) should be used instead.

We corrected the wording, when appropriate.

Minor comments
* * *
p. 2, l. 34: Not clear what 'the latter case' refers to. – Do you mean all three, i.e. immersion, contact or condensation freezing? If so, it might not be fully correct, since droplets may scavenge INP before ice nucleation and therefore INP and CCN are not the same.

We meant the « condensation freezing » case. We corrected this by saying « in the condensation case ». See p3 L 4

p. 7, l. 18: The beginning of the sentence is not clear. Does the AMPS model predict a liquid phase or not?

AMPS simulate clouds predominantly composed of ice with very little or even zero liqui water. We corrected the sentence.

See p8 L3-5

p. 8, l. 10: What do you mean exactly with 'twice to four times more liquid clouds'? – Does it refer to the number of clouds or to the liquid water content of the clouds? Figures 2 and 3: In particular, in the upper panels, the dotted lines are pretty hard to distinguish.

We have corrected the wording :

* Using the Morrison scheme, twice to four times more liquid cloud mass is simulated than when using the WSM5 scheme.

* We made the dotted lines appear more clearly in Figures 2, and 3.

See p.8 L23-28

p. 11, l. 6: I think (ice) should be removed – or SWC0 added at the beginning of the sentence (?)

We edited the couple of sentences.

See p13 L7-11

p. 17, l. 5: Can an estimate be given how much the observed mass is biased due to the small and large cut-off diameters? – An extrapolation of the measured size spectrum

might be better than the current complete omission.

We have approximated the crystal size distribution by exponential distributions to get some estimate of the error (bias) related to the particles that are missed above D=1.6 mm. Overall, the average relative error (bias) due to this aspect on the ice water content is about 5% in 2010, and 8% in 2011. See explanations :

See p18 L12-32 and Figure 11

Technical comments
p. 1, l. 9: 'struggle' is rather colloquial
p.2, l. 10; p. 3, l. 7, and other places: Antarctic
p. 3, l. 6: Southern
p. 4, l. 14: discriminated
p. 4, l. 26: one domain
p. 5, l. 11: developed
p. 6, l. 10: cloud formation
p. 6, l. 25: flight measurements
p. 7, l. 21: observed
p. 8, l. 1: domain
p. 8, l. 27: demonstrate
p. 9, l. 2: tests
p. 11, l. 8: Add unit to LWC (0.02)
p. 12, l. 28: wind regimes
p. 12, l. 33: either 'shows a: : :transect' or 'shows : : :transects
p. 16, l. 19: flights
p. 17, l. 14: equal
p. 19, l. 27: increases
p. 21, l. 9: simulation
p. 21, l. 24: are consistent
p. 23, l. 7: agrees
p. 25, l. 3: crystal
p. 26, l. 15: 'The transects in Figure 5a and b clearly show..'

These are corrected.
* * *
**Deleted paragraph and figure from the previous version of the paper, about the secondary ice process Hallett-Mossop (former section 5.2, now section5.3)**

As discussed in Part 1, the distribution of ice crystals as a function of the temperature shows that below -10 C the production of ice crystals is dominated by primary ice production processes, which are represented in the cloud schemes by the INP parameterisations. However, a secondary ice production process peaking around -5 C, and identified as the Hallett-Mossop (HM) process (Hallett and Mossop, 1974) (see Part 1, section 3.2) has to be accounted for with a dedicated parameterisation in the temperature range -10 C to -3 C. Except from WSM5 and WDM6, which do not account for that process, the Milbrandt, Thompson and Morrison schemes use the same parameterisation for that mechanism (Reisner et al., 1998). It is derived using the equations of collection of water droplets by snow and graupels, multiplied by a temperature-dependent parameter based on Figure 2 of (Hallett and Mossop, 1974). Figure 11 shows the median number density of ice crystals and snow particles hydrometeors

per temperature bin along all the flight tracks. The primary ice production peak which relies on the INP parameterisations clearly appears for the three schemes although with different amplitudes, that can be related to the above discussion on the various efficiencies of the INP parameterisations. In the HM temperature regime the Thompson scheme does show a clear increase of the number density around -5 C, while the Morrison schemes does not. The Milbrandt scheme shows more or less steady and larger concentrations than the Thompson, and the Morrison schemes, but with no clear signature of the HM process. The one to two orders of magnitude difference in the predicted total number concentrations of icy condensates between the Milbrandt scheme, and the

other schemes (Figure 10b) below -10 C suggests that the INP parameterisations used in the Milbrandt scheme would blur any possible signal from the HM parameterisation (the -5 C peak). The fact that the Morrison scheme does not show the HM process while the Thompson scheme does can be explained by a threshold effect. The mechanism is triggered only if the snow content is > 0.1 g kg1  and LWC > 0.5 g kg1 , values 5  which are above simulated averages for the snow and for LWC in the Morrison scheme (Figure 4c, and Figure 6). Interestingly the median number of ice and snow hydrometeors for the Thompson

scheme agree within a factor of 2 with the observed median around -5 C. This shows that the Hallett-Mossop process can be to some extent correctly accounted for at this spatial resolution with the current parameterisation.

[Figure]

**Figure 11.** Median of the observations versus the temperature, as long as the simulated total number concentrations of hydrometeors along the flight tracks for Morrison, Thompson, and Milbrandt.

[revised manuscript text omitted]